# The transcriptional coactivator RUVBL2 regulates Pol II clustering with diverse transcription factors

Hui Wang[1,2,12], Boyuan Li [1,12], Linyu Zuo[3], Bo Wang[4], Yan Yan[5,6], Kai Tian[1], Rong Zhou[1], Chenlu Wang[1], Xizi Chen[7], Yongpeng Jiang[1], Haonan Zheng[1], Fangfei Qin[8], Bin Zhang [9], Yang Yu [10], Chao-Pei Liu [11], Yanhui Xu [7], Juntao Gao[5,6], Zhi Qi [3], Wulan Deng [4] & Xiong Ji [1] ✉

RNA polymerase II (Pol II) apparatuses are compartmentalized into transcriptional clusters. Whether protein factors control these clusters remains unknown. In this study, we find that the ATPase-associated with diverse cellular activities (AAA + ) ATPase RUVBL2 co-occupies promoters with Pol II and various transcription factors. RUVBL2 interacts with unphosphorylated Pol II in chromatin to promote RPB1 carboxy-terminal domain (CTD) clustering and transcription initiation. Rapid depletion of RUVBL2 leads to a decrease in the number of Pol II clusters and inhibits nascent RNA synthesis, and tethering RUVBL2 to an active promoter enhances Pol II clustering at the promoter. We also identify target genes that are directly linked to the RUVBL2-Pol II axis. Many of these genes are hallmarks of cancers and encode proteins with diverse cellular functions. Our results demonstrate an emerging activity for RUVBL2 in regulating Pol II cluster formation in the nucleus.

RNA polymerase II (Pol II) is concentrated in specific regions of the nucleus, which were previously named transcription factories or Pol II clusters. These clusters consist of dozens of Pol II molecules, with the exact number depending on the quantitation method used and cell type analyzed[1–5]. Many functions have been attributed to for these clusters, such as transcriptional coordination among different genomic loci, enhancement of gene expression efficiency, 3D genome organization, and facilitation of chromosome translocation in cancers[6–10]. Increasing evidence suggests that Pol II clusters is directly connected to transcriptional activation or genome organization[11–15]; therefore, investigations into the molecular mechanism by which Pol II clustering is controlled are important.

Liquid-liquid phase separation has recently become one of the most exciting topics in cell biology, impacting the understanding of many fundamental aspects of biology[16–21]. In this respect, transcription factors, coactivators, and Pol II have been proposed to form transcriptional condensates in the nucleus[22–27]. Specifically, the carboxy-terminal domain (CTD) of the largest Pol II subunit (RPB1) is composed of 52 heptad repeats, which are highly disordered low complexity

domains (LCDs) involved in Pol II clustering, likely through a liquid-liquid phase separation mechanism. Recent studies have suggested that CTD phosphorylation and RNA lead to the Pol II release from transcriptional condensates[12,25,28,29]. Although interactions among LCDs (such as the transactivation domains in transcription factors and RPB1 CTD) likely contribute to Pol II cluster formation, it is difficult to imagine how these weak and transient interactions could be sufficient for pre-initiation complex assembly and robust transcription initiation at promoters. One explanation involves unknown factors that may dynamically promote the weak interactions among LCDs. Inspired by recent evidence showing that molecular chaperones and ATP modulate biomolecular condensate formation in cells[30–34], we aimed to identify protein factors that regulate Pol II clustering in mammalian cells.

The highly conserved ATPase-associated with diverse cellular activities (AAA + ) ATPase family member RUVBL2 (also known as Reptin or Tip48) functions as a molecular chaperone to prevent the formation of protein aggregates[35,36]. RUVBL2 forms part of the PAQosome/R2TP complex for Pol II assembly in the cytoplasm, and

persistent defects in Pol II assembly lead to Pol II accumulation in the cytoplasm and decreased levels of Pol II in the nucleus[37,38]. RUVBL2 can form heterohexamers or heterododecamers (RUVBL complex) with RUVBL1 (also known as Pontin or Tip49). A previous study showed that the RUVBL complex remained associated with the Pol II holoenzyme through multiple chromatographic purifications, suggesting that the RUVBL complex plays a role in regulating Pol II[39], but its relevance to chromatin and function is unclear.

RUVBL2 plays an essential role in transcriptional regulation. In the classical model, RUVBL2 cooperates with specific oncogenic transcription factors and recruits Pol II, chromatin remodelers, or signaling factors to drive specific gene transcriptional programs[40]. However, this model lacks a clear mechanistic explanation of how RUVBL2 connects transcriptional and posttranscriptional multimolecular complexes to gene expression regulation. In addition, the results of most previous studies depended on experiments involving gene overexpression, gene knockdown by RNAi, or mutations coupled with reporter expression; measurements obtained by RT–qPCR or RNA-Seq analysis, usually with long-term perturbation leading to results that are difficult to interpret. Moreover, high-quality RUVBL2-chromatin binding datasets are lacking, which makes distinguishing transcriptional and posttranscriptional regulatory effect a challenge.

Investigations into transcription regulation mechanisms have usually focused on Pol II initiation, elongation, and/or termination. The regulation of Pol II clustering at promoters has been proposed as a mechanism for transcription regulation[12,41], but the factors that control Pol II clustering to induce general transcription activation remain unknown. Our work demonstrates that RUVBL2 directly regulates Pol II clustering and transcription activation. Mechanistically, RUVBL2 enhanced the co-phase separation of the RPB1 CTD and transcription factors (i.e., EWS-FLI1). By performing time series analyses of nascent and mature transcriptomes after rapid protein degradation, we identify 45 direct transcriptional targets (i.e., *c-Myc*, *Bmp4* and *Junb*, etc.) of the RUVBL2-Pol II axis in mouse embryonic stem cells (mESCs). Furthermore, we find that these genes are involved in diverse molecular functions, and postulate that the RUVBL2-Pol II axis may contribute to various cellular functions under different biological circumstances. Our work provides a foundation for investigating RUVBL2-mediated Pol II clustering and functions in various biological systems, as it suggests that RUVBL2 might sense environmental stimuli, developmental cues, or disease signals to modulate Pol II cluster-related functions.

## Results

### RUVBL2 interacts with the unphosphorylated RPB1 CTD on chromatin

To systematically identify candidate protein factors that regulate Pol II clustering, Pol II chromatin immunoprecipitation with mass spectrometry (ChIP-MS) with mESCs was performed. We identified protein factors that were preferentially enriched in Pol II ChIP-MS and H3K27ac/H3K4me3 ChIP-MS data that had been previously obtained with the same mESC line[42]. We found that RUVBL1 and RUVBL2 were among these enriched proteins, suggesting that they may play roles in regulating Pol II on chromatin (Fig. 1a). RUVBL1 and RUVBL2 largely overlap at active gene promoters (Supplementary Fig. 1a–c). As RUVBL2 usually forms heterohexamers or heterododecamers with RUVBL1, we focused mainly on RUVBL2 in this study, including RUVBL1 as a control for comparison.

We performed RUVBL2 ChIP-MS, and the candidate RUVBL2-interacting proteins in mESCs were identified through a stringent cutoff peptide number ratio with high confidence (Supplementary Data 1, Supplementary Fig. 1d). Many transcription-related proteins were detected in the RUVBL2 ChIP-MS preparations (Supplementary Data 1), and we determined that Pol II subunits were preferentially enriched in the RUVBL2 ChIP-MS preparations (Fig. 1b). The ChIP-Seq heatmap signals consistently showed that Pol II and RUVBL1/2

colocalized in the genome (Fig. 1c). The size exclusion chromatography also confirmed the copurification of RUVBL1/2 and Pol II holoenzymes in the chromatin fractions (Fig. 1d–f). As reported previously, the RUVBL1/2 complex is a molecular chaperone of multifunctional protein complexes, including chromatin remodelers[40,43]. We identified INO80, and TIP60/P400, in addition to Pol II in RUVBL2 ChIP-MS preparations (Supplementary Fig. 1d, Data 1). Western blot analyses showed that RPB1 interacted with RUVBL1/2 but not INO80 or P400 under stringent wash conditions (Fig. 1g), which was consistent with previous studies showing that SWI/SNF remodelers interact with Pol II but not INO80 or P400[44–48].

To gain further mechanistic insights into the interactions between RUVBL1/2 and Pol II, we analyzed the state of Pol II phosphorylation through RUVBL1/2 immunoprecipitation with native chromatin fractions. The results showed that RUVBL1/2 preferentially interacted with unphosphorylated Pol II but not with Pol II phosphorylated at serine 2 or serine 5 in the chromatin fraction (Fig. 1h). This result suggests that the RPB1 CTD may mediate the interactions between Pol II and RUVBL1/2. A recent study showed that the insertion of a degron tag into the RPB1 C-terminus led to the specific degradation of the RPB1 CTD but not full-length RPB1[49]. We confirmed this with our RPB1 CTD degron cells (Fig. 1i)[14]. Then, Pol II immunoprecipitation was performed with antibodies that recognized the N-terminal domain (NTD) of RPB1 in the chromatin fractions obtained from cells with and without RPB1 CTD. Western blot results showed that the interactions of RUVBL1/2 with Pol II in the chromatin fraction were dependent on the RPB1 CTD (Fig. 1i).

RUVBL2 is part of the PAQosome/R2TP complex, which assembles the Pol II complex in the cytoplasm[37,40]. We also examined the interactions of RUVBL2 and another R2TP subunit, RPAP3, with Pol II in isolated cytoplasm and nucleoplasm fractions and found that RUVBL2 interacted with Pol II in both fractions, while RPAP3 interacted with Pol II mostly in the cytoplasmic fraction, but not in the chromatin fractions (Supplementary Fig. 1e). Previous studies showed that the RUVBL2-containing R2TP complex assembles the Pol II complex through the RPB5 subunit in the cytoplasm[50,51]. This is in contrast to our finding that RUVBL2 interacts with the RPB1 CTD in chromatin, suggesting that RUVBL2 regulates chromatin-associated Pol II, which seems to be different from the R2TP complex-mediated Pol II assembly in the cytoplasm, but it is also possible that RUVBL2 binds Pol II and cotransport into the nucleus. As RUVBL1/2 are components in INO80 and TIP60 remodelers, we performed ChIP–qPCR analyses of Pol II, RUVBL2, INO80, and P400 1 h after RPB1 CTD depletion. The ChIP–qPCR results showed that Pol II and RUVBL2 binding to chromatin was significantly decreased at the RUVBL2-targeted genes identified by ChIP-Seq but not at the intergenic region (Fig. 1j). In contrast, neither INO80 nor P400 binding to chromatin was obviously altered (Fig. 1j). These results point to a potential nonchromatin remodeling role for RUVBL2.

### Immediate depletion of nuclear RUVBL2 decreases the number of Pol II clusters

We next sought to determine whether RUVBL2 depletion exerts adverse effects on Pol II cluster formation in cells. We first examined and ensured that the GFP and degron tags did not affect RUVBL2 ChIP-Seq signals or gene expression compared with those in wild-type mESCs (Supplementary Fig. 2a–c). Then, a time series of auxin-inducible RUVBL2 protein degradation assays were performed with mESCs. After auxin treatment for 0.5–1 h, nuclear RUVBL2 was mostly degraded (Fig. 2a, Supplementary Fig. 2d). The number of Pol II clusters was decreased by 34 and 60% after auxin treatment for 0.5–1 h, respectively (Fig. 2a, b). The nuclear Pol II levels were not obviously changed immediately after auxin treatment (Fig. 2c, Supplementary Fig. 2d) because the accumulative effect on the Pol II level in the nucleus may take time to manifest. We also performed immunofluorescence analysis of CTCF. CTCF formed protein clusters in the

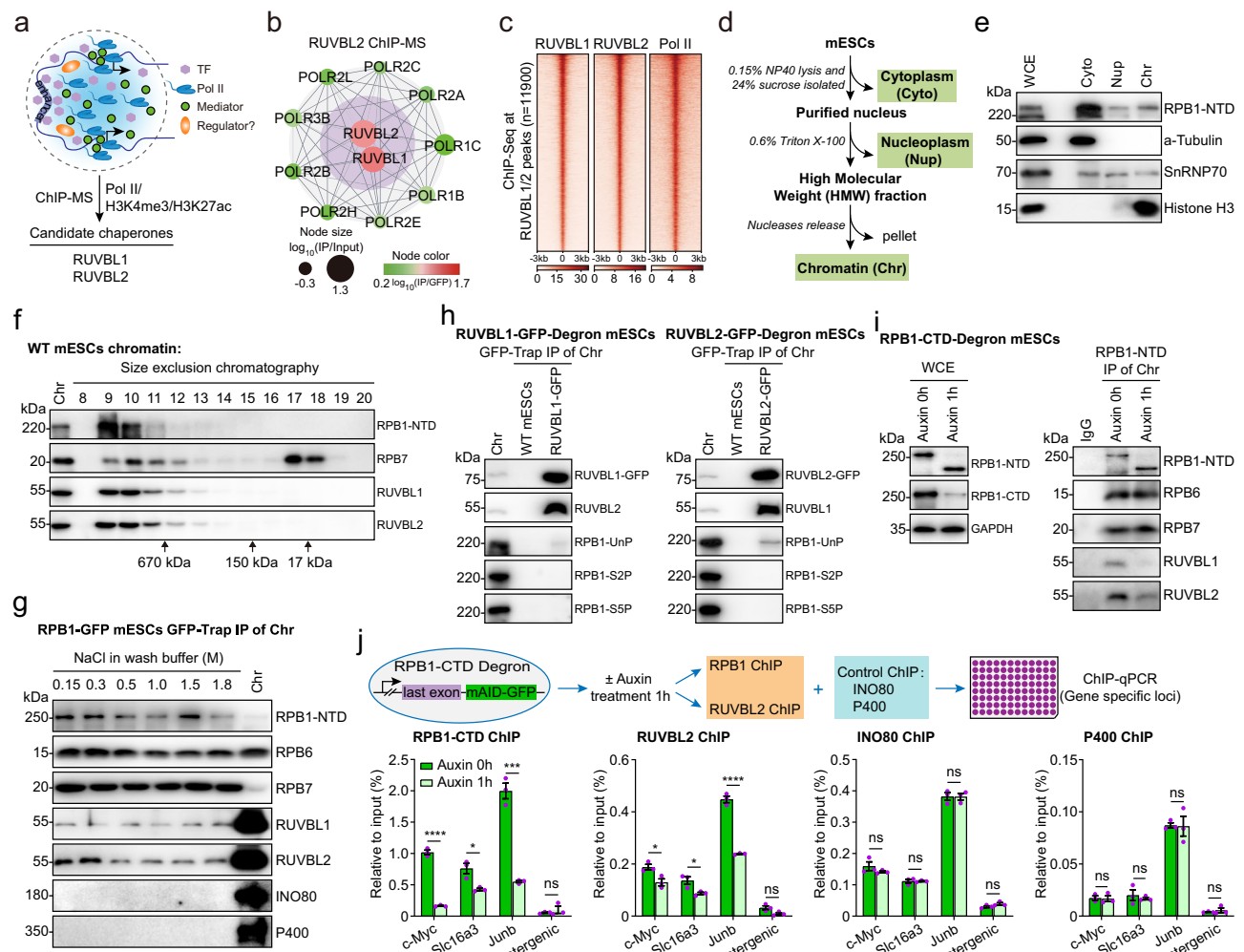

**Fig. 1 | RUVBL2 interacts with RPB1 CTD on chromatin. a** An illustration of the Pol II-mediated clusters. RUVBL1/2 were identified by Pol II, H3K27ac, and H3K4me3 ChIP-MS. **b** RNA polymerase I/II/III subunits were detected in RUVBL2 ChIP-MS preparations. The lines indicate the protein-protein interactions detected in the STRING database. **c** Heatmap illustrating that RUVBL1 and RUVBL2 globally colocalized with Pol II on the mESC genome. **d** Schematic diagram showing the subcellular fractionation (cytoplasm, nucleoplasm, and chromatin) procedure, which was adopted from Damianov et al.[112]. **e** Western blotting was performed to examine the different subcellular fractions of mESCs. a-Tubulin, histone H3, and snRNP70 were used as marker proteins of the cytoplasm, chromatin, and nucleoplasm, respectively. WCE indicates whole-cell extracts. **f** Chromatin fractions in wild-type mESCs were separated through size exclusion chromatography, and western blotting was performed to identify RPB1, RPB7, RUVBL1, and RUVBL2 in the gradient eluents based on their different molecular weights. **g** Protein-protein

interactions between Pol II (RPB1) and RUVBL1 or RUVBL2 were examined under different concentrations of NaCl. **h** The phosphorylation states of Pol II interacting with RUVBL1/2 using RUVBL1 and RUVBL2 degron mESCs were examined. RPB1-CTD or RPB1-Unp indicates the unphosphorylated RPB1 CTD, and RPB1-S2P and RPB1-S5P indicate RBP1 phosphorylated at serine 2 and serine 5 in the CTD, respectively (lower panel). **i** The RPB1 CTD was acutely degraded, and the native chromatin fractions were subjected to an antibody recognizing the NTD of RPB1 during immunoprecipitation. **j** Schematic diagram showing the method used to confirm that RUVBL1/2 interacted with RPB1 CTD on chromatin (upper panel). The ChIP-qPCR data shown represent 3 biological replications. Two-tailed Student's *t* tests were performed to determine the significance of differences between groups, data are presented as mean +/- SDs (ns, not significant, *p < 0.05, ***p < 0.001, ****p < 0.0001), the exact p values can be found in the source data.

nucleus, as reported previously[52–54], but did not show a trend similar to that of Pol II clusters after RUVBL2 depletion (Fig. 2a, b). These findings suggest that RUVBL1/2 promoted the formation of Pol II clusters in the nucleus.

We next used chemical inhibitors to further investigate the mechanisms underlying Pol II clustering regulation. To obtain direct evidence of a causal relationship, we performed Pol II clustering analyses after transcription was inhibited with actinomycin D. The Pol II clusters increased in size and number and exhibited aggregate-like morphology after actinomycin D treatment (Supplementary Fig. 2e, f). These results were consistent with a recent finding showing that RNA-mediated feedback controlled transcriptional condensate formation; specifically, transcription initiation leads to low levels of nascent RNA production and stimulates transcriptional condensate formation,

whereas transcription elongation leads to high levels of RNA production and dissolves transcriptional condensates[29]. To mimic defective Pol II assembly in the cytoplasm, inhibited nuclear import of Pol II with importazole and then analyzed Pol II clustering. The number of Pol II clusters in mESCs treated with importazole for a short time (such as 1 h) did not change, but the number of Pol II clusters as well as protein abundance, was decreased in mESCs treated for a long time (such as 6 h) (Supplementary Fig. 2e, f). These findings suggest that inhibited nuclear import did not immediately reduce nuclear Pol II cluster formation but after long-term perturbation, it caused a reduction in Pol II clusters.

Recent studies have proposed that transcription factors and coactivators form phase-separated condensates to concentrate the transcription apparatuses at super-enhancers and that Pol II is

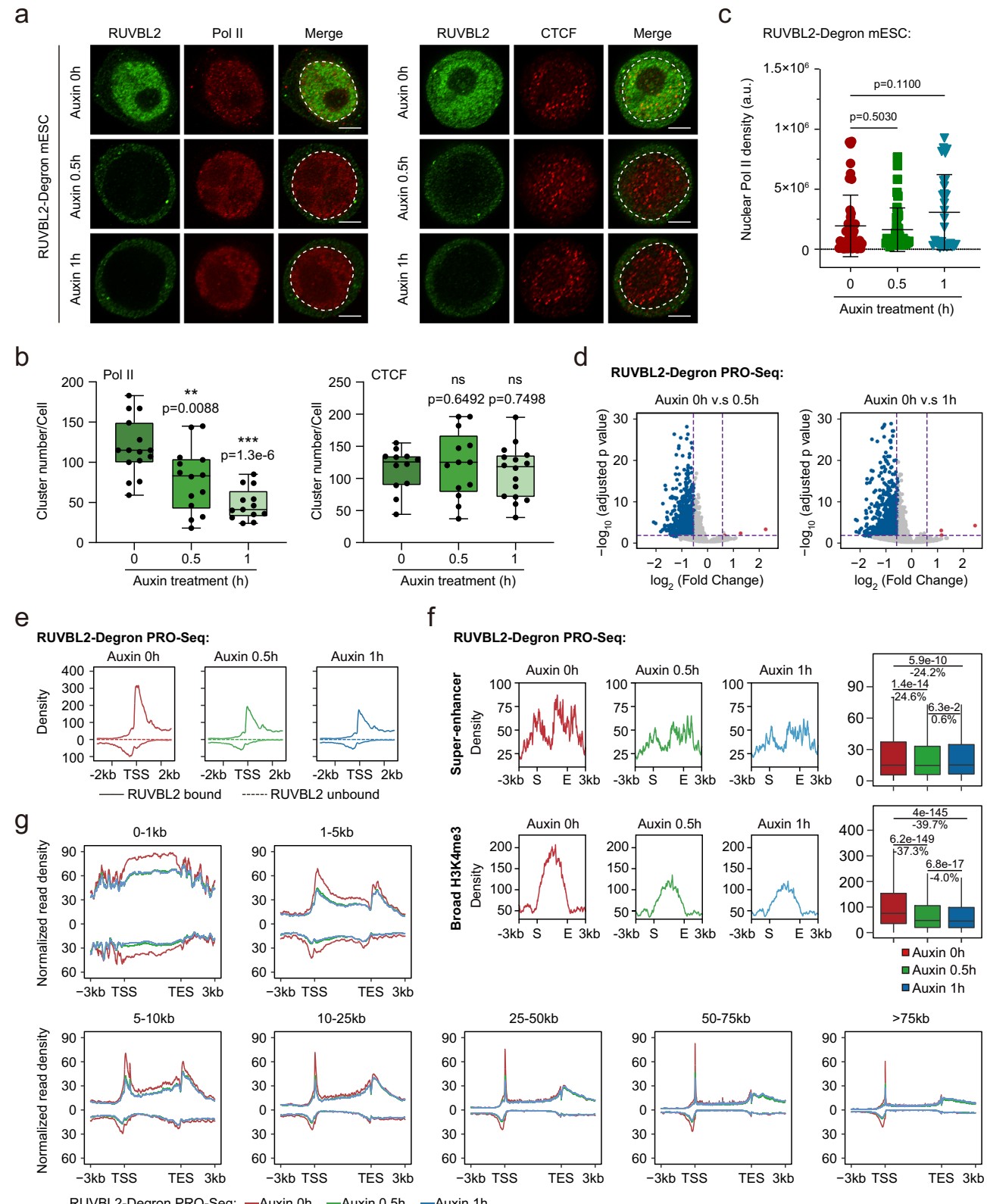

incorporated into these condensates and subsequently released after CTD phosphorylation[23,24,28]. We performed PRO-Seq after auxin treatment for 0.5–1 h. Global analyses indicated that auxin treatment for 0.5–1 h led to global transcriptional repression (Fig. 2d, e). Although RUVBL2 is enriched at promoters and super-enhancers usually contact promoters through chromatin looping[55–57], we investigated the molecular effects of RUVBL2 depletion at super-enhancers. We also

analyzed the overall effect on broad H3K4me3 promoters, which usually indicate enhanced transcriptional consistency[58]. The signals of nascent RNA at super-enhancers and broad H3K4me3 promoters were also decreased after RUVBL2 depletion (Fig. 2f). These results indicated that RUVBL2 was required for Pol II activity at super-enhancer and promoter regions, which was consistent with the decrease in the number of Pol II clusters after RUVBL2 depletion. Additionally, auxin

**Fig. 2 | RUVBL2 depletion decreases the number of Pol II clusters and inhibits global transcription. a** Representative confocal microscopy images of Pol II (left) and CTCF (right) after RUVBL2 degradation. Scale bar, 3 μm. **b** The number of clusters of Pol II (left) and CTCF (right) per cell was quantified and shown as box-plot, all the cells data were shown as single black points (Pol II $n = 15, 14, 13$, respectively; CTCF $n = 12, 13, 16$, respectively). Two-tailed Student's $t$ tests were performed to determine the significance. (ns, not significant, ** $p < 0.01$, *** $p < 0.001$). **c** The nuclear densities of Pol II per cell were measured in untreated and auxin-treated cells. The data represent 3 biological replicates and cells from different fields ($n = 45, 43, 29$, respectively) were used for each quantification. Two-tailed Student's $t$ tests were performed to determine the significance of differences between groups, and the error bars indicate the mean +/− SDs (* $p < 0.05$, ** $p < 0.01$). **d** Volcano plot illustrating the results of the differential gene expression

analysis using PRO-Seq after RUVBL2 depletion. FDR < 0.05 and |Fold Change (FC)| > 1.5 are used to reported the differential genes. The FDR are extracted from DESeq2 by adjusting $p$-values using BH method. **e** The results of metagene analyses of the PRO-Seq signals. The profile above 0 indicates the signal on the sense strand, the profile below 0 indicates the signal on the antisense strand. **f** The PRO-Seq profiles of the super-enhancers (upper) and broad H3K4me3 regions (lower). Merged sense and antisense strand intensities at super-enhancers ($n = 232$) and broad H3K4me3 ($n = 1019$) regions were quantified and are displayed as box plots in the right panel. Two-tailed Wilcoxon tests were performed to calculate the significance of differences (ns, not significant, *** $p < 0.001$) and the exact $p$ value can be found in the source data. **g** The graph illustrates the PRO-Seq signal changes at different gene length intervals. The profile above 0 indicates the signal on the sense strand, the profile below 0 indicates the signal on the antisense strand.

treatment for 0.5–1 h led to decreased PRO-Seq signals at both the promoter-proximal regions and gene bodies of both short and long genes, suggesting transcription initiation inhibited (Fig. 2g). Together with the unchanged levels of nuclear Pol II after immediate auxin treatment (Fig. 2c, Supplementary Fig. 2d), these results indicate that the immediate depletion of RUVBL2 (0.5–1 h) in our system did not change nuclear Pol II abundance but decreased the number of nuclear Pol II clusters and inhibited transcription initiation.

Pol II foci were small and dynamic, and they formed clusters before transcription initiation in living cells[5]. We then examined the small dynamic Pol II clusters in living cells by time-correlated photo-activated localization microscopy (tcPALM) after acute depletion of RUVBL2. mEos3.2 was tagged in the N-terminus of RPB1, which did not affect the protein level of RPB1 in the mESCs (Supplementary Fig. 3a, b). Our Pol II tcPALM live-cell imaging exhibited a good signal-to-noise ratio, and both transient and stable clusters were observed, consistent with the previous report[59] (Fig. 3a, Supplementary Fig. 3c). Notably, within 1 h of RUVBL2 depletion, the number of both transient and stable clusters was significantly decreased (Fig. 3b–e). We also performed stimulated emission depletion microscopy (STED) imaging of the initiation factor TBP and found that RUVBL2 contributed to pre-initiation cluster formation in the nucleus (Fig. 3f, g, Supplementary Fig. 3d). These results demonstrate that RUVBL2 is necessary for the formation of preinitiation Pol II clusters in living cells.

## Tethering RUVBL2 to active gene promoters enhances Pol II cluster formation

As RUVBL2 depletion led to a decreased number of Pol II clusters, we next investigated whether RUVBL2 was sufficient for the formation of Pol II clusters in cells. We used a well-documented cellular system, U2OS-2-6-3 cells, with a stably integrated doxycycline (Dox) response transgene[60,61]. Transient expression of Lac repressor-BFP (LacR-BFP) and rtTA in the presence of Dox allowed the simultaneous visualization of Lac operator (LacO) DNA repeats, the tethered protein (LacR-BFP fusion), and the recruited protein (green fluorescence) (Fig. 4a). The LacO recruitment reporter was a tet-on driven mini-CMV promoter, and it was silent in the absence of Dox. We noted that the LacO loci visualized by LacR fusion-emitted fluorescence were enlarged and varied in size among cells as the silent heterochromatin at the LacO loci was increasingly loosened after transcriptional activation by Dox. The signals emitted by the recruited protein also varied among cells, possibly because of stochastic gene expression or variations in trans-fection efficiency (Fig. 4b). The intensities of the recruited-proteins at the LacO position were normalized to that of the averaged recruited-protein signals at the non-LacO regions in the same cell using a method similar to that published previously[60–63], which ruled out the effect of cell-to-cell variation in the expression of transfected proteins.

We first tested whether RUVBL1 or RUVBL2 was recruited to active transcription sites by transiently expressing LacR-BFP and rtTA. The results showed that RUVBL1 and RUVBL2 were concentrated at active LacO loci after Dox treatment but not after mock treatment (Fig. 4b, c),

suggesting that the presence of RUVBL1/2 was correlated with tran-scriptional activation. We next sought to determine whether RUVBL1 or RUVBL2 tethering has effects on Pol II at promoters by transiently expressing LacR-BFP fused to RUVBL1 or RUVBL2. The Pol II signals were visualized by IF of initiation-associated unphosphorylated Pol II. The control was the BFP fluorescence intensity at LacO loci, which was similar among cells transfected with different RUVBL plasmids under mock conditions (Fig. 4d), which implied that the amount of protein tethered to LacO loci was saturated under the test conditions. The results under the mock condition (when the gene was silenced) revealed no increase in Pol II signals at LacO loci after RUVBL1 or RUVBL2 tethering (Fig. 4e), suggesting that neither RUVBL1 nor RUVBL2 was sufficient to recruit Pol II.

Tethered wild-type RUVBL1 or RUVBL2 was associated with enhanced Pol II signal intensity (a 42 % increase for RUVBL1 and a 61 % increase for RUVBL2) at LacO loci when the gene was activated with Dox (Fig. 4f), implying that RUVBL1 or RUVBL2 promoted Pol II clus-tering when the transactivation domain of rtTA was activated. If tethering of RUVBL2 promotes the known R2TP-dependent function of RUVBL2 on Pol II, we would anticipate an increase in nuclear Pol II level due to an increased level of assembled Pol II complex. Therefore, we quantified Pol II density in the nucleus with and without RUVBL2 tethering and found no significant change (Fig. 4g). These data suggest a function of RUVBL2 in Pol II clustering in U2OS-2-6-3 cells. In addi-tion, we performed CTD tethering to LacO loci. The IF results showed that CTD tethering to LacO loci led to the recruitment of a small amount of endogenous RUVBL2 to the LacO loci without transcription activation, but the amount was significant compared with that when only BFP was tethered to LacO loci (Fig. 4h).

## RUVBL1/2 directly promote RPB1 CTD clustering and Pol II transcription initiation in vitro

We then wondered whether RUVBL1/2 regulate Pol II clustering directly. To investigate this, RUVBL1 and RUVBL2 were coexpressed in bacteria and purified, and RPB1 CTD mEGFP fusion proteins were purified according to the procedure previously published[22,64,65]. We used size-exclusion chromatography to isolate RUVBL1 and RUVBL2 oligomer of the correct size. RPB1 CTD mEGFP was immobilized on GFP Trap beads, incubated with purified RUVBL1/2 proteins, and examined by western blotting after extensive washing (Fig. 5a). The results showed that RUVBL1/2 directly interacted with the RPB1 CTD in a concentration-dependent manner but not with similarly purified mEGFP, which was a negative control (Fig. 5b). Since the RPB1 CTD consists of heptapeptide repeats and forms protein clusters in vivo and in vitro[22], we wondered whether RUVBL1/2 exerted a direct impact on RPB1 CTD clustering. Indeed, our RPB1 CTD formed clusters in vitro, similar to previous reports[12,22,28]. The area and condensate fractions of RPB1 CTD clusters increased with increasing RUVBL1/2 concentration (Fig. 5c–e). The in vitro purified mCherry proteins used as controls did not exert an effect (Fig. 5e and Supplementary Fig. 4a). Interestingly, RUVBL1/2 tended to localize to the periphery of the RPB1 CTD clusters

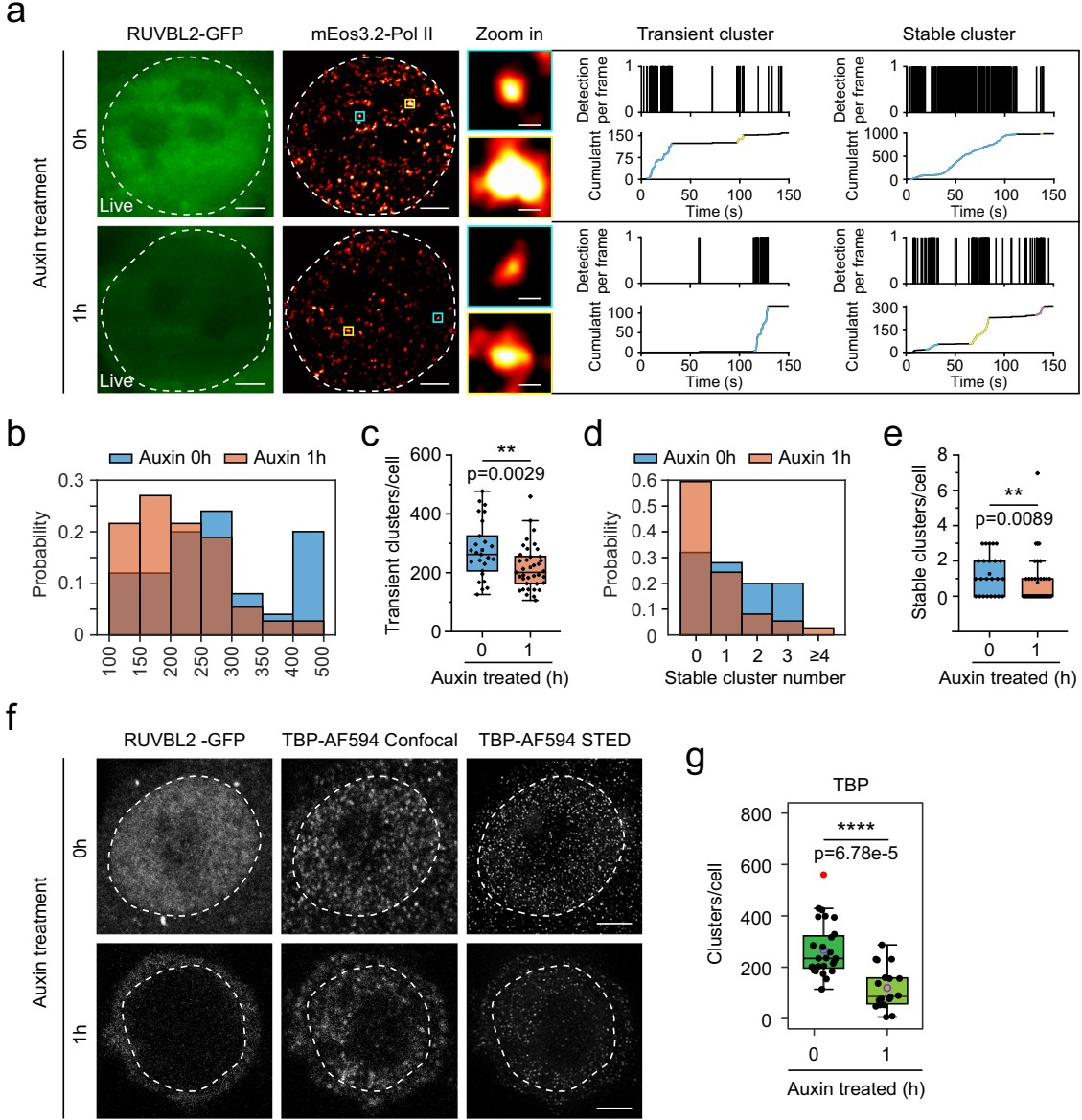

**Fig. 3 | RUVBL2 regulates the preinitiation Pol II clusters. a** Super-resolution image of endogenous Pol II labeled with mEos3.2 in living RUVBL2 and degron mESCs (left) and representative images of transient and stable Pol II clusters and corresponding time-correlated photoactivation localization microscopy (tcPALM) traces (right). Transient and stable Pol II clusters correspond to areas boxed in blue and yellow, respectively. The dashed box indicates the nucleus; scale bars, 1 μm in the whole-cell in the super-resolution image (left), and 200 nm in the transient and stable Pol II cluster images (right). **b** The distributions of the number of transient Pol II clusters before and after RUVBL2 degradation. **c** The transient clusters per cell ($n = 25$ for auxin treatment at 0 h and $n = 37$ for auxin treatment after 1 h) were calculated, and two-tailed Student's $t$ tests were performed for determining statistical significance (box plot), **$p < 0.01$. **d** The number of stable Pol II cluster

distributions observed before and after RUVBL2 degradation. **e** The stable clusters per cell ($n = 25$ for auxin treatment at 0 h and $n = 37$ for auxin treatment after 1 h) were calculated and statistically analyzed (box plot). Two-tailed Student's $t$ test was performed to calculate statistical significance. **$p < 0.01$. **f** TBP was examined using STED and representative images are shown before and after auxin treatment, the scale bar is 3 μm. **g** Cells from at least 10 different fields ($n = 13$ for auxin treatment at 0 h and $n = 18$ for auxin treatment after 1 h) were used to calculate the number of clusters per cell. Box plots indicate median (middle line), 25th and 75th percentile (box) and 5th and 95th percentile (whiskers), average (purple circle) as well as outliers (single red points). Statistical significance was evaluated based on two-tailed Student's $t$ tests. ***$p < 0.001$.

(Fig. 5c), indicating that RUVBL1/2 might have promoted RPB1 CTD cluster formation by trapping Pol II molecules. The effect of 1,6-hexanediol treatment on Pol II clustering has been published[22,24,66–68]. RPB1 CTD clustering was reduced by disrupted ion interactions (by NaCl) and hydrophobic interactions (by 1,6-hexanediol)[22], indicating that ion interactions and hydrophobic interactions might contribute to RUVBL2-mediated Pol II clustering.

To obtain direct functional evidence for RUVBL1/2 interactions with Pol II, we carried out an in vitro transcription initiation assay with purified general transcription factors (TFIIA, TFIIB, TFIID, TFIIF, TFIIE,

and TFIIH), Pol II, RUVBL1/2, and the HDM2 promoter by following a previously published protocol[69–72]. Notably, we observed an increase in transcription initiation activity after the amount of RUVBL1/2 protein was increased (Fig. 5f, g). In contrast, no similar enhancement was observed with negative controls (GFPs or in the absence of NTP). These results demonstrate that RUVBL1/2 play a direct role in promoting Pol II transcription initiation.

To achieve further mechanistic insights into the interactions of RUVBL1/2 with Pol II, we took advantage of a previously established system consisting of the oncogenic transcription factor EWS-FLI1 and

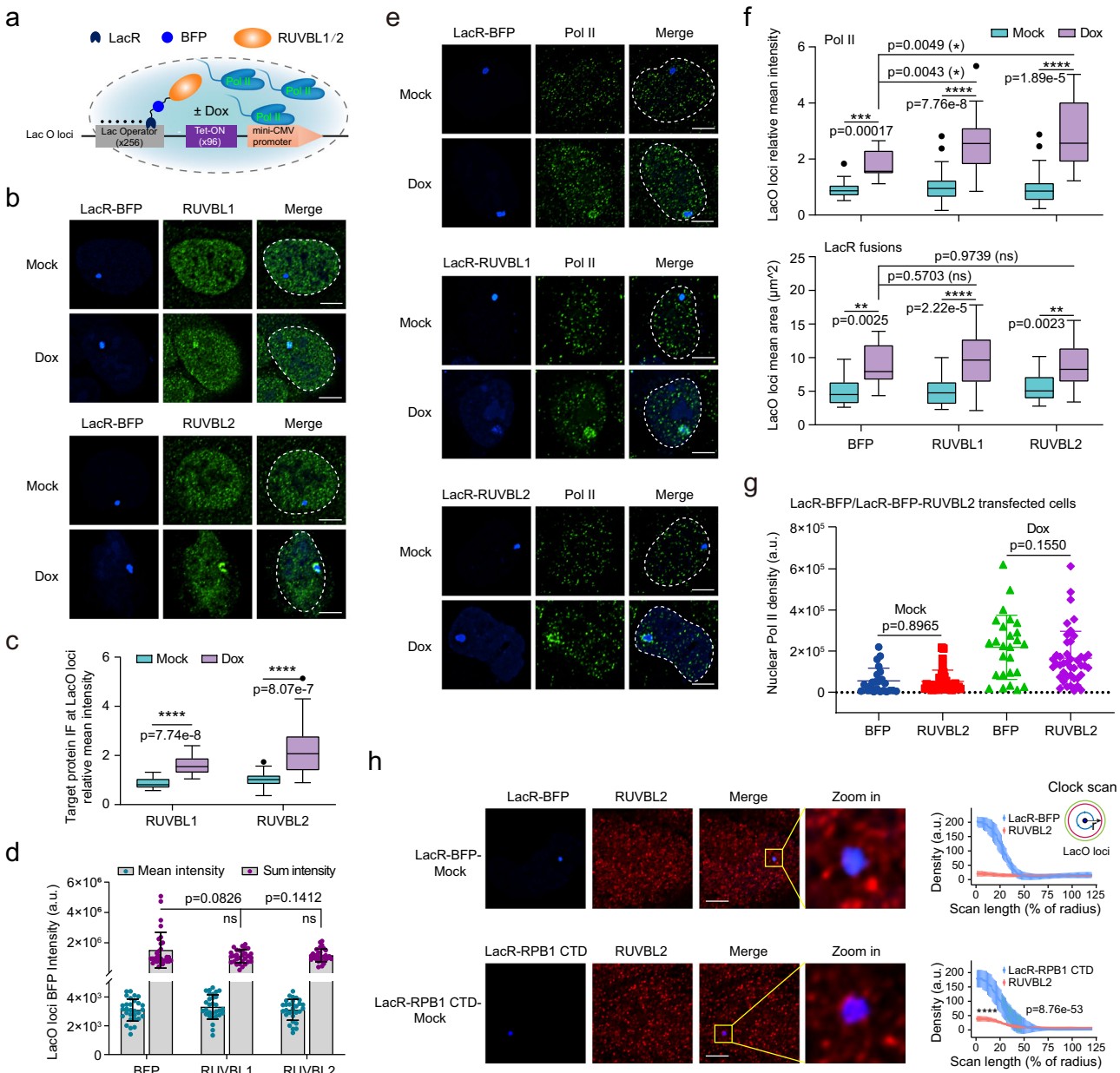

**Fig. 4 | Tethering RUVBL2 to an active gene promoter enhances Pol II cluster formation. a** Schematic showing the system used for simultaneous imaging of tethered proteins and recruited proteins, as published previously[60].
**b** Representative images of endogenous RUVBL1 and RUVBL2. Scale bar, 5 μm.
**c** The relative IF signals of the target proteins at the LacO loci under mock (RUVBL1 $n = 24$, RUVBL2 $n = 30$) and Dox treatments (RUVBL1 $n = 18$, RUVBL2 $n = 30$). Box plots indicate median (middle line), 25th and 75th percentile (box) and 5th and 95th percentile (whiskers), as well as outliers (single black points), differences between groups were analyzed by two-tailed Student's $t$ tests (****$p < 0.0001$). **d** The mean and total intensity of LacR-BFP fusion signals at LacO loci per cell were plotted in bar graphs. The cotransfected cells ($n = 30$) in different fields from 3 transfection replicates were used for quantification. Differences between groups were analyzed by two-tailed Student's $t$ tests. "ns" indicates no significant difference. The error bars indicate mean +/– SDs. **e** Representative images of endogenous Pol II before

and after transcriptional activation. Scale bars, 5 μm. **f** The relative IF signals of Pol II and LacR-BFP fusions at the LacO loci were plotted as boxplot. Cells from a minimum of 10 different fields (BFP $n = 11$, RUVBL1 $n = 28$, RUVBL2 $n = 29$ in mock state, and BFP $n = 13$, RUVBL1 $n = 24$, RUVBL2 $n = 16$ in Dox activation state) were used for quantification. Differences between groups were analyzed with two-tailed Student's $t$ tests (ns, not significant, *$p < 0.05$, **$p < 0.01$, ***$p < 0.001$, ****$p < 0.0001$). **g** Nuclear Pol II (RPB1) density was quantified from 3 biological replicates. Two-tailed Student's $t$ tests were performed to determine the significance of differences between groups, and the error bars indicate the mean +/– SD. **h** IF assay was performed to examine endogenous RUVBL2. Cells (BFP $n = 22$, CTD $n = 19$) from a minimum of 10 different fields were used for clock scanning and quantification, and the error bars indicate the mean +/– SDs. The clock scan densities were analyzed with two-tailed Student's $t$-tests (****$p < 0.0001$). Scale bars, 5 μm.

RPB1 CTD[73] (both contain LCDs) to perform cophase separation and DNA curtain assays in the presence of RUVBL1/2. The cophase separation assay indicated that RUVBL1/2 promoted the co-condensation of EWS-FLI1 and RPB1 CTD (Fig. 5h, i, RPB1 with 26 hepta repeats was used in the assay). EWS-FLI1 bound the GGAA sequences of the DNA curtain, and RUVBL1/2 increased the number of

RPB1 CTD puncta colocalized with EWS-FLI1 puncta by 64 % with the working buffer flow rate was fast (Fig. 5j–l). We speculate that RUVBL2 enhanced the binding between EWS-FLI1 and RPB1 CTD, indicating that RUVBL2-mediated Pol II clustering might enhance LCD-LCD interactions. However, we could not rule out the possibility the RUVBL2 may slightly recruit Pol II, which would also contribute to the co-phase

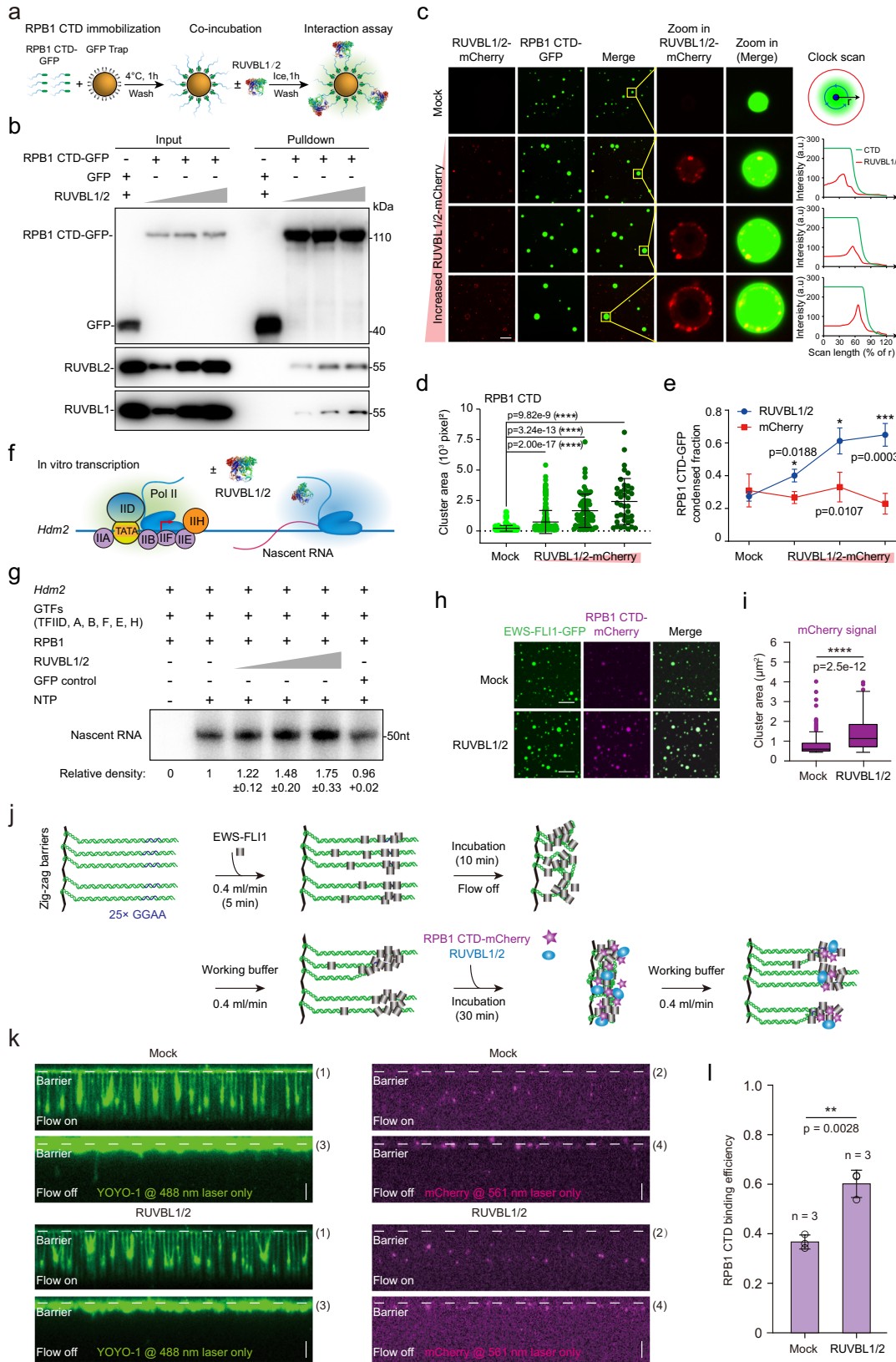

separation signals and DNA curtain signals of RPB1 CTD. We also quantified the nuclear concentration of RUVBL1/2 in mESCs (Supplementary Fig. 4b) and similar concentration of RUVBL1/2 were used for the in vitro experiments. At this concentration, we showed that RUVBL1/2 enhance 48% of in vitro transcription activity, 60% of the condensed fraction of RPB1 CTD clusters, 60% of the condensed

fraction of RPB1 CTD and EWS-FLI1 co-phase separation, 64% of RPB1 CTD puncta counts at EWS-FLI1 puncta recruited to the GGAA regions at DNA curtains. RUVBL2 is a coactivator that does not increase transcription to the same degree as transcriptional activators. For example, recent publications showed that, with the same system as we used in this study, the coactivator MED1 enhanced transcription activity by

**Fig. 5 | RUVBL1/2 directly promote RPB1 CTD clustering and transcription initiation in vitro. a** Schematic diagram showing the RPB1 CTD and RUVBL1/2 pull-down assays. **b** The complexes pulled down by beads were analyzed by western blotting. **c** RUVBL1/2 promoted RPB1 CTD clustering in vitro. The scale bar is 20 μm. **d** Plot showing the cluster area described in (**c**), clusters larger than 100 square pixels were counted (Mock $n = 139$, RUVBL2 in order $n = 300$, 71 and 39, respectively) from no less than 8 different fields for each condition. Two-tailed $t$ tests were performed to determine the significance of differences between groups, and the error bars indicate the mean $+/-$ SD (****$p < 0.0001$). **e** Plot showing the condensed fraction in the different fields per condition presented in (**d**). Two-tailed Student's $t$ tests were used to determine the significance and the data are reported as the mean $+/-$ SD (*$p < 0.05$, ***$p < 0.001$). **f** Schematic diagram showing the in vitro Pol II transcription initiation system. **g** Nascent RNA synthesis by the pre-initiation complex on the HDM2 promoter. This result represents 3 replicates. **h** In vitro droplet assay with GFP-EWS-FLI1 and RPB1 CTD-mCherry. **i** Boxplot showing the droplet area of mCherry signals. The total number of droplets examined in one in vitro droplet experiment was $n = 127$ under the mock condition and $n = 197$ under the RUVBL1/2 addition condition. The bottom edge of the box represents the 25th percentile, and the top represents the 75th percentile. Most data points are covered by whiskers (1.5x interquartile range). Two-tailed Student's $t$ tests were performed to determine the significance of differences between groups (****$p < 0.0001$). **j** Strategy for detecting EWS-FLI1 condensates at sites of DNA and RPB1 CTD recruitment by RUVBL1/2[73]. **k** Wide-field total internal reflection microscopy (TIRFM) images of EWS-FLI1 and RPB1 CTD$_{N26}$-mCherry in DNA Curtain experiments. **l** The efficiency of RPB1 CTD$_{N26}$-mCherry recruitment as described in (**j**). Three independent DNA curtain experiments were repeated. The error bars refer to mean $+/-$ SD. Statistical significance was evaluated by two-tailed Student's $t$ test (**$p < 0.01$); confidence level: 95%.

less than 2-fold[74], and the corepressor RPAP2 repressed transcription by approximately 65%[72], an effect similar to that of our study. In summary, our quantification experiments suggest that the RUVBL2-mediated effects on Pol II clustering and transcription activation in vitro were largely consistent.

Considering previous literature, we found that RUVBL2 E300Q mutation inhibited the ATPase activity of the RUVBL1/2 complex[75]. Domains DI and DIII are involved in oligomerization and ATPase activity, and domain DII interacts with other client partners, as analyses of the crystal structures of RUVBL1/2[76]. We repeated the tethering experiments with an ATPase mutant and truncated RUVBL2 and found that neither the DI-DIII domain nor ATPase mutant exhibited capacity similar to that of wild-type RUVBL2 (Supplementary Fig. 4c–h). DI, DII or DIII domain of RUVBL2 also could not significantly enhance Pol II clustering as the full-length RUVBL2 (Supplementary Fig. 4f–h). We next performed an RPB1 CTD droplet assay with purified RUVBL1/2 ATPase mutant. The results showed that the ATPase mutant could promote RPB1 CTD clustering, but it is slightly less efficient than wild-type RUVBL1/2 (Supplementary Fig. 4i, j). The outcome did not change in the presence or absence of ATP at the working concentration reported previously[65] (Supplementary Fig. 4k, l). Moreover, we performed in vitro droplet assays with RUVBL 1/2 DI-DIII or the RUVBL 1/2 DII domain. Interestingly, the DII and RPB1 CTD clusters almost completely colocalized, and DI-DIII localized to the periphery of the RPB1 CTD clusters (Supplementary Fig. 4k), suggesting that DII may directly interact with the RPB1 CTD, while DI-DIII may contribute to the regulation and interactions at the periphery of the clusters. The results further showed that both RUVBL1/2 DI-DIII and RUVBL1/2 DII contributed to the enhancement of RPB1 CTD clustering in vitro, but their impact was less pronounced than that of full-length RUVBL1/2 (Supplementary Fig. 4l). Together with the tethering experiment with RUVBL2 mutants, the droplet assays provided direct insights into RUVBL2 action in RPB1 CTD clustering in vivo and in vitro. Notably, the in vitro droplet assay included only purified proteins, and the tethering experiments included wild-type endogenous RUVBL1/2 proteins in cells and the tethered 256 repeats. The limitations of both assays may create some biases in different systems. Actually, the RUVBL2 ATPase and truncation mutants partially enhanced RPB1 clustering in vitro but not in the tethering experiment, indicating that these mutants may have exerted dominant-negative effects on Pol II clustering in cells. The detailed molecular mechanisms need further investigations. These results collectively imply that the ATPase activity, domain DII-mediated interaction with Pol II, and domain DI-DIII-mediated oligomerization might together facilitate the Pol II cluster-promoting function of RUVBL2, and further characterizations are still warranted in the future.

## RUVBL2 is required for both transcriptional and posttranscriptional regulation of gene expression

We revealed the activity of RUVBL2 in the regulation of Pol II clustering and then used multiomics techniques to further investigate the molecular functions of RUVBL2. To comprehensively identify genes whose expression is regulated by the RUVBL1/2-Pol II axis, we specifically degraded RUVBL1/2 in mESCs. RUVBL2 degradation was found to cause destabilization of RUVBL1 and vice versa (Supplementary Fig. 5a), as observed previously[77]. Thus, we believe that the effects of RUVBL2 depletion were caused by the disruption of the RUVBL1/2 complex. A cell growth assay was performed after RUVBL1 or RUVBL2 degradation through the aforementioned degron system, and the results showed that RUVBL1 and RUVBL2 were essential for the growth of mESCs (Fig. 6a, Supplementary Fig. 5b). Time series RNA-Seq analyses were then performed after RUVBL2 degradation (Fig. 6b, c).

Two replicates of Poly(A) RNA-Seq datasets were generated for each time point (Supplementary Data 2, 3). Ccnd1 and Ccne2 expression was downregulated, consistent with the roles played by RUVBL1/2 in cell cycle regulation (Fig. 6d, Supplementary Fig. 5c)[78]. Notably, the expression of the oncogene *c-Myc* was particularly sensitive to RUVBL2 degradation (Fig. 6d). The mRNA levels of ER stress genes, such as Chop and Atf4, were increased after RUVBL2 depletion (Fig. 6d, Supplementary Fig. 5c), consistent with the previous finding showing that RUVBL2 was required for repressing the ER stress pathway in *C. elegans* and mammalian cells[79]. The RNA-Seq signals of housekeeping genes were not noticeably changed after RUVBL2 depletion (Supplementary Fig. 5d). Consistent with this finding, the amount of extracted total RNA was similar after RUVBL2 depletion, indicating that the global level of mature RNA was not dramatically changed (Supplementary Fig. 5e). The expression changes identified by RNA-Seq were validated through RT–qPCR (Supplementary Fig. 5f). Additionally, our western blot analyses confirmed that prolonged RUVBL2 degradation (approximately 24 h) caused a decrease in the C-MYC protein level and an increase in the CHOP protein level (Supplementary Fig. 5g). Furthermore, we generated RNA-Seq data after Pol III subunit Polr1c depletion via the same degron system used with mESCs and found that the changes in the expression of our model genes (*c-Myc, Ccnd1, Chop*, and *Atf4*) were different after Polr1c depletion than after RUVBL2 depletion (Supplementary Fig. 5h). Collectively, these results suggest that our RUVBL2 degron RNA-Seq datasets were reliable and biologically relevant.

We next sought to investigate the RUVBL2-regulated molecular cascades through cluster analyses of RNA-Seq gene expression data after depletion at different time points. GO functional analysis of differentially expressed genes indicated enrichment of signal transduction genes in Cluster 2 and enrichment of RNA metabolism genes in Cluster 3 (Fig. 6e). The signal transduction genes enriched in Cluster 1 were upregulated at later time points, and RNA processing and chromosome segregation genes enriched in Clusters 4 and 5 were downregulated at later time points (Fig. 6e). Previous studies showed that RUVBL2 interacts with specific transcription factors, as indicated by the expression of specific genes[40]. We identified the promoter-associated motifs for RUVBL2-affected genes with time-series RNA-Seq data. The results showed that the motif did not exhibit enrichment

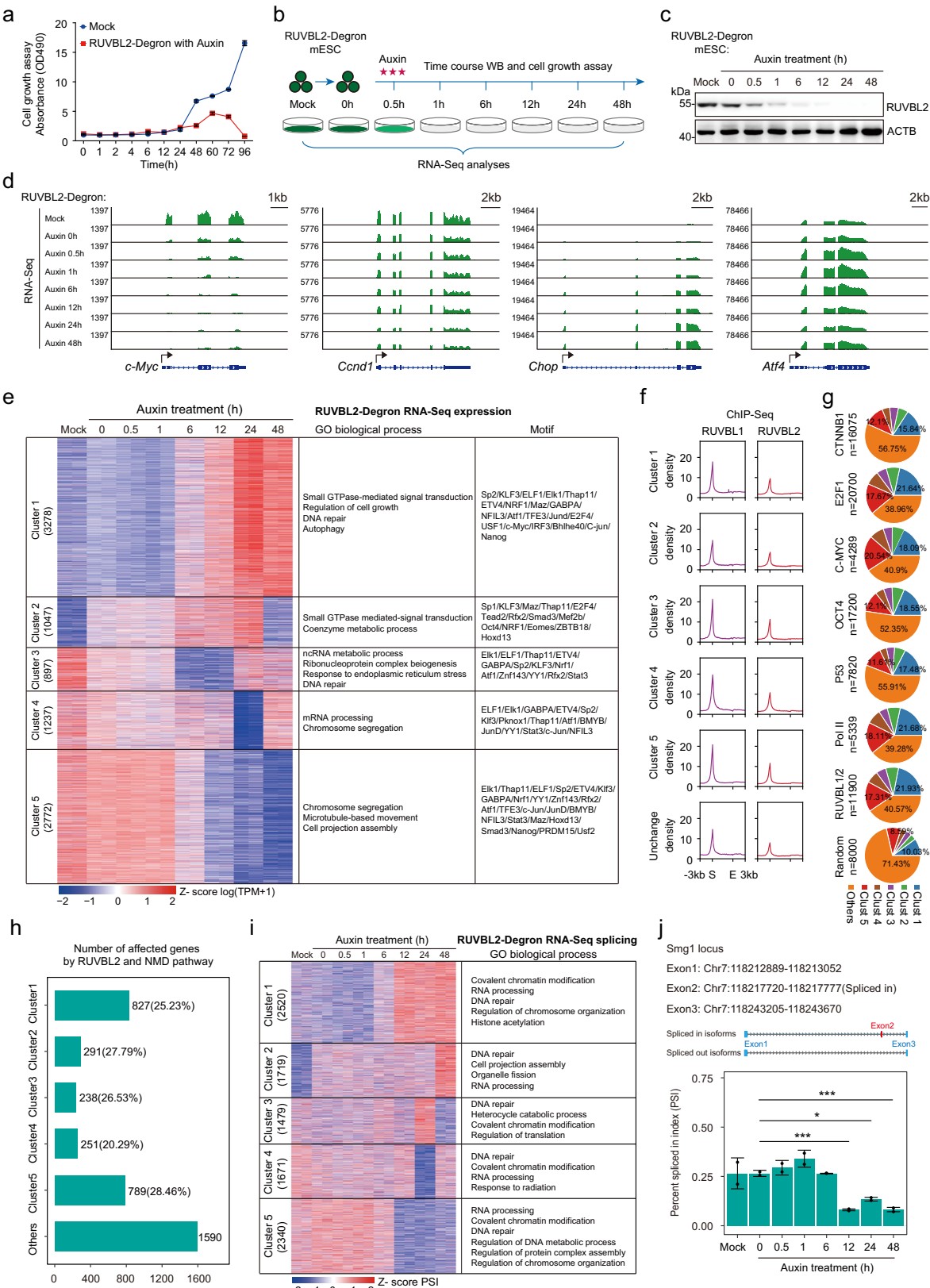

for specific factors (Fig. 6e). We showed that RUVBL2 depletion led to global transcriptional repression. We then found that the genes that were upregulated (Clusters 1 and 2) after RUVBL2 depletion usually had less RUVBL1/2–chromatin binding at their promoters (Fig. 6f, Supplementary Fig. 5i), indicating that the upregulated Cluster 1 and 2 genes may be less dependent on RUVBL2-mediated transcriptional

activation. In addition, the distribution of the target genes of transcription factors in each cluster known to interact with RUVBL2 and presented in Fig. 6e were found to be nonspecific (Fig. 6g). These results indicated that RUVBL2 depletion led to dramatic changes in gene expression that cannot be simply explained by chromatin binding of specific transcription factors.

**Fig. 6 | The RUVBL1/2 complex is required for the precise expression of genes, but does account for the specific binding of transcription factors. a** Cell growth analyses after RUVBL2 depletion (*n* = 3 biological replicates), error bar indicates the mean +/− SD. **b** Diagram showing the experimental design for the treatment of RUVBL2-degron cells. **c** Western blot analyses of RUVBL2 protein levels at different time points during RUVBL2 degradation. **d** Snapshots of poly(A) RNA-Seq signals on the c-Myc, Ccnd1, Chop, and Atf4 loci at different time points during RUVBL2 degradation. **e** Cluster analyses of RNA-Seq expression signals after RUVBL2 degradation at different time points. The enriched GO terms and the motifs were shown accordingly. **f** The results of the metagene analyses of RUVBL1 and RUVBL2 ChIP-Seq signals at the gene promoters of each cluster are shown in **e**. **g** Pie charts showing the distribution of target genes corresponding to specific

transcription factors peak belonging to the clusters in **e**. The "other" cluster indicates the targeted genes that did not belong to any other cluster. "Random" indicates the distribution of 8000 genes randomly selected from the genome. **h** The numbers of genes in each cluster shown in **e** that were simultaneously affected by UPF1 knockdown[80] are plotted in a bar graph. The percentages in the bracket were the ratio of the number of genes in each cluster as shown in **e**. **i** Cluster analyses of the percentage splicing index (PSI) after RUVBL2 degradation at different time points. The corresponding enriched GO terms are shown. **j** The PSI of Smg1 exon 2 is shown on bar graphs. Error bars indicate the mean +/− SDs. rMATS were used to calculate the significance difference of PSI (*n* = 2, *$p < 0.05$, **$p < 0.01$, ***$p < 0.001$). The position of the exon and the splicing event are illustrated in the upper panel. The FDR are extracted from rMATS by adjusting p-values with BH method.

Previous studies reported that the RUVBL1/2 complex is required for nonsense-mediated decay (NMD) complex formation[79]. We sought to determine whether the gene dysregulation evident after RUVBL2 depletion might be due to disruptions at the posttranscriptional level. The critical NMD complex component SMG1 was preferentially detected in RUVBL2 ChIP-MS preparations (Supplementary Data 1). We then reanalyzed previously published UPF1-knockdown RNA-Seq data (UPF1 is a key component in the NMD complex)[80] and found that many of the genes affected by the NMD pathway largely overlapped with the genes affected by RUVBL2 depletion (Fig. 6h, Supplementary Fig. 5j). Additionally, we found that the expression of the key NMD complex components Upf3a and Upf3b was decreased at late time points after RUVBL2 depletion (Supplementary Data 3), suggesting that RUVBL2 regulated the NMD complex at multiple levels. The RUVBL1/2 complex has been shown to be required for spliceosome formation[81–83]. Therefore, we next investigated the roles played by RUVBL2 in alternative splicing (Fig. 6i). GO analysis showed that genes with altered splicing were enriched in DNA and RNA metabolism (Fig. 6i), and the expression of most of these genes changed after RUVBL2 degradation (Supplementary Fig. 5k). Interestingly, SMG1 splicing (Fig. 6j) was significantly decreased 12 h after RUVBL2 depletion. RUVBL2 also interacted with key components of the pre-snoRNP complex and the mRNA processing complex (Supplementary Fig. 1c, Data 1), which explains RUVBL2 regulation of RNA fate at the posttranscriptional level. Our results collectively suggest that RUVBL2 may facilitate both transcriptional and posttranscriptional regulation of gene expression.

## Transcriptional defects after acute depletion of RUVBL2 could not be directly explained by INO80 and TIP60 chromatin remodeling

We downloaded publicly available INO80, TIP60, and SRCAP (ZNHIT1) ChIP-Seq data obtained from mESCs and compared them with our RUVBL1/2 ChIP-Seq data. We found that RUVBL1/2 colocalized with INO80, TIP60, and SRCAP complexes both genome-wide and at specific genomic loci (Supplementary Fig. 6a, b). We then performed P400, INO80 ChIP-Seq, and MNase-Seq 0.5 h and 1 h after RUVBL2 depletion and found that P400 chromatin binding first decreased at 0.5 h and then increased at 1 h, but INO80 binding mildly decreased at both timepoints, and the +1 nucleosome positioning (which is critical for transcriptional activation) was decreased 0.5 h and increased 1 h after RUVBL2 depletion (Supplementary Fig. 6c, d). However, Pol II transcription was dramatically reduced both 0.5 h and 1 h after RUVBL2 depletion (Fig. 2d, e). We further compared our RUVBL2-depletion gene expression data with publicly available INO80-, TIP60-, and P400-knockdown gene expression data obtained with mESCs[84,85] and found that only 1.7–16.2% of the genes (both up- and down-regulated genes) impacted by RUVBL2 depletion overlapped with the genes affected by INO80, TIP60 or P400 knockdown individually and by 15.4–20.3% with the pool of all genes affected by INO80, TIP60, or P400 knockdown (Supplementary Fig. 6e). As a positive control, 30–77% of TIP60- and P400-affected genes overlapped. The depletion of RUVBL2 would lead to both P400 and INO80 remodelers working

unproperly, which may create synergistic effects, but our results were consistent with a defect of RUVBL2 on transcription mostly independent from the RUVBL2 chromatin remodelers.

## Identification of direct targets of the RUVBL2-Pol II axis

To gain further understanding of the functional relevance of the roles played by RUVBL2 in Pol II regulation, we identified direct targets of the RUVBL2-Pol II axis in mESCs. Previous RNA-Seq or microarray studies with knockout cells revealed RUVBL2-regulated genes. However, the genes directly targeted by RUVBL2 are still unknown, as RUVBL2 interacts with multimolecular complexes with vastly different functions. We analyzed a nascent RNA-Seq dataset 0.5 h and 1 h after RUVBL2 depletion, and identified 45 direct targets of the RUVBL2-Pol II axis in mESCs (through identification of genes with RUVBL2 binding and immediate expression responses. For details, see the Methods section and Fig. 7a). Functional annotations showed that many genes of these genes encoded transcription factors or proteins involved in transcription regulation (Fig. 7b, c). For example, the proto-oncogene *c-Myc* and the *TGF-beta* signaling factor *Bmp4* were direct targets of RUVBL2, consistent with previously identified roles played by RUVBL2 in carcinogenesis. In addition, many target genes were involved in RNA metabolism, signaling, differentiation, and mitochondrial function (Fig. 7b), which was consistent with the previous findings showing that the functions of RUVBL2 in cellular activities are extensive.

To further explore the molecular determinants of genes directly targeted by RUVBL2, we took advantage of the extensive and publicly available mESC ChIP-Seq datasets. Forty-five nontargeted genes were selected on the basis of their unchanged PRO-Seq signals and an expression level similar to that of the target genes. Then, 153 ChIP-Seq datasets were obtained from the Cistrome database (Fig. 7d). The ChIP-Seq signals within 2 kb of TSSs in the target and nontarget genes were extracted, and a subset of the data was used to train three different classification models, which were then evaluated with the reserved test genes. These models predicted molecular determinants with high fidelity (Supplementary Fig. 7a). Analysis of the relative occupancy of the 10 factors with the largest and smallest coefficients in the generalized linear model (GLM) showed that DPPA2, JUN, ELL3, ZMYND8, and KDM2B were preferentially enriched at target genes compared with nontarget genes; however, no single factor characterized all 45 target genes (Fig. 7e), and the chromatin binding of the RUVBL2 direct targets C-MYC and CXXC1 explained only subsets of the genes dysregulated after RUVBL2 depletion (Fig. 7f). These results further suggest that RUVBL2 generally interacts with diverse transcription factors.

As RUVBL2 plays key roles in posttranscriptional RNA processing[79], we sought to identify the immediate-early response genes targeted by RUVBL2 at the posttranscriptional level. We identified these genes by following the criteria described in the Methods section (Supplementary Fig. 7b). Multiple environmental stimulus-related genes, such as Atf4, Atp1a3, and Ddit4, were identified as immediate-early response genes induced by RUVBL2 depletion at the posttranscriptional level (Supplementary Fig. 7c, d), suggesting that RUVBL2 exhibits functions in addition to its role in transcriptional

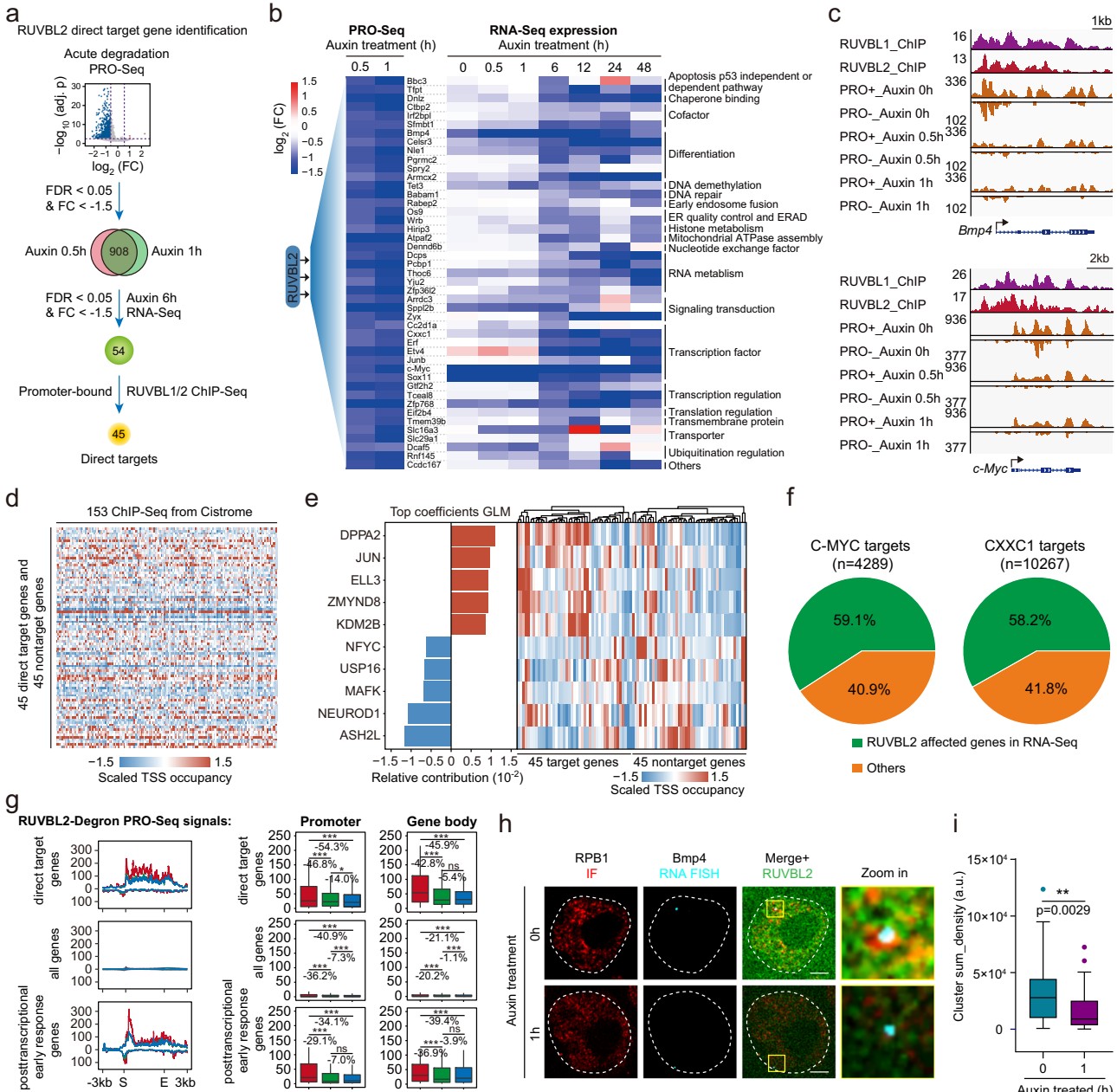

**Fig. 7 | Identification of direct target genes of the RUVBL2-Pol II axis.**
**a** Workflow of the process used to identify the genes directly targeted by RUVBL2. The FDR are extracted from DESeq2 by adjusting p-values using BH method. **b** Heatmap analyses of RUVBL2 direct target genes based on PRO-Seq expression. **c** Snapshots showing the RUVBL1 and RUVBL2 ChIP-Seq signals and nascent RNA (PRO-Seq) signals at the *Bmp4* and *c-Myc* loci after RUVBL2 degradation. **d** ChIP-Seq signals in the promoter regions of 45 genes directly targeted by RUVBL2 and 45 nontargeted genes in 153 datasets obtained from the Cistrome DB (Supplementary Data 5). **e** The 10 factors with the largest and smallest coefficients in the elastic net were selected. The heatmap shows the relative occupancy of these 10 factors in the 45 genes directly targeted by RUVBL2 and in 45 nontarget genes. **f** Pie chart indicating the percentage of C-MYC or CXXC1 ChIP-Seq-determined target genes that overlap the RUVBL2-affected genes as determined by the RNA-Seq data shown in Fig. 6g. **g** Profiles of PRO-Seq signals at the gene regions of directly

targeted genes ($n = 45$), all genes ($n = 36821$) and posttranscriptional early response genes ($n = 41$) after RUVBL2 degradation. The profile above 0 indicates the signal on the sense strand, the profile below 0 indicates the signal on the antisense strand. Two-tailed Wilcoxon tests was performed to calculate the significance ($*p < 0.05$, $**p < 0.01$, $***p < 0.001$). The center line of boxplot represents median, the box limits represent upper and lower quartiles, and the whiskers represents 1.5x interquartile range or maximum/minimum value. **h** The RUVBL2 direct target gene *Bmp4* was examined by single-molecule RNA FISH combined with RPB1 immunofluorescence. Representative images are shown, scale bar is 3 μm. **i** The signals of the Pol II (RPB1 IF) cluster that overlap with nascent RNA foci were measured, the boxplot indicated median (middle line), 25th and 75th percentile (box) and 5th and 95th percentile (whiskers), as well as outliers (single black points). Statistically analyzed (two-tailed Student's *t* test) from at least 10 different fields of each condition (Auxin 0 h $n = 61$, Auxin 1 h $n = 38$). $**p < 0.01$.

regulation. To examine the molecular effects of these direct target genes on Pol II after RUVBL2 depletion, we analyzed the PRO-Seq datasets after RUVBL2 degradation at different time points. The results showed that the PRO-Seq signals at direct target genes, all genes and posttranscriptional early response genes were decreased both 0.5 h

and 1 h after RUVBL2 depletion (Fig. 7g). To provide direct evidence showing RUVBL2 mediates Pol II clustering on chromatin of RUVBL2 target genes, we performed Pol II IF and RNA fluorescence in situ hybridization (FISH) after immediate depletion of RUVBL2. *Bmp4* is a direct target of RUVBL2, and its nascent RNA synthesis was

dramatically decreased after RUVBL2 depletion, as measured by PRO-Seq (Fig. 7c). We then performed Pol II IF and RNA FISH with probes against nascent *Bmp4* transcripts within 1 h of auxin treatment. The results showed that RUVBL2 depletion led to significantly inhibition of Pol II clustering and a decrease in nascent RNAs at the *Bmp4* locus (Fig. 7h, i). Together, these results demonstrate that RUVBL2 directly regulates Pol II clusters at a specific genomic locus.

### RUVBL2 occupies promoters with diverse transcription factors and is required for the expression of gene-encoded hallmarks of cancer

To provide further mechanistic insights, we explored the transcription factors that are potentially critical for RUVBL2-mediated Pol II transcription, and a binding motif analysis of RUVBL1/2 revealed many transcription factors known to cooperate with these proteins (Fig. 8a)[40]. In contrast to a previously reported model in which RUVBL1/2 was found to be associated with specific factors[86–93], our clustering analyses of RUVBL1/2 ChIP-Seq signals with previously identified oncogenic transcription factors (such as C-MYC, E2F1, and P53) revealed that very few RUVBL1/2-binding sites showed relative specificity for these transcription factors (Fig. 8b). For example, the target genes *Ccne2* and *Chac1* were relatively enriched with C-MYC, and *Atf3* showed relative enrichment with P53, but many other key genes did not show specific enrichment of oncogenic transcription factors at their promoters (Supplementary Fig. 8a, Data 4). The distribution of the ChIP-Seq peaks attributed to C-MYC, E2F1, CTCF, YY1, and KLF5 from the K1 to K6 categories shown in Fig. 8b was calculated. The results showed that many peaks of these transcription factors colocalized with RUVBL1/2 in a nonspecific manner (Fig. 8c). These results indicate that RUVBL1/2 generally occupy active promoters and that many transcription factors preferentially bind gene promoters to function cooperatively. Thus, in the literature, many transcription factors have been shown to interact with RUVBL1/2[40].

We next sought to investigate the physiological relevance of the RUVBL2-Pol II axis in cancer cells. RUVBL1/2 are constitutively expressed in mammalian cells and have been reported to be overexpressed in liver, colorectal, and many other cancers[40,94,95]. Examining the gene expression data in the Genotype-Tissue Expression (GTEx) database, we found that RUVBL2 was generally more highly expressed than RUVBL1 across human tissues. Interestingly, RUVBL2 was highly expressed in the testis (Supplementary Fig. 8b). The examination of the gene expression levels of RUVBL1/2 in different cancer types, as reported in the Gene Expression Profiling Interactive Analysis (GEPIA) database showed that RUVBL1/2 were generally overexpressed in cancers and seemed to be preferentially overexpressed in B-cell lymphoma and thymoma (Supplementary Fig. 8c). We performed western blot analyses with different cancer cell lines and found that RUVBL1/2 were highly expressed in MCF-7 and K562 cells and were expressed at relatively lower levels in GM12878, U2OS, and HepG2 cells (Supplementary Fig. 8d). RUVBL2 ChIP-Seq was then performed with different human cell lines. K-means clustering analyses of the ChIP-Seq peaks indicated that RUVBL2 mostly occupied promoters of genes (Supplementary Fig. 8e). Clustering analyses were performed with transcription factor ChIP-Seq data obtained from MCF-7 cells because considerable ChIP-Seq data obtained from this cell line have been published in the Encyclopedia of DNA Elements (ENCODE) and Gene Expression Omnibus (GEO) databases. Similarly, the results showed nearly no specificity of RUVBL2 binding with oncogenic transcription factors in this cell line. These results indicated the prevalent co-occupancy of RUVBL2 with different transcription factors in mammalian cells (Fig. 8d, e). Metagene analyses indicate that RUVBL2 was preferentially enriched in direct target genes compared with all genes and posttranscriptional immediate-early response genes in multiple human cancer cell lines (Fig. 8e). Moreover, many cancer hallmark genes were directly targeted by RUVBL2 or were immediate-early

response genes bound by RUVBL2 in cancer cells (Fig. 8f, g), suggesting conserved RUVBL2 function in mammalian cells.

In our study, we found that RUVBL2 activated the expression of the *c-Myc* and *Ccnd1* oncogenes; these genes are important in carcinogenesis[40,96]. Because RUVBL2 was relatively more abundant in MCF-7 cells, according to our western blot analyses (Supplementary Fig. 8d), we performed survival analysis of patients using the Kaplan–Meier plotter tool[97–99] with data obtained from patients with breast cancer, liver cancer, and cervical squamous cell carcinoma and high or low expression of RUVBL1 or RUVBL2. The results showed that overexpression of RUVBL2 was associated with survival in breast cancer patients and that overexpression of RUVBL1 was associated with survival in liver cancer and cervical squamous cell carcinoma patients (Fig. 8h, Supplementary Fig. 8f). A recent study showed that cordycepin inhibited RUVBL2 function in the circadian system of mammals[100]. Therefore, the expression of key genes regulated by RUVBL2 in MCF-7 cells was investigated after cordycepin treatment. The results showed that cordycepin indeed perturbed the expression of key genes (*C-MYC* and *SLC16A3*) in MCF-7 cells (Supplementary Fig. 8g). RUVBL2 has also been shown to regulate *C-MYC* in myeloid leukemia[101], suggesting that RUVBL2 may play a key role in carcinogenesis in multiple cancers.

## Discussion

Here, we reported that RUVBL2 promotes Pol II clustering and nascent RNA synthesis (Fig. 8i). We performed chromatin immunoprecipitation coupled with DNA sequencing and mass spectrometry to identify RUVBL2 DNA binding targets and interacting protein partners in mESCs and found that RUVBL1/2 generally co-occupied gene promoters with Pol II and diverse transcription factors. Taking advantage of rapid protein degradation technology, we carried out global time-series analyses of mature mRNA and nascent RNA levels and performed protein localization imaging and biochemical analyses. We found that RUVBL2 activates transcription and promotes Pol II clustering at active promoters.

The high-quality ChIP-Seq, mass spectrometry, and time series perturbation datasets generated in this study allowed us to develop a transcription regulatory cascade model for RUVBL2-regulated gene expression (Fig. 8i). RUVBL2 depletion inhibits transcription, while defects in RNA metabolism (such as disruption to the NMD pathway) increase RNA levels by increasing RNA stability. Therefore, the final readout of global mature mRNA expression is relatively unchanged, as transcriptional inhibition is balanced out by enhanced RNA stability. Therefore, both upregulation and downregulation of static gene expression was observed after RUVBL2 degradation.

RUVBL2 regulates many protein complexes, such as the snoRNP, telomerase, spliceosome, NMD, Pol II, PIKK family, INO80/SWR remodeler, oncogenic transcription factor, and signaling factor complexes, and RUVBL2 is usually considered a promoter of the assembly of these complexes[40,65,77,102,103]. On the other hand, RUVBL2 has been shown to function as a protein chaperone in the disassembly of protein aggregates[35,36,104]. We showed that RUVBL1/2 directly interacted with the RPB1 CTD. It is possible that multiple RPB1-binding sites within the complex promote clustering via multivalent interactions with the RPB1 CTD to mediate Pol II clustering. The RUVBL1/2 proteins cluster at the periphery, which is consistent with a recent finding showing that condensate dynamics are regulated by protein clusters absorbed on the condensate interface, which follows the Pickering emulsion theory[105]. On the other hand, we did not observe changes in the Pol II level in the nucleus after marked degradation of RUVBL2, suggesting that effects on Pol II assembly by the RUVBL2-containing R2TP complex likely did not contribute to the immediate defects on Pol II clustering in the nucleus. We also showed that chromatin binding, nucleosome positioning and regulated gene expression by the chromatin remodeling complexes INO80 and TIP60 did not directly explain

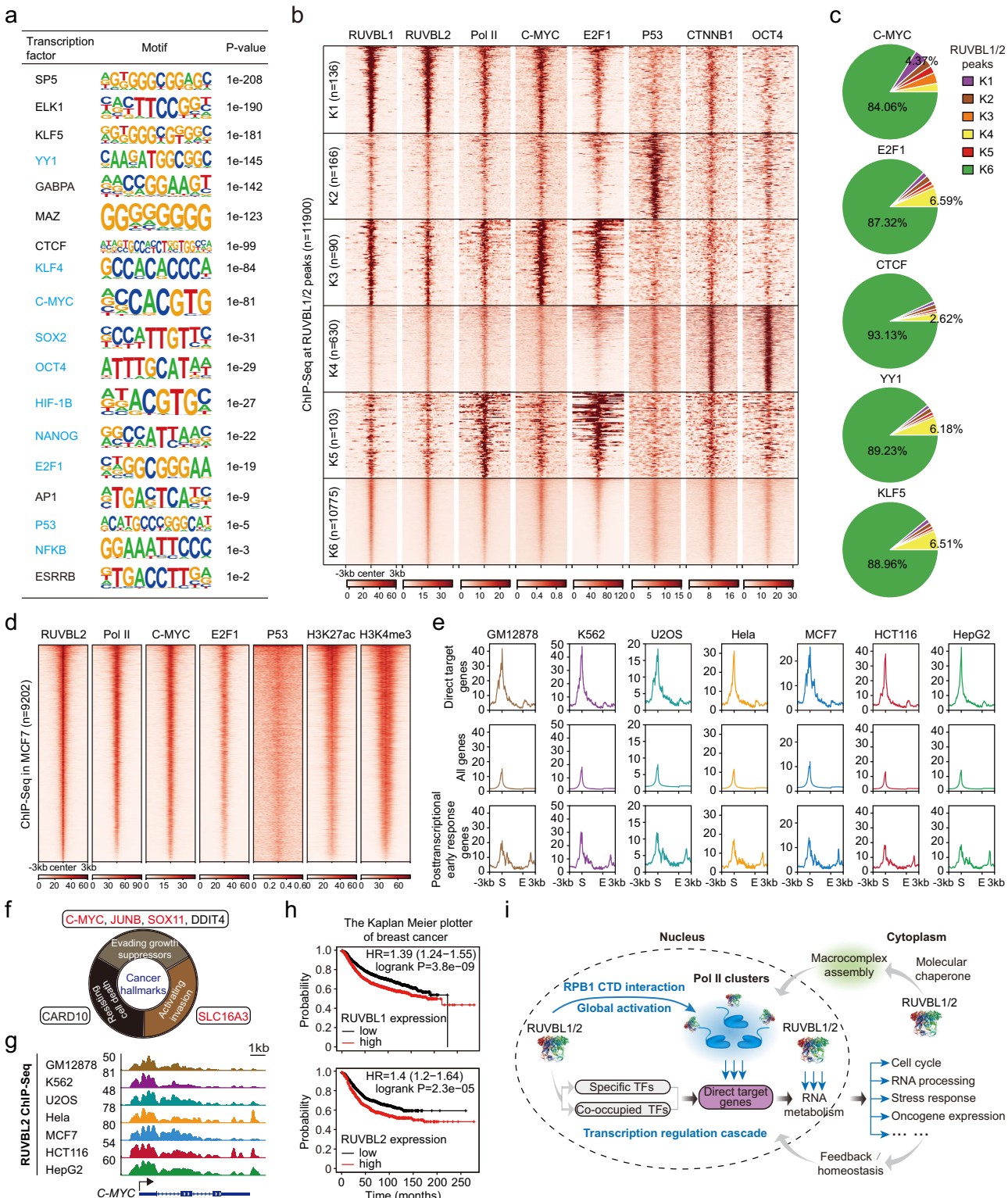

the profound inhibition in transcription initiation after rapid depletion of RUVBL2. These results collectively demonstrate that RUVBL2 is a bona fide protein regulator of Pol II cluster formation in the nucleus, and the protein chaperon-like potential of RUVBL2 in Pol II clustering is warranted for further investigation in the future.

In this study, extensive in vitro biochemical assays provided direct evidence for RUVBL2-mediated RPB1 CTD clustering and transcriptional activation. Our time series-based protein degradation analyses allowed us to separate the nuclear function of RUVBL2 from the R2TP-

mediated Pol II assembly function of RUVBL2 in the cytoplasm. Moreover, using ChIP-Seq, we showed that RUVBL2 occupied gene promoters along with Pol II and that tethering RUVBL2 to LacO loci did not lead to further recruitment of Pol II to silenced genes but enhanced Pol II clustering in the presence of transactivation domains of the artificial transcription factor rtTA, implying that RUVBL2 might promote weak protein interactions among low-complexity domains at gene promoters. RUVBL2 is often overexpressed in cancers[40,94,95]. According to our findings, RUVBL2 plays a critical role in global

**Fig. 8 | RUVBL2 occupies gene promoters with Pol II and diverse transcription factors in cancer cells. a** The list of transcription factor motifs that were enriched at the RUVBL1- and RUVBL2-overlapping peaks and the corresponding *p*-values (which are calculated using HOMER with cumulative binomial distributions) is shown on the right. The transcription factors in blue were previously reported to function with RUVBL1 or RUVBL2. **b** Cluster analysis with RUVBL1, RUVBL2, Pol II, C-MYC, E2F1, P53, CTNNB1, and OCT4 in RUVBL1/2-overlapping peaks. **c** Distribution pie charts showing the percentage of the corresponding transcription factor peaks in each category (K1 to K6) shown in Fig. **b**. **d** The heatmap illustrating RUVBL2 globally colocalization with various transcription factors in MCF-7 cells. **e** Metagene analyses of RUVBL2 ChIP-Seq data for the regions of genes directly targeted by RUVBL2, all genes and posttranscriptional early response genes. **f** The cancer hallmark genes directly targeted by RUVBL2 (red) and the posttranscriptional early response genes after RUVBL2 depletion (black). **g** Snapshots showing RUVBL2 ChIP-Seq signals at *C-MYC* loci in GM12878, K562, U2OS, HeLa, MCF-7, HCT116 and HepG2 cells. **h** The survival probability changes over time for patients with breast cancer and high (red line) or low (black line) expression of RUVBL1 (upper panel) and RUVBL2 (bottom panel) as determined by the curve constructed with the Kaplan–Meier plotter. Significance was calculated by log-rank method. HR indicates the hazard ratio, and the confidence interval is shown in brackets. **i** Model of RUVBL2-activated transcription of genes with diverse cellular functions and Pol II clustering. RUVBL1/2 were shown to function as global transcription coactivators with various transcription factors and to directly promote Pol II cluster formation mediated by the RPB1 CTD.

transcriptional activation and Pol II clustering, suggesting a potential transcriptional amplification mechanism in cancer cells. RUVBL1/2 may also be involved in the functions of other RNA polymerases (Pol I and Pol III). In addition to its involvement in gene expression regulation, RUVBL2 has been implicated in the cell cycle, cell invasion, DNA damage, and repair[40,43,103,106,107]. RUVBL2-mediated Pol II cluster formation may constitute a general mechanism through which RUVBL2 regulates other functional protein clusters on chromatin. Further studies on the functions of RUVBL2 in multimolecular complexes are expected to advance our understanding of the mechanisms underlying cancers. Designing drugs to target RUVBL2 may provide tools or methods for cancer studies or treatments.

## Methods
### Cell culture
The V6.5 mouse embryonic stem cell line was a gift from Richard Young's lab at the Whitehead Institute. mESCs were grown in 2i/mLIF medium[108], which was composed of ES-qualified Dulbecco's modified Eagle's medium (DMEM) (Millipore, SLM-220-M) containing 1 mM L-glutamine, 1× nucleosides (Sigma, ES-008), 1× nonessential amino acids (Milipore, TMS-001-C), 1× penicillin/ streptomycin (Gibco, 15140-122), 0.1 mM β-mercaptoethanol (Sigma, M3148), 3 µM CHIR99021 (Selleck, S1036) and 1 µM PD0325901 (Selleck, S1263), 1000 U/ml mouse leukemia inhibiting factor (mLIF) (Milipore, ESG1107), and also supplemented with 15% fetal bovine serum (FBS) (Gibco, 10099-141), incubated at 37 °C with 5% $CO_2$ in a humidified incubator. For DNA/ RNA- sequencing and mass spectrometry (MS), the mAID tagged mES cell lines were cultured in 0.2% gelatin-coated plates with irradiated mouse embryonic fibroblasts (MEF) in DMEM-KO (Gibco, 10829018) containing 15% FBS, 1000 U/ml mLIF, 1 mM L-glutamine, 0.1 mM β-mercaptoethanol, 1× nonessential amino acids, and 1× Penicillin/ streptomycin at 37 °C with 5% $CO_2$[14]. For human cells and cancer cells RUVBL2 ChIP-Seq, GM12878 (human lymphoblastoid cells, Female, from laboratory of Bing Ren, University of California, San Diego) cells were cultured in RPMI 1640 media (Gibco, 11879020) with 2 mM L-glutamine supplemented 15% FBS, 1 mM sodium pyruvate (Gibco, 11360070) and 1× penicillin/ streptomycin; K562 cells (Human chronic myeloid leukemia cells, gift from laboratory of Jiazhi Hu, Peking University) were cultured in RPMI 1640 media with 2 mM L-glutamine supplemented 15% FBS, 1 mM sodium pyruvate (Gibco,11360070), 1x nonessential amino acids (Milipore, TMS-001-C), 0.1 mM β-mercaptoethanol (Sigma, M3148), and 1× penicillin/ streptomycin; MCF-7 (human breast cancer cells, from laboratory of Hong Wu, Peking University) were cultured in Minimum Essential Medium (MEM, Gibco, A4192201) containing 15% FBS and 1× penicillin/ streptomycin; HCT116 cells (Human colon carcinoma cells, from laboratory of Fuchou Tang, Peking University) were cultured in DMEM (Gibco, C11995500BT) supplemented 10% FBS and 1× penicillin/ streptomycin; U2OS cells (human osteosarcoma cells, ATCC, HTB-96) were cultured in DMEM supplemented 10% FBS and 1× penicillin/ streptomycin; Hela cells (human cervical carcinoma cells, from Laboratory of Jingyan Fu, China Agriculture University) were cultured in DMEM supplemented 10% FBS and 1× penicillin/ streptomycin; HepG2 cells (human hepatoblastoma cell, from Laboratory of Yuanchao Xue, Institute of Biophysics, Chinese Academy of Sciences) were maintained in DMEM supplemented 10% FBS with 1× penicillin/ streptomycin. All the cell lines were regularly tested negatively for mycoplasma contamination and were used for further experiments.

### Cell lines generation
RUVBL2 and RUVBL1 degron mESC cell lines were generated[109]. Briefly, the equal amounts of gRNAs targeting to RUVBL2 or RUVBL1 stop codon region and mAID-GFP donor fragments with geneticin resistance were encapsulated using FuGENE HD transfection reagent (Promega, E2311) and co-transfected into the parental cell lines Tet-ON-hPGK-OsTIR1 (V6.5) mESC in 6 well plates. After 24 h, cells were treated with trypsin and seeded into 10 cm dishes, then 500 µg/ml geneticin was added simultaneously to culture 5–7 days to select the drug-resistant clones, fresh medium with selection chemicals was changed every day. Finally, the single clones were picked up under the microscope and disseminated seeding into 48 well plates; To increase the success rate of knock-in, we also carried out the procedure below at the same time. The drug-resistant clones were collected for flow cytometry sorting, and green fluorescence positive cells were collected in 96 well plates for further identification. One week later, the survived clones were checked by fluorescence imaging and genotyping. The pair primers from the distal ends of homologous arms were used for genotyping, and only the homozygotes were expanded for further western blot validation. Finally, the positive clones with both right molecular weight and localization were stored and used for downstream analyses. RPB1 was endogenously tagged by mCherry at the C-terminal by following the procedure described above with only the mAID-GFP tag replaced by mCherry and adopted hygromycin resistance within the RUVBL2 degron mESCs, and similarly, RPB1 was also endogenously tagged by GFP at the C-terminal within wild type mESCs. mEos3.2 was homogenously tagged into the N-terminus of RPB1 in RUVBL2 degron mESCs for tcPALM imaging. The primers were available in Supplementary Data 6.

### Cell treatments
For degron cell lines, 2 µg/ml Doxycycline (Dox, Sigma Aldrich, D9891) was added into the medium for 24 h to induce the OsTIR1 expression, and then 500 µM auxin (3-indole acetic acid, Sigma Aldrich, 1I5148) were appended for the indicated time to induce the target protein degradation. The mES cell lines were harvested for the follow-up experiments off two passages of MEF feeders.

For Cordycepin (Selleck, S3610) treatment, MCF-7 cells were cultured in 24 well plates to confluent about 80%, 25 µM cordycepin (DMSO dissolved) was added[110,111]. After 1 h, 6 h, 12 h, and 48 h, and 72 h, cells were directly lysed by Trizol and harvested for RNA extraction according to the instruction of the Trizol reagent (Ambion, 115596018). Then 1 µg total RNA was reverse transcribed using the HiScript® III 1st Strand cDNA Synthesis Kit (+gDNA wiper) (Vazyme, R312-02), the

expression of RUVBL2 target genes was examined through RT-qPCR, and the primers were available in Supplementary Data 6.

For transcription and importin inhibition, the RUVBL2 degron/RPB1-mCherry mESCs were seeded and cultured on the coverslips. For the Pol II transcription inhibition, 5 µM Actinomycin D (Selleck, S8964) was added to the medium and incubated for 3 h to 6 h. For the importin inhibition, 40 µM importazole (Selleck, S8446) was used to treat the cells for 1 h, 3 h, and 6 h, respectively. Then, the inhibited cells were fixed for 10 min in 4% paraformaldehyde at room temperature, then penetrated using 0.25% Triton X-100/PBS for 10 min, and nuclei were stained with DAPI in PBS (2 µg/ml) for 5 min at room temperature. After being washed 3 times with PBS, coverslips were mounted on slides using Vectashield and sealed with nail polish. Images were acquired by Nikon A1R microscope with 100x oil immersion lens. All the fluorescence images were post-processed and exported using volocity software (PerkinElmer) under the same optical parameters and further performed the cluster analysis.

## Chromatin fraction isolation and western blot analysis

To examine the interactions between RUVBL1/2 and RPB1 in nucleus, the chromatin fractions were isolated in this study, according to the published protocol[112]. Briefly, 15 cm plate 90% confluent cells were gently trypsinized (wild-type mESCs, RPB1-GFP mESCs, RUVBL1/RUVBL2-Degron mESCs, and RPB1-Degron mESCs with/ without auxin treatment) and removed from the plate by pipetting and collected by centrifugation (100 g, 4 °C for 5 min). Cell pellets were rinsed twice with PBS/1 mM EDTA, and then the plasma membranes were lysed by resuspension in 1 ml ice-cold NP-40 lysis buffer I (10 mM Tris-HCl [pH7.5], 0.15% NP40, 150 mM NaCl) for 5 min on ice. The lysate was then layered on top of 2.5 volumes of a chilled sucrose cushion (24% sucrose in NP-40 lysis buffer I) and centrifugated for 10 min, 4 °C, 13,000 g. The supernatant (cytoplasmic fraction) was collected. The nuclei were washed twice with PBS/1 mM EDTA, then lysed for 5 min in 1 ml of ice-cold lysis buffer II (20 mM HEPES-KOH [pH 7.5], 150 mM NaCl, 1.5 mM MgCl$_2$, 0.5 mM DTT, 1.25×protease inhibitors, and 0.6% Triton X-100) on ice. The nucleoplasm fraction was collected by centrifugation at 13,000 g for 30 min at 4 °C. To avoid cytoplasm components contamination, the pellets were washed once more with ice-cold lysis buffer II and three times with PBS/1 mM EDTA. Finally, 1 ml lysis buffer II with the nucleases (0.5 µl Benzonase (Sigma Aldrich, E8263), 10 µl DNase I (NEB, M0303S), 2 µl RNase A (Sigma Aldrich, R6148), 2 µl Micrococcal Nuclease (MNase, NEB, M0247) and 5 µl 1 M CaCl$_2$) were added into the chromatin pellet, and incubated at 25 °C on a rotator with 1,000 rpm for 1-2 h until the chromatin pellet was resuspended thoroughly, and then centrifuged for 30 min at 13,000 g at 4 °C. The supernatant was saved as the chromatin fractions, which were used in this study. 50 µl cleared chromatin fractions were sampled for input, and the rest were used for immunoprecipitation of different targets. To investigate the interactions between RUVBL2 and RPB1, GFP-Trap beads (Chromotek, gtma-20) were used for immuno-precipitation of the chromatin fractions following the manufacturer's instruction with the gradient NaCl concentration (0.15, 0.3, 0.5, 1.0, 1.5 and 1.8 M) in the wash buffer. For RPB1-CTD degradation experiment, 2 µl RPB1-NTD antibodies (CST, 14958) were used for immuno-precipitation of the chromatin fractions. The IPed samples were analyzed with RPB1-CTD (Abcam, ab26721, dilution 1:3000) antibodies (to confirm the Pol II degradation), RPB1-NTD antibodies (dilution 1:3000) and RUVBL1 (Santa cruz, sc393905; dilution 1:2000) or RUVBL2 antibodies (Santa cruz, sc374135, dilution 1:2000), and RPAP3 (ABclonal, A15239; dilution 1:2000). We usually used native cells for chromatin isolation. Only the 1% formaldehyde crosslinked wildtype mESCs were used for chromatin extraction for the molecular sieve isolation to avoid protein degradation during protein sample preparations. The chromatin extracts were loaded on the AKTA system, and the superdex 200 (GE) column was engaged to separate complexes with different

molecular weights. For western blot, samples were boiled for 10 min and loaded on 10% (w/v) SDS-PAGE. The proteins were transferred onto polyvinylidene fluoride (PVDF) membranes together with a visible pre-stained protein marker through 300 mA for 2 h in ice bath. Membranes were blocked with 5% (w/v) skim milk/ TBST (1‰ Tween-20) for 1 h at room temperature and then incubated with the primary antibodies (RPB6: Proteintech, 15334-1-AP, dilution 1:1000; RPB7: Santa cruz, sc398213, dilution 1:1000; INO80: Proteintech, 18810-1-AP, dilution 1:2000; P400: Bethyl, A300-541A, dilution 1:1000) overnight at 4 °C. After 3 times washing with TBST, primary antibodies bound membranes were incubated with corresponding secondary antibody (Goat anti-Mouse IgG (H + L) Secondary Antibody, HRP (ThermoFisher, 31430) and Goat anti-Rat IgG (H + L) Secondary Antibody, HRP (Thermo Fisher, 31470), diluted 1: 10000 in TBST with 1% BSA following the manufacturer's instruction) for 1.5 h at room temperature. Then washed 5 times in TBST, and the reactive bands were visualized using enhanced chemiluminescence reagents (BioDragon, BF06053-50) and captured on G.E AI 600 RGB imaging system. Pictures were further analyzed using Fiji Is Just ImageJ (Fiji).

## Immunofluorescence assay

Immunofluorescence (IF) assays for target proteins localization were performed[28]. Briefly, pretreated cells and then seeded on glass coverslips or glass-bottom dishes to grow for 3–5 h, followed by fixation for 10 min in 4% paraformaldehyde (PFA)/PBS at RT. After washing 3 times with PBS, cells were permeabilized with 0.25% Triton X-100/PBS for 10 min at room temperature, followed 3 times washing using PBS, and then blocked with 3% bovine serum albumin (BSA) /PBS for 60 min prior to incubation with the primary antibodies. Cells then were incubated with antibodies against RPB1-CTD (1:500 dilution, Abcam, ab817) and CTCF (1:500 dilution, ABclonal, A1133), RUVBL1 (1:200 dilution, Santa cruz, sc393905), RUVBL2 (1:200 dilution, Santa cruz, sc374135) overnight at 4 °C in humidifying chamber, followed by PBS washing 3 times and then stained with corresponding secondary antibodies (Alexa Fluor 568 Donkey anti-Mouse IgG (1:1000 dilution, Thermo Fisher, A-10037); Alexa Fluor 594 Goat anti-Rabbit IgG (1:1000 dilution, Thermo Fisher, A-11012); Alexa Fluor 488 Goat anti-Mouse IgG (1:1000 dilution, Thermo Fisher, A-11001); Alexa Fluor 488 Goat anti-Rabbit IgG (1:1000 dilution, Thermo Fisher, A-11008)) for 1 h at room temperature protected from the light, the nuclei were counterstained with DAPI (2 µg/ml) for 5 min, after 3 times washing with PBS, coverslips were subsequently mounted onto microscope slides. Confocal images were taken on Nikon A1R microscope with epifluorescence optics using an oil immersion lens with 100× magnification. All the fluorescence images were post-processed and exported using volocity software (PerkinElmer) under the same optical parameters for later comparative analysis.

## LacR tether imaging and data analysis

U2OS 2-6-3 cells were seed on the glass coverslips in 12 well plates. To examine the RUVBL1/2 and RPB1 recruitment during the transcription activation, the Lac repressor fused Blue Fluorescence Protein (BFP) plasmids (LacR-BFP), and TetON elements plasmids were co-transfected into U2OS 2-6-3 cells, 24 h later, 2 µg/ml Dox was added to induce the transcription activation for another 24 h, then, cells were fixed for immunofluorescence using the antibodies that recognize endogenous RUVBL1/2 or RPB1 according to the protocol as described above. For further investigation of the roles of RUVBL1/2 in transcription activation, the equal amounts of plasmids (LacR-BFP, LacR-BFP-RUVBL1 or LacR-BFP-RUVBL2, LacR-mRUVBL2 (ATPase mutant), and LacR-BFP-RUVBL2 DI, LacR-BFP-RUVBL2 DII, LacR-BFP-RUVBL2 DIII, LacR-BFP-RUVBL2 DI-DIII) were transfected into U2OS 2-6-3 cells, respectively, 24 h later, 2 µg/ml Dox was added to incubate for further 24 h, then, cells were fixed for immunofluorescence using the initiation-associated unphosphorylated RPB1 antibodies.

All pictures were captured using the Nikon A1R microscope at 100× oil lens, and the same parameters for respective channels (BFP: 90% laser 405 and HV 160, FITC: 20% laser 488 and HV 40, and RFP: 5% laser 560 and HV 20). For each condition, pictures from at least 10 different fields and no less than 15 cells were acquired for quantification. To exclude the interference by the variations of immunofluorescence and transfection efficiency of plasmids, fluorescence signals at the LacO (Lac operator) loci were normalized using the mean signals in the area without clusters in the same cell (background), defined as the relative enrichment of target signals. For normalization, first, the BFP concentrated ROI (region of interest) according to the signal of BFP was defined through volocity software (PerkinElmer), then the mean and total fluorescence intensity of blue, green and red channels in the ROI region were measured. Three zones with the same size of BFP concentrated ROI at the BFP unconcentrated regions in the same cell were randomly selected as BFP unconcentrated ROI for quantification. The average values of the mean and total fluorescence intensities of each channel in the three BFP unconcentrated ROIs were used as the background intensity of the cell. Finally, the intensity in BFP concentrated ROI was divided by the background intensity to get the relative intensity. To further clarify the variations of tethered protein and transfection efficiency of the plasmids, the raw intensities of the blue channel within BFP concentrated ROI were quantified, and at least 30 cells of each transfection were used for Student's t test. Similarly, LacR-BFP-CTD and the control plasmid LacR-BFP were transfected into the U2OS 2-6-3 cells, respectively. 24 h later, the cells were fixed using 4% PFA, and following the RUVBL2 IF (Alexa Fluor 594), then the images were acquired by Nikon A1R microscope with 100× oil lens. To directly show the recruitment of RUVBL2 by the tethered RPB1-CTD, images were analyzed using "Clock Scan" in FiJi. Briefly, take the LacR-BFP or LacR-BFP-CTD (LocO locus) as the center and add to ROI, set the same ROI radius (20 pixels, 0.12 μm/pixel), then carry out the "Clock Scan combined" under plugin with the default parameters. The BFP and RUVBL2 IF (Alexa Fluor 594) density from no less than 20 different cells were used for statistics and plotted as Clock Scan distribution.

## Confocal imaging and cluster quantification

RUVBL2 degron mESCs labeling with RPB1-mCherry was grown on coverslip. After being treated with Dox and auxin, cells were fixed by 4% PFA/PBS incubating at room temperature for 10 min, washed 3 times with PBS, then permeabilized with 0.25% Triton X-100/PBS for 10 min at room temperature, followed 3 times washing with PBS. The nucleus was stained by DAPI (2 μg/ml) for 5 min at room temperature, after 3 times washing with PBS, coverslips were subsequently mounted onto microscope slides. Images for RPB1 and CTCF cluster changes during RUVBL2 degradation were acquired on Nikon A1R microscope with 100× oil-immersion objective and the according to pinhole (0.7 U), HV for RUVBL2-mAID-GFP (Green channel) set as "40", and "80" for both RPB1 and CTCF (Red channel); laser power was set both at 20%, and the same resolution (1024 × 1024) was set for different fields. Images were post-processed and exported using volocity software (PerkinElmer).

For RPB1 and CTCF cluster quantification, the same optical parameters confocal mages were deconvolved and post-processed using Fiji. Clusters were called for the fluorescence channels following two-step procedure. For each group, firstly, the background was subtracted by default parameters for each image. Then the minimal "thresholds" were set, and "watershed" was used to make sure the cluster's boundary could be recognized as individual foci. To calculate the number and size of the cluster, the cluster minimal size filter was set to "10- Infinity". The number of clusters per cell from at least 10 independent fields was counted for each time point and compared with each other using Student's t-test.

## Nuclear and cytoplasm Pol II (RPB1) quantification

For nuclear RPB1 quantification, the raw confocal images acquired under the same parameters were post-processed using Fiji. For RUVBL2 degron mESCs, DAPI stains were used for nuclear definition. Briefly, the RGB format of the DAPI channel was firstly transformed into "8-bit" format, set "Auto Threshold", and carried out "Fill Holes" & "Watershed" in binary, regions bigger than 2000 pixels were selected as ROI (indicating the nucleus), then the integrated densities of Pol II signals (RPB1-mcherry used in Fig. 2a and endogenous RPB1 used in Fig. 4e) of ROI were measured. Because the LacR was located in the nucleus, we used BFP signals for the definition of the nucleus in LacR-BFP or LacR-BFP-RUVBL2 transfected cells. To further examine whether Pol II protein level changed or not after RUVBL2 acute depletion, the cells without or with auxin treatment 0.5 h and 1 h were subjected to nucleus isolation adapted from the chromatin fraction isolation protocol. The isolated cytoplasm and nuclei were collected for western blot to examine the RUVBL2 degradation and RPB1 protein level. To quantify the nuclear concentration of RUVBL1/2, gradient dilution of recombinant RUVBL1/2 proteins and the nuclei ($3.5 \times 10^{5}$ cells) were resolved by SDS-PAGE and visualized through western blot. The recombinant and RUVBL1/2 densities in the nucleus were measured by Image J, the concentration/ density curve (linear equation) was fitted based on the gradient concentration of recombinant RUVBL1/2. Radius of mESC was an average 3 μm, so the average volume was about 113.04 $(4/3\pi R^{3})$ x $10^{-15}$L. Based on the molecular weight of RUVBL1 (50214 g/mol) and RUVBL2 (51113 g/mol), the concentration of RUVBL1 and RUVBL2 in the nucleus was calculated, respectively. To represent the hexamer of RUVBL1/2 complex, 3 molecules (RUVBL1 or RUVBL2) were calculated as one stoichiometry unit.

## Time-correlated of photoactivation localization microscopy (tcPALM)

The N-terminal of RPB1 was homogenously tagged with mEos3.2 in RUVBL2-degron mESCs with CRISPR genome editing. RUVBL2 degron mESC were treated with Dox and auxin. Live cell super-resolution tcPALM imaging was performed on a custom-built Nikon Eclipse Ti2 microscope with a 100×/NA 1.49 oil-immersion TIRF objective and motorized laser illumination to achieve highly inclined and laminated optical sheet illumination[113], similar as the method published previously[5,52,59]. Activation (405 nm for photoconversion of mEos3.2) and excitation (488 nm for pre-converted mEos3.2 and 561 nm for post-converted mEos3.2)[114] laser beams were modulated by an acousto-optic Tunable Filter. Images were acquired with an Andor iXon EM-CCD camera with an image pixel size of 160 nm. We acquired movies of 3000 frames with a frame rate of 50 ms. Each frame consisted of a 50 ms continuous excitation 561 nm laser, followed by a ~447 μs relatively low-power 405 nm photo-activation laser within camera transition time to achieve sparse labeling. Axial drift during acquisition was corrected by a perfect focusing system. The RUVBL2 degradation was checked manually through the GFP signals. Raw imaging data with single-molecule signals were analyzed with a modified version of MTT algorithm[115]. Briefly, single molecules are localized using two-dimensional Gaussian fitting followed by a generalized log-likelihood ratio test. Single molecule localizations detected in all 3000 frames were rendered with a two-dimension Gaussian filter to acquire reconstructed images corresponding to 50 nm localization accuracy, while density based spatial clustering of applications with noise (DBSCAN)[116,117] was performed to define the regions of clusters. For each clustered region of interest (ROI), localization detection and cumulative count of each frame were calculated and plotted, while a self-written denoising MATLAB script was conducted to automatically define temporal clusters (bursting events). Only bursts with more than 10 localization detection were selected to calculate several metrics for the following statistical analysis. In our study, stable clusters were defined as those temporal clusters lasting for at least half of the

acquisition window, while transient clusters, as report previously[117], were acquisition results within a limited temporal window and localization elsewhere were set to zero. Then the transient and stable clusters per cell were comparatively analyzed between before and after RUVBL2 degradation. Significant differences were calculated by two tailed Student's t test.

### Single-molecular RNA FISH

To examine the nascent RNA of the target gene, the probes targeted to the intron of *Bmp4* transcript were designed using the "Oligostan"[118] (Supplementary Data 6). The primary probes set were annealed with double Cy5 labeled secondary probes (FLAP-Y-Cy5)[118]. RUVBL2 degron cells were plated on coverslips and fixed by 4% paraformaldehyde for 20 min at room temperature after auxin treatment 1 h, washed twice with PBS and permeabilized in 70% ethanol overnight at 4 °C. The probe hybridization was performed according to the protocol published[118]. Simply, the hybridization buffer was prepared in 100 μl for 2 coverslips reactions which mix by an equal volume of mixture 1 (5 μl 20x SSC buffer, 1 μl 20 μg/μL *E. coli* tRNA, 15 μl 100% formamide, 2 μl FLAP-annealed probes, 26.3 μl DNA-RNase free water) and mixture 2 (1 μl 20 mg/mL RNase-free BSA, 1 μl VRC, 26.5 μl 40% dextran sulphate, 21.5 μl DNA-RNase free water, mix mix1 and mix2). Coverslips were immersed in the hybridization buffer and incubated at 37 °C overnight. Next day, the coverslips were washed using the freshly prepared 15% formamide/1x SSC at 37 °C for 30 min, and rinsed twice in PBS. For the staining of Pol II, coverslips were labeled using RPB1 antibody as described above. The coverslips were washed 3 times in PBS and finally mounted on slides using an anti-fade mounting medium and sealed with nail polish. The images were acquired using the Nikon Live SR CSU W1 microscopy with 100×/1.4 vc oil immersion objective and sCMOS Prime 95B camera. Based on the FISH loci focusing, each condition was captured in more than 10 different fields. Images were post-processed using Imaris 9.7 (Oxford). For analyzing the immunofluorescence signals of the *Bmp4* locus, all the channels were filtered medium to subtract the background noises and Gaussian filtered by default parameters in the Imaris 9.7 process menu. Then nascent RNA FISH foci were manually identified by adjusting the maximum and minimum brightness and γ contrast, and were defined as FISH "Surface". Simultaneously, the IF channel for all conditions was set with identical brightness and γ contrast. IF channel clusters were called using the "Spot" construction with "Different Spot Sizes (Region Growing)" and "Estimated XY Diameter (300 nm)" based on the "Absolute Intensity", which led to get the cluster diameter as real as possible but without size selection, then the "Shortest distance" between IF cluster and FISH foci were measured, only if the IF clusters overlapped with the FISH foci (Shortest distance<0) were measured and the integrate density (IntDen) were plotted by Prism 8.0. No less than 25 cells from independent fields for each condition were analyzed. The significance was calculated using the two-tailed Student's t test.

### In vitro assays

RPB1 CTD pull down, droplet, transcription initiation assay and DNA curtain. Recombinant proteins purification. For recombinant protein purification, full-length RUVBL1/2 RUVBL1/2 ATPase mutant, RUVBL1/2 DII and RUVBL1/2 DIDIII were cloned into the pRSFDuet1 plasmid for RUVBL2 N-terminal fused a His-tag, and RUVBL1 C-terminal fused with or without mCherry (for curtain assay) but no His-tag, which allowed us to get RUVBL1/2 complex through the His-tag affinity purification. RPB1 CTD52-mEGFP, and mEGFP were cloned into pET28a-sumo expression vector (gift from Dr. Yanli Wang, CAS, China) using Gibson ligation kit, Then the products transformed into Transetta (DE3) Chemically Competent Cells, induced to express using the 0.5 mM IPTG for 18–24 h at 18 °C. Sonication in protease inhibitors contained binding buffer were performed with 30% energy, 2 s ON, 3 s OFF, work

on 10 min. After sonication, 5 μl RNase A (Sigma Aldrich, R6148) and 5 μl Benzonase (Sigma Aldrich, E8263) for 400 ml LB medium was added and incubated for 30 min at RT. The soluble fractions were purified by affinity chromatography using Ni-NTA agarose following the manufacturer's instruction. To obtain the hexameric or dodecameric form RUVBL1/2-mCherry and RUVBL1/2 complex without fluorescent tags, His tag affinity column purified extracts were further purified by Superdex S200 (GE Healthcare, USA) size exclusion chromatography (SEC). The fractions with a molecular size of about 300–600 kDa were collected for CTD pulldown and in vitro droplet assays. The purified proteins were assessed using SDS-PAGE and western blot. High-quality eluents were combined, then buffer was exchanged and concentrated in storage buffer (20 mM Tris HCl (pH 7.4) and 500 mM NaCl) through Amicon Ultra centrifugal filters (Millipore, 50 K MWCO).

**RPB1 CTD pull down.** To examine the direct interaction between Pol II CTD and RUVBL1/2, CTD pull down assay was performed[119]. Briefly, the equal mole of mEGFP and CTD-mEGFP peptides (0.5 nmol) were immobilized with GFP-Trap magnetic beads in 50 μl of binding buffer (25 mM Tris-HCl, pH 8.0, 50 mM NaCl, 1 mM DTT, 5% glycerol, and 0.03% Triton X-100) for 1 h at 4 °C with low-speed rotation. After 3 times washing with the binding buffer, the RUVBL1/2 proteins with gradient concentrations (0.1 nmol, 0.25 nmol, and 0.5 nmol) were added and mixed with the CTD immobilized beads, and 0.5 nmol mEGFP were used as control, and then incubated for 30 min on ice. After 3 times washing with the binding buffer, the beads were resuspended in 100 μl 1x SDS-PAGE and boiled for 10 min. Aliquots of the bead-bound fraction (10%) were subjected to SDS-PAGE and visualized by western blot.

**Droplet assay.** An in-house protocol was performed for RUVBL1/2 and RPB1 CTD$_{52}$ droplet assay by following the overall consideration based on[12,22,25,120]. Briefly, RPB1 CTD$_{52}$-mEGFP and mEGFP were diluted to 10 μM in phase separation buffer (20 mM Tris HCl (pH7.4), 150 mM NaCl and 16% Dextran), RUVBL1/2-mCherry proteins were added to get the final gradient concentration (0.065 μM, 0.125 μM, 0.625 μM, which around the endogenous concentration in mESC nucleus), and incubated at room temperature for 0.5 h. After incubation, the droplet formation was immediately examined using Nikon A1R microscope with 100× oil-immersion objective based on the same parameters. The acquired images were post-processed using Fiji and analyzed as described above, but only the particles' minimal size filter was set to "100- Infinity". The condensed fraction was analyzed[121], simply, the integrated intensities in all droplets of the acquired field (I-in) and the total intensity outside the droplets (I-out) were measured with image J. Condensed fraction was calculated as (I-in)/((I-in) + (I-out)). To evaluate the contribution of ATPase activity and oligomerization in RUVBL1/2 enhancing RPB1 CTD cluster, the recombinant proteins of ATPase mutant, single RUVBL2, RUVBL1/2 DII and DI-DIII with similar nuclear concentration (0.125 μM) were performed droplet assay with 10 μM RPB1 CTD$_{52}$ as above under mock and or 20 μM ATP (RUVBL1/2 work concentration in vitro[65]). For GFP-EWS-FLI1 and CTD droplet assay, msfGFP-EWS(1-265)-FLI1(220-453) (GFP-EWS-FLI1), SNAP-EWS-FLI1 and Pol II CTD$_{N26}$-mCherry were purified by following the previous ref. 74. GFP-EWS-FLI1 were incubated with CTD$_{N26}$-mCherry with (0.167 μM) or without RUVBL1/2 (no fluorescent protein fusion) in the buffer containing 40 mM Tris-HCl (pH = 7.5), 150 mM KCl, 1 mM DTT, 2 mM MgCl$_2$ and 0.2 mg/ml BSA in 10 μl volume. The reactions were incubated for 30 min under RT. Data acquisition was performed using confocal microscopy (Leica TCS SP8, 100× Oil). Images were analyzed with Fiji. The background was subtracted by rolling ball radius of 50 pixels and the threshold was set automatically for particle analysis. Particles larger than 0.5 μm$^2$ and circularity under 0.98 were selected for the area and mean intensity analysis.

**Transcription initiation assay.** In order to investigate the direct role of RUVBL1/2 in transcription, the eukaryotic GTFs and *HDM2* promotor DNA were prepared and assembled, and in vitro transcription initiation assay was performed as previously described[71,72]. Briefly, 1.3 pmol of *HDM2* promoter DNA was combined with 0.75 pmol of TFIID, 1.5 pmol of TFIIA, 1.5 pmol of TFIIB, 1.5 pmol of TFIIF, 1.5 pmol of TFIIE, 0.75 pmol of TFIIH, either 1 pmol of *S. scrofa* Pol II with an increasing amount of RUVBL1/2 (0 μM, 0.0625 μM, 0.125 μM and 0.25 μM) in a volume of 10 μl containing 30 mM HEPES pH 7.9, 100 mM KCl, 6 mM $MgCl_2$, 2 mM DTT and 5% (v/v) glycerol at 25 °C for 30 min, mEGFP was used as the negative control. Reactions were initiated by adding an equal volume of buffer containing 24 mM HEPES-KOH pH 8.0, 120 mM KCl, 10 mM $MgCl_2$, 1.2 mM DTT, 24 % (v/v) glycerol, 100 μg/ml BSA, 200 μM GTP, CTP, ATP, UTP, respectively, and 99 nM [a-32P] UTP. Mix and incubated for 30 min at 25 °C, and then, add 20 μl 2× RNA loading dye into the mix to stop the reaction, heat for 5 min at 95 °C, chill on ice for 2 min, and spin briefly. Then the products were subjected to 10% denaturing acrylamide gel containing 7 M Urea in TBE buffer, electrophoresis at 100 V for 120 min. Finally, the undried gels are exposed to phosphorImager screens at −20 °C for visualization of radiolabeled transcripts. The nascent RNA bands were analyzed by Image J, and all the other bands were normalized to the no-RUVBL1/2 lane.

**DNA curtains.** The DNA curtain was set up followed previous protocols[122,123]. All experimental data were acquired with a prism-type total internal reflection fluorescence microscope (TIRFM) (Nikon Inverted Microscope Eclipse Ti-E). The experiment contained steps: (i) Use imaging buffer (40 mM Tris-HCl (pH=7.5), 150 mM KCl, 1 mM DTT, 2 mM $MgCl_2$, 0.2 mg/ml BSA, 1 mM DTT, and 0.2 nM YOYO-1) to extend Lambda DNA containing 25× GGAA motifs for 1 min at 0.4 ml/min and then 50 μl 1 μM SNAP-EWS-FLI1 loaded on the loading loop were flushed into the flowcell with imaging buffer. After 5 min washing, only proteins assembled on DNA were kept in the flowcell and stopped flow for 10 min incubation; (ii) Prepare samples of 1 μM $CTD_{N26}$-mCherry with or without 0.167 μM RUVBL1/2 and load it into the loop. Samples were injected into the flowcell at 1 ml/min flow rate, stopped flow as soon as the mCherry-labeled proteins reached into flowcell; (iii) After 30 min incubation, wash out free proteins with the imaging buffer at 0.4 ml/min and data were acquired using 2-s shutter with both 9.9 mW 488 nm laser and 28.2 mW 561 nm laser. DNA Curtains data was analyzed with Fiji to count the green puncta and magenta puncta numbers on DNA. We first defined the green puncta and magenta puncta by selecting the puncta that appeared with the curtain flow on, and disappeared with the curtain flow off. Then the number of Pol II puncta (magenta) colocalized with the EWS-FLI1 puncta divided by the total number of the EWS-FLI1 puncta (green) was the "Pol II CTD binding efficiency" described in the Fig. 5l.

## ChIP-Seq and ChIP-MS

ChIP-Seq and ChIP-MS were performed[42]. Briefly, cancer cells, wild-type mESCs or dox, and auxin-treated RUVBL2 degron mESCs and RPB1-CTD degron mESCs were dispersed by trypsinization and collected for downstream experiments. For crosslinking, formaldehyde (final concentration 1% (wt/vol)) was added and incubated for 10 min at room temperature (RT), then quenched by 0.125 M glycine (final concentration) for 5 min at RT. Cells were pelleted through 800 g for 5 min at 4 °C, washed twice with ice-cold PBS, split to 20 million aliquots per 1.5 ml Eppendorf tube, and then stored at −80 °C. For chromatin immunoprecipitation, 20 million crosslinked cells were lysed gently with 0.5 ml of ice-cold NP-40 lysis buffer (10 mM Tris·HCl (pH 7.5), 150 mM NaCl and 0.05% Nonidet P-40) on ice for 5 min, the cell lysates were then transferred on top of 1.25 ml sucrose cushion (24% sucrose (wt/vol) in NP-40 lysis buffer), centrifuged at 13,000 g for 10 min at 4 °C, discarded the supernatant of cytoplasmic fraction, the nuclei

pellets were washed once with 1 ml PBS/1 mM EDTA. Resuspended the nuclei pellet gently with 0.5 ml glycerol buffer (20 mM Tris-HCl (pH 8.0), 75 mM NaCl, 0.5 mM EDTA, 0.85 mM DTT, 50% glycerol (vol/vol)), then 0.5 ml nuclei lysis buffer (10 mM Hepes (pH 7.6), 1 mM DTT, 7.5 mM $MgCl_2$, 0.2 mM EDTA, 0.3 M NaCl, 1 M urea, 1% Nonidet P-40) were added and incubated on ice for 2 min. Centrifuged at 13,000 g for 2 min at 4 °C, discard the supernatant represented the soluble nuclear fraction. Washed the chromatin pellet twice with 1 ml PBS/ 1 mM EDTA, centrifuged at 13,000 g for 1 min at 4 °C, discarded the supernatant. The isolated chromatin was dissolved in 1 ml sonication buffer (20 mM Tris HCl (pH 8.0), 150 mM NaCl, 2 mM EDTA (pH 8.0),0.1% SDS,1% Triton X-100) with 5 mM $CaCl_2$, 200 U MNase (NEB, M0247) were added and incubated for 15 min at 37 °C with 700 rpm shaking, then 20 μl 0.5 M EDTA and 40 μl 0.5 M EGTA were added and mixed thoroughly on ice to inactivate MNase. The MNase digested chromatin fractions were divided into 300 μl per tube and sonicated using the bioruptor system (Diagenode, Bioruptor Plus) with high energy, 30 s ON, 60 s OFF for 20 cycles. Lysates were centrifuged twice at 13,000 g for 10 min at 4 °C, 20 μl supernatant were sampled as input, the rest supernatant was transferred into 2 new DNase free tube (500 μl/ tube), then 1 μl GFP abs (Abcam, ab290), and equal isotype IgG control (Normal Mouse IgG: Merck Millipore, 12-371, and Normal Rabbit IgG: Merck Millipore, 12-370), 1 μg RPB1-CTD abs (Abcam, ab26721), 1 μg INO80 abs (Proteintech, 18810-1-AP) and 1 μg P400 abs (Bethyl, A300-541A), and 5 μl RUVBL2 Rb pAb (ABclonal, A1905) were added for each ChIP reaction, respectively, and incubated overnight at 4 °C with low-speed rotation. Pre-washed 30 μl Protein G magnetic beads with sonication buffer were added into the mixture and incubated for another 4 h, then washed the beads through Magnetic frame once with sonication buffer, twice with high-salt wash buffer (20 mM Tris·HCl (pH 8.0), 500 mM NaCl, 2 mM EDTA, 0.1% SDS,1% Triton X-100), once with LiCl wash buffer (10 mM Tris HCl (pH 8.0), 250 mM LiCl,1 mM EDTA,1% NP-40), three times with TE buffer (1 mM EDTA,10 mM Tris HCl (pH 8.0)). For ChIP-Seq, beads were eluted with 300 μl elution buffer (50 mM Tris HCl (pH 8.0),10 mM EDTA,1% SDS) at 65 °C for 30 min with vigorously shake. Then the eluted DNA and input were decrosslinked with 4 μl Proteinase K (Invitrogen, AM2548, 20 mg/ml), incubated at 65 °C overnight and inactivated protease K at 80 °C for 20 min. DNA was purified through the Tris saturated equilibrium phenol (phenol: chloroform: isoamyl alcohol (25: 24: 1) (pH8.0), to examine the enrichment, ChIP-qPCR was carried out using 2× ChamQ SYBR qPCR Master Mix (Vazyme) on Bio-Rad CFX Connect™ Real-Time PCR Detection System according to the manual instruction. Then the ChIPed DNA was delivered to library preparation through TELP method[124], or the NEBNext Ultra II DNA library prep kit according to the manufacturer's instruction, the 200-400 bp length amplified DNA was recovered and analyzed with qubit assays and fragment analyzer, and finally subject to HiSeq Xten PE150 sequencing (Novogene, Beijing). For ChIP-MS, the beads and input were directly boiled twice using the SDS-PAGE loading buffer for 12 min[42], then the samples were separated with SDS-PAGE, the separated sample gels were cut (IgG bands were analyzed alone) and subjected to Orbitrap Fusion Lumos Tribrid Mass Spectrometer analysis.

## Transcriptome libraries preparation

RNA-Seq and PRO-Seq. For RNA-Seq, RUVBL2 degron mESCs were treated with 2 μg/ml Dox for 24 h, then 500 μM auxin (final concentration) were added into the medium, the cells after the auxin treatment 0 h, 0.5 h, 1 h, 6 h, 12 h, 24 h, and 48 h were harvested for total RNA extraction. Cells were lysed in Trizol reagent and extracted RNA according to the manufacturer's instruction, after examination the concentration and integrity of the total RNA, the untreated (Mock), and time-series auxin treated samples were delivered for poly(A) RNA library generation by Novogene Com., Beijing, and HiSeq Xten

PE150 sequencing (Novogene, Beijing). Two technical duplicates were performed. An equal number of cells was used to examine whether the total RNA abundance changed during the RUVBL2 degradation (Supplementary Fig. 5e). The results of RNA-Seq analyses were confirmed using RT-qPCR to examine the *C-MYC*, *CCND1*, *CHOP* and *ATF4* gene expression, and primers were available in Supplementary Data 6.

The precision nuclear run-on sequencing (PRO-Seq)[125,126] protocol was performed with some modifications. Briefly, Dox and auxin-treated RUVBL2 degron mESCs with ~80% confluency were harvested using the cell scraper and centrifugated at 1,000 *g* for 5 min at 4 °C. 3% *drosophila* S2 cell as spike-in cells were added to each sample and then washed once with 10 ml of ice-cold PBS. Cell pellets were rinsed twice with ice-cold permeabilization buffer ($1 \times 10^6$ cells per ml). Permeabilized cells were resuspended in a storage buffer ($10–20 \times 10^6$ cells per 100 μl of storage buffer) for nuclear run-on reaction. The 2× reaction mix of "1-Biotin run-on" with biotin-11-CTP was adopted for run-on in this study. Permeabilized cells (in storage buffer) were added into an equal amount of preheated 2× reaction mix by gently but thoroughly pipette the mixture, incubated for 3 min at 37 °C in a thermomixer with 700 rpm. Then, Trizol LS were added to stop the reaction. RNA extraction was performed according to the manual instruction of Trizol LS. The purified total RNA was heat-denatured at 65 °C, and immediately put on ice. The RNA were hydrolyzed by NaOH, and then neutralized by addition of 1 M Tris HCl (pH 6.8). Before biotin enrichment, buffer exchange was performed through a P-30 column according to the manufacturer's instruction, then the fragmented RNAs were added into streptavidin M280 beads for incubation 20 min at room temperature with low-speed rotation. The biotin enriched RNA fragments were thoroughly washed using ice-cold high-salt wash buffer, binding buffer and low-salt wash buffer, respectively, in addition, washed the beads twice with DEPC $H_2O$ to remove the nonspecific binding RNA fragments as much as possible. The enriched RNAs were extracted using Trizol reagent twice and then reverse-transcribed. Simply, SuperScript IV RT buffer (Invitrogen) included random primers were added into the RNA dissolved solution to synthesize first-strand cDNA, the excess primers were removed by Exonuclease I (Exo I) (NEB, M0293S). The RNA was eliminated by adding 1 M NaOH and then neutralized with 1 M HCl. The cDNAs were extracted using Tris saturated equilibrium phenol (pH 8.0). Finally, the purified cDNA was subjected to library preparation according to the TELP protocol[124]. The 150-400 bp length amplified DNA was selected for library analysis and finally subject to HiSeq Xten PE150 sequencing (Novogene, Beijing). Each condition performed two technical duplicates.

## MNase-Seq library preparation

The native cell MNase-Seq was performed according to the protocol described previously[127]. Briefly, Dox and auxin-treated RUVBL2 degron mESCs with ~80% confluency were fixed and collected by trypsinization and centrifugation (20 million cells/ 5–10 ml). The cell pellets were resuspended with ice-cold lysis buffer on ice to obtain the nuclei. Then the nuclei were resuspended with MNase digestion buffer with 5 mM $CaCl_2$ and 12.5 U MNase (NEB, M0247) per 1 million cells. The MNase was inhibited by the addition of EDTA and EGTA. The genomic DNA was released by proteinase K digestion and heat at 65 °C. The DNA was then extracted using an equal volume of Phenol/chloroform/ isoamyl alcohol (25:24:1) (Solarbio, P1012). The purified DNA was loaded onto 1% agarose gel for electrophoresis examination, and the excised mononucleosome bands (about 150 bp size) were cut and purified using gel extraction kit (Megen, D2111-03). The mononucleosome DNA was next constructed for the NGS library using the NEBNext Ultra II DNA library prep kit (NEB, E7645) by following the manufacture's instruction. Each condition was performed with two technical duplicates and finally subject to HiSeq Xten PE150 sequencing (Novogene, Beijing) and analysis.

## ChIP-Seq and ChIP-MS analysis

**ChIP-Seq analysis.** For ChIP-Seq data, paired-end sequencing reads were removed low-quality reads and adaptors with Cutadapt V1.18 (TELP libraries: -a CCCCCCCCCAGATCGGAAGAGCACACGTCTGAAC TCCAGTCAC, -A AGATCGGAAGAGCGTCGTGTAGGGAAAGAGTGT -q 15,15; Illumina Prep Kit libraries: -a AGATCGGAAGAGCACACGTCTGA ACTCCAGTCACC, -A AGATCGGAAGAGCGTCGTGTAGGGAAAGA -q 15,15)[128], and then the filtered reads were mapped to the mm10 or hg19 genome with bowtie2 (for TELP libraries, the parameter --very-sensitive-local was used; for Illumina Prep Kit libraries the default parameter was used.)[129]; in order to improve the accuracy of mapping, duplicate, discordant and multi-mapping reads were removed with Sambamba V 0.6.8[130]. In order to visualize the data, the final unique bam files were converted into bigwig files and visualized with Integrative Genomics Viewer (IGV)[131]. To get the genome-wide reliable binding sites of RUVBL1/2, peaks were called with the final unique bam files using MACS2 V2.2.1[132], and in mouse, only the peaks shared by RUVBL1/2 were used for downstream analysis and the peaks overlapped with blacklist (https://sites.google.com/site/anshulkundaje/projects/blacklists) were removed with BEDTools V2.25.0[133]. In humans, because there were only the ChIP-Seq data of RUVBL2, a more stringent cutoff is used, only peaks with IDR < 0.05 (IDR V2.0.3[134]) were used for downstream analysis and the peaks overlapping with blacklist were removed with BEDTools V2.25.0[133]. In order to identify the transcription factors that were enriched in the peak regions, the transcription factor motifs were searched with findMotifsGenome.pl from HOMER[135].

**Transcription factor binding sites clustering analysis.** In order to find the specific binding region of C-MYC, E2F1, P53, CTNNB1, and OCT4 in RUVBL1/2 overlapped peak regions, we downloaded the transcription factor ChIP-Seq data from GEO (with fastq-dump V 2.10.8 and V2.3.5) and Cistrome database[136], and the ChIP-Seq density of C-MYC, E2F1, P53, CTNNB1, and OCT4 in each peak was quantified with multiBigwigSummary V3.3.0 from deepTools[137]. Then the signal matrix was clustered into 10 categories with K-means method of pheatmap V1.0.12. To prevent the large differences in signal strength among various factors to dominate the clustering results, quantitative normalization was performed on the signals of each factor before clustering with preprocessCore V1.46.0[138]. Each category was verified on the track, and similar categories were merged together. Finally, deepTools V3.3.0[137] was used to display heatmaps in each category. The clustering of RUVBL2 in tumor cells used the same method.

**Bound and unbound gene definition.** In order to define the bound and unbound genes of RUVBL1/2, bedtools multicov[133,139] was used to calculate the signals of RUVBL1 and RUVBL2 at the gene TSS regions, then the sum of RUVBL1 and RUVBL2 ChIP-Seq signals in each gene's TSS were sorted, and the corresponding genes of the top 1000 TSS with the most high RUVBL1/2 signals were defined as bound genes, and the corresponding genes of the bottom 1000 TSS with the lowest RUVBL1/2 signals were defined as unbound genes.

**Definition of regulatory regions.** Super-enhancers: Super-enhancers were downloaded from a previous study[140], and the coordinates were converted to mm10 with CrossMap V0.2.7[141].Broad H3K4me3: Broad H3K4me3 were downloaded from a previous study[58] and the coordinates were converted to mm10 with CrossMap V0.2.7[142].

**ChIP-MS analysis.** To find reliable proteins that interact with RUVBL2 from the results of RUVBL2 ChIP-MS, IP/GFP >= 1.49 and IP/Input >= 0.5 were used as the cutoff, which could keep most of the proteins that were reported to interact with RUVBL2. In order to distinguish the complexes enriched in these proteins, the filtered proteins were searched in STRING Protein Network with Cytoscape V3.7.1[142].

To display these results, confidence score cutoff 0.8, maximum additional interactors 30 were used, and log10(IP/GFP) was assigned to color, and $\log_{10}$ (IP/Input) was assigned to size, then clustering with MCL cluster method (with Granularity parameter as 8) from clusterMarker[143] was performed. RPB1 ChIP-MS data were also filtered to get the reliable proteins with the cutoff (IP/IgG > 1.5 and IP/Input > 1.5).

### RNA-Seq and PRO-Seq analysis

**RNA-Seq analysis.** The paired-end sequencing fastq files were removed adaptors and low-quality reads with Cutadapt V1.18[128], then the filtered reads were aligned to mm10 genome with STAR V 2.7.3a[144]. And the bam files after removing duplicates were used to count the genes with annotation files from GENCODE (version M23)[145] with RSEM V1.3.1[146]. Differential analysis was performed with DESeq2 V1.24.0[147], and the significant genes were selected with foldchange >1.5 and FDR < 0.05. For the clustering analysis of RNA-Seq, a method in[148] was used. Briefly, differential analysis between any two time points were performed; then all genes that were differential expression in at least one comparison were selected; finally, the K-means clustering analysis was performed for these genes with pheatmap V1.0.12. To explore the characteristics of each category, clusterProfiler V3.12.0[149] was used to perform GO analysis, and findMotifsGenome.pl from HOMER[135] was used to find the motifs enriched in these gene's TSSs. The expression of housekeeping genes[150] at various time points were calculated and displayed with boxplot. For the aesthetic perception, the outliers in boxplot have been removed. In addition, the alternative splicing analysis was performed with the method in[151] with rMATS V3.1.0[152]. The differential analysis of alternative splicing events was performed between any two-time points with ΔPSI > 10% and FDR < 0.05 as cutoff.

**PRO-Seq analysis.** Cutadapt V1.18[128] was used to remove the adaptors (TELP libraries adaptor) and low-quality reads from the paired-end sequencing reads, then the filtered reads were mapped to the mm10 genome with bowtie2 (--very-sensitive-local)[129]. In order to obtain the more reliable nascent RNA signals, the duplicate, discordant and multi-mapping reads were removed, rRNA reads were also removed with split_bam.py from RseQC V 2.6.6[153]. To gain a more accurate transcription changes, 5% of drosophila cells were added to each condition during PRO-Seq library preparation. Then, the PRO-Seq signals from mESCs were normalized to the PRO-Seq signals from drosophila cells to avoid the interference of differential efficiency for the library among different conditions, or global changes. In order to display the information of the plus and minus chains at the same time, bamCoverage[137] was used to make the tracks for plus and minus chains separately with parameters –filterRNAstrand forward/reverse and –scaleFactor, and the scaling factor was determined by drosophila reads per million in the final unique bam files. And IGV[131] was used to visualize. To compare the changes of gene expression after RUVBL2 degradation, we calculated the read counts on the sense strand of each gene body (300 bp downstream of TSS to TES) with featureCounts (v1.6.3)[154], then DESeq2 was used for differential analysis. The number of drosophila reads is used as the spike-in to adjust the size factors. And the results of the differential analysis were illustrated as volcano plot. In order to explore whether the gene expression changes are associated with the gene lengths, the expressed genes were divided into 7 categories (<1 kb, 1–5 kb, 5–10 kb, 10–25 kb, 25–50 kb, 50–75 kb, >75 kb), and the changes of PRO-Seq signals at these 7 categories of genes were showed separately. In order to draw the profiles of PRO-Seq at each cluster's genes, computeMatrix[137] was used to calculate the density of PRO-Seq on these genes' regions, then the signals in top 5% and tail 5% were removed and ggplot2 was used to display the plus and minus chain profiles simultaneously. For the quantification of PRO-Seq signals in promoters, the signals in sense strand at ±200 bp around the TSSs were quantified with multiBigwigSummary V3.3.0 from deepTools[137],

and displayed with boxplot. For the quantification of PRO-Seq signals in gene bodies, the signals in sense strand from 300 bp downstream of TSS to TES were used to quantify the PRO-Seq signals at the gene bodies with multiBigwigSummary V3.3.0[137], and also displayed with boxplot. For the aesthetic perception, the outliers in boxplot have been removed.

### Identification of RUVBL2 direct target genes and post-transcriptional early response genes

RUVBL2 direct target genes identification. To identify the genes directly regulated by RUVBL2, the genes with the same changing trend in PRO-Seq 0.5 h and 1 h (signals in the gene body: 300 bp downstream of TSS to TES; FDR < 0.05, FC < −1.5) were selected. To ensure these RUVBL2 effected genes are indeed affected by RUVBL2. The RUVBL2 affected genes owing the similar pattern in the mature mRNA level at 6 h (RNA-Seq expression signals; FDR < 0.05, FC < −1.5) and bound by RUVBL2 were regarded as RUVBL2 direct target genes, and finally 45 direct target genes were identified. To explore the RUVBL2 binding in these genes' regions in tumor cells, direct target genes were used to find the homologous genes in human with homologene V1.4.68[155] and the corresponding RUVBL2 binding profiles were plotted at the genes' regions of these genes in each human cell line.

### Posttranscriptional early response genes identification

To identify genes that are not directly regulated by RUVBL2 at transcription level, but respond very early at the posttranscriptional level, the genes owing the similar pattern in RNA-Seq 0.5 h, 1 h and 6 h and owing the opposite pattern with PRO-Seq signals changes at the gene bodies were selected. For posttranscriptional early response up-regulated genes, the up-regulated genes in RNA-Seq 0.5 h, 1 h and 6 h (FDR < 0.05 & FC > 1.5) were selected, and these genes that didn't increase the PRO-Seq signals of gene bodies at 0.5 h and 1 h conditions (FDR < 0.05 & $\log_2$ (FC) < 0.1) were further selected. For posttranscriptional early response down-regulated genes, the down-regulated genes in RNA-Seq at 0.5 h, 1 h and 6 h conditions (FDR < 0.05 & FC < −1.5) were selected, and these genes that didn't decrease the PRO-Seq signals of gene bodies at 0.5 h and 1 h conditions (FDR < 0.05 & $\log_2$ (FC) > 0.1) were further selected. In order to explore the RUVBL2 binding in these genes' promoters in tumor cells, homologene V1.4.68[155] was used to identify the homologous genes of the posttranscriptional early response genes in human and the corresponding RUVBL2 binding profiles were also plot in these genes' regions in each human cell lines.

### Identification of potential TFs together with RUVBL2 to regulate direct target genes

In order to identify the potential transcription factors that regulate the RUVBL2 direct target genes, we first sampled the same amount genes with similar expression level to the RUVBL2 direct target genes from genes that were not changed at PRO-Seq 0.5 h condition (FDR > 0.1 & |log2FC| <= 0.1), which were used as RUVBL2 nontarget genes. Then 153 ChIP-Seq bigwig files of chromatin-associated proteins were downloaded from the cistrome database[136]. The signals of these 153 factors in target and nontarget genes' promoters were calculated with multiBigwigSummary from deepTools[137]. To perform effective feature selection, the signals after scaling and centering were divided into training (75%) and test (25%) sets. And elastic net, gradient boosting machine, and SVMs with linear kernels were trained using leave-one-out cross-validation (elastic net, and SVMs using CARET package[156], gradient boosting machine using scikit learn module), and the performance of each model was tested with the test sets. Due to the high interpretability of elastic net, the coefficients of 153 factors in predicting target genes can be clearly given. We finally selected the 10 factors with the largest and smallest coefficients in the elastic net for downstream analyses. The signals of these 153 factors in target and

nontarget genes promoters were extracted to plot the heatmap with pheatmap V1.0.12.

## MNase-Seq analysis

The adaptors and low-quality reads (MAPQ ≤ 15) were removed from paired-end sequencing reads, then the remained reads were mapped to the mouse genome mm10 with bowtie2 with default parameters[129]. The duplicate, discordant and multi-mapping reads are removed, and reads in the range of 160–190 bp were selected as the classic nucleosome with alignmentSieve from the deepTools[137], and the midpoint of the fragment was defined as the position of the nucleosome. In order to explore the changes of MNase at the genome-wide level, bamCoverage (--MNase) from deepTools[137] was used to obtain the MNase-Seq track. In order to get a smoother track, gsmooth function of scipy (python package) was used to smooth the track. The final tracks were used for meta-analysis in different chromatin regions. In order to obtain the MNase-Seq profile around the CTCF binding sites, the CTCF motif MA0139.1 was downloaded from the JASPAR database[157], and FIMO[158] was used to detect the CTCF motif within the CTCF peaks at genome-wide level.

## Cancer-related analysis

The expression levels of RUVBL1/2 in each tissue were downloaded from GTEx[159], the expression levels of RUVBL1/2 in tumor and normal tissues were downloaded from GEPIA database[160]. The survival analysis of RUVBL1/2 was downloaded from Kaplan-Meier Plotter[97–99] (https://kmplot.com/analysis/). The Cancer hall markers were downloaded from a previous study[161], and RUVBL2 direct target genes and the posttranscriptional early response genes presented in the hall markers were selected to draw the donut chart.

## Reporting summary

Further information on research design is available in the Nature Research Reporting Summary linked to this article.

## Data availability

The data that support this study are available from the corresponding author upon reasonable request. The RNA-Seq, PRO-Seq and ChIP-Seq data generated in this study have been deposited in the GEO database under accession code GSE160739 (RNA-Seq: GSE160698, PRO-Seq: GSE160696, ChIP-Seq: GSE160738). The MNase-Seq data generated in this study have been deposited in the GEO database under accession code GSE185347. The mass spectrometry proteomics data have been deposited to the ProteomeXchange Consortium via the PRIDE partner repository with the dataset identifier PXD036528. Published sequencing datasets analyzed in this study are listed in Supplementary Data 5. Source data are provided with this paper.

## Code availability

The analysis code in this study have been deposited in Github [https://github.com/lbyybl/RUVBL2].

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

## Acknowledgements

We thank the members of the Ji laboratory for engaging in helpful discussions. We thank Dr. David L. Spector for sharing the LacR-Lac operator imaging system. We thank Dr. Jie Huang for helping with RNA splicing and decay-related analyses. This work was supported by funds from the National Key R&D Program of China and the National Natural Science Foundation of China (grants 2017YFA0506600, 31871309 and 32170569) and the Qidong-SLS Innovation Fund and by grants from the

Peking-Tsinghua Center for Life Sciences and the Key Laboratory of Cell Proliferation and Differentiation of the Ministry of Education at Peking University School of Life Sciences to X. J. This research was supported in part by the State Key Research Development Program of China (2021YFE0201100 and 2017YFA0505503), the National Natural Science Foundation of China (81890991), Beijing Municipal Natural Science Foundation (Z200021), and CAS Interdisciplinary Innovation Team (JCTD-2020-04) to J.G. We thank the National Center for Protein Sciences at Peking University in Beijing, China, for assistance with high-resolution fluorescence imaging and Ms. Liyin Du for help with flow cytometry sorting, Ms. Dong Liu for help with mass spectrometry detection, and Mr. Shitang Huang for technical assistance in the transcription assay involving [γ–32P] UTP. H.W. was supported in part by a Postdoctoral Fellowship from the Peking-Tsinghua Center for Life Sciences.

## Author contributions

X.J. conceived and supervised the project. H.W. performed most of the experiments. B.Y.L. performed all the bioinformatic analyses. L.Y.Z. and Z.Q. performed the Pol II in vitro cophase and DNA curtain assays. K.T. performed gene expression analyses involving cordycepin treatment. R.Z. performed the RNA-Seq validation analyses. X.Z.C. and Y.H.X. helped with the in vitro transcription assay. C.L.W., H.N.Z. and F.F.Q helped with the chromatin-associated protein analyses. Y.P.J. performed the RPB1 ChIP-MS. B.Z. and Y.Y. helped with the tethering experiments. B.W. and W.L.D. performed the tcPALM imaging. Y.Y., and J.T.G. helped with the RNA FISH experiment. C.P.L. helped with in vitro RUVBL1/2 protein purification. All authors contributed to the data analyses and data interpretation. X.J. wrote the manuscript with input from H.W., B.Y.L. and assistance from the other authors.

## Competing interests

The authors declare no competing interests.

## Additional information

¹Key Laboratory of Cell Proliferation and Differentiation of the Ministry of Education, School of Life Sciences, Peking-Tsinghua Center for Life Sciences, Peking University, Beijing 100871, China. ²Department of Pathogenic Biology, Chengdu Medical College, Chengdu 610500, China. ³Center for Quantitative Biology, Peking-Tsinghua Center for Life Sciences, Academy for Advanced Interdisciplinary Studies, Peking University, Beijing 100871, China. ⁴Biomedical Pioneering Innovation Center (BIOPIC), Academy for Advanced Interdisciplinary Studies, Beijing Advanced Innovation Center for Genomics (ICG), Peking-Tsinghua Center for Life Sciences (CLS), School of Life Sciences, Peking University, Beijing 100871, China. ⁵Institute for TCM-X; MOE Key Laboratory of Bioinformatics; Bioinformatics Division, BNRist (Beijing National Research Center for Information Science and Technology); Department of Automation, Tsinghua University, Beijing 100084, China. ⁶Center for Synthetic and Systems Biology, Tsinghua University, Beijing 100084, China. ⁷Fudan University Shanghai Cancer Center, Institutes of Biomedical Sciences, Shanghai Medical College of Fudan University, Shanghai 200032, China. ⁸Synthetic and Functional Biomolecules Center, Beijing National Laboratory for Molecular Sciences, Key Laboratory of Bioorganic Chemistry and Molecular Engineering of Ministry of Education, College of Chemistry and Molecular Engineering, Peking University, Beijing 100871, China. ⁹Departments of Pathology and Laboratory Medicine, and Pediatrics, University of Rochester Medical Center, 601 Elmwood Ave, Box 608 Rochester, NY 14642, USA. ¹⁰Key Laboratory of RNA Biology, CAS Center for Excellence in Biomacromolecules, Institute of Biophysics, Chinese Academy of Sciences, Beijing 100101, China. ¹¹National Laboratory of Biomacromolecules, CAS Center for Excellence in Biomacromolecules, Institute of Biophysics, Chinese Academy of Sciences, Beijing 100101, China. ¹²These authors contributed equally: Hui Wang, Boyuan Li. ✉e-mail: xiongji@pku.edu.cn

