## [Peer Review File · Nature Communications]

REVIEWER COMMENTS

Reviewer #1 (Remarks to the Author):

This article by Wang et al shows that RUVBL2 works as a co-activator for many transcription units by co-occupying promoter DNA with Pol II and transcription factors. Indeed, depletion of RUVBL2 decreases the number of Pol II clusters and nascent RNA synthesis, while tethering of RUVBL2 has the reverse effect. The authors further identify target genes that are linked to what they call the RUVBL2-Pol II axis, many being involved in cancer. The authors conclude for a novel activity for RUVBL2 in regulating Pol II cluster formation in the nucleus.

Overall, this is a dense, but interesting article. The conclusions are generally well supported by the data. The various experiments are well controlled and provide new mechanistic conclusions. However, I have criticisms that should be addressed.

Comments:

Fig 1. RUVBL2 associates with POL II in chromatin. What about POL II not in chromatin? Would it be possible that RUVBL2 also associates with POL II during enzyme assembly in the cytoplasm, and/or with nuclear POL II disengaged from chromatin? Has this been tested directly? RUVBL2 has been shown to be part of the so-called PAQosome/R2TP, a HSP90/HSP70 co-chaperone complex involved in the cytoplasmic assembly of Pol II (prior to nuclear import) and many additional complexes (other nuclear RNA pol, splicing factors, the ribosome, kinase networks, etc). Is the transcriptional co-activator function of RUVBL2 reported here a consequence of its co-chaperone function by assisting pre-initiation complex formation on the chromatin? Are other PAQosome/R2TP subunits (RPAP3, prefoldins, etc) found in the RUVBL2-Pol II clusters?

Quantification data described at lines 233-239 shows very weak increases (48%, 60%, 64%). These pretty modest increases may be misleading. Can the authors explain why we should trust these results and the conclusions they lead to. Do alternate ways to do these experiments exist, leading to more convincing results?

Reviewer #2 (Remarks to the Author):

Wang et al. focused on transcriptional regulation of chaperones and identified the molecular chaperones RUVBL1/2 that could interact with RNA-polymerase II (Pol II) large subunit (RPB1), as evident from ChIP-MS analysis. Although the involvement of RUVBL1/2 was suggested in transcriptional regulation through interactions with specific transcription factors, the detailed mechanism was not clear owing to their involvement in RNA processing and other biological phenomena. In this study, the authors found a direct interaction between RPB1 and RUVBL1/2 through biochemical experiments. They also found that RUVBL1/2 may promote Pol II clustering, which may be a mechanism for enhancing transcriptional activity. Furthermore, from PRO-seq analyses with the auxin degron system, they found that RUVBL1/2 could directly regulate transcription initiation. RNA-seq analysis indicated the involvement of RUVBL1/2 in post-transcriptional regulation through RNA processing and other mechanisms.

I recognize that the novelty of this study is high enough, as the authors show that RUVBL1/2 is involved in Pol II clustering, and suggest, through different experimental approach, that these factors might directly regulate transcription. The absence of ATP in the droplet experiments indicates that the chaperone function of RUVBL1/2 is not involved in RPB1 clustering; however, this has not been fully investigated. RUVBL1/2 have been proposed to promote clustering via RPB1 CTD, but there is insufficient evidence. In addition, the structure of the paper is not well organized and needs substantial revision, and resubmission of the paper is recommended.

Major comment

- RUVBL1/2 are known molecular chaperones, involved in Pol II assembly in the cytoplasm and numerous other vital functions; in the droplet experiment, addition of RUVBL1/2 promoted RPB1 clustering in the absence of ATP (Figure 2c). This indicates that the chaperone activity of RUVBL1/2 was not involved in RPB1 cluster formation. In addition, appropriate control experiments like adding mCherry instead of RUVBL1/2-mCherry were not performed; therefore, it is not possible to determine whether RUVBL1/2 promotes clustering or it is an artifact owing to an increased protein concentration. The authors probably think that chaperone activity promotes RPB1 clustering (line 644). LacR experiments with RUVBL1/2 mutants (if can be generated), that can oligomerize but have no chaperone-like activity, may show whether RUVBL1/2 chaperone activity is required for RPB1 recruitment. LacR experiments, using a mutant that suppresses oligomerization (if such mutants are available), can indicate whether oligomerization is important for RUVBL1/2 chaperon function or RPB1 recruitment.
- Using the RPB1 degron cell line and droplet analysis, the authors conclude that RUVBL1/2 promote RPB1 clustering via RPB1 CTD. However, auxin treatment for 1 h did not completely eliminate RPB1-CTD (Figure 1I). Furthermore, in IP assays using the RPB1-NTD antibody, RUVBL1/2 binding was not completely inhibited. Therefore, it is not still clear whether RUVBL1/2 binds only to the CTD of RPB1 or to other domains within RPB1. RPB1 has a tendency to form clusters via the CTD. In addition, oligomerized RUVBL1/2 were used in this experiment, and it is conceivable that there are multiple RPB1 binding sites within the complex, promoting aggregation via multivalent interactions (Musacchio, EMBO J, 2022). One main topic of this paper is that RUVBL2 promotes Pol II clustering, and this point should be clearly emphasized.
- One key claim of the paper is that "Mechanistically, RUVBL2 enhances LCD-LCD interactions between transcription factors and Pol II CTD." However, only EWS-FLI1 has been examined for co-clustering with RPB1, and therefore, it cannot be generalized as a phenomenon. ChIP-seq data show the colocalization of RUVBL1/2 with many transcription factors (Figure 7B); however, its relationship to RPB1 clustering is not clear. Indeed, while INO80 appears to co-localize on ChIP-seq (Extended Data Figure 3B), it does not interact directly with Pol II (Figure 1G). Also, the relationship to LCD-LCD interaction has not been examined at all, and it is extremely wild to come to this conclusion. Did RUVBL2 promote EWS-FLI1 to form puncta alone? If RPB1 can bind to both EWS-FLI1 and RUVBL2, and if RUVBL2 further facilitates RPB1 recruitment, then the addition of RUVBL2 may allow a definite detection of RPB1 condensate and therefore, "RUVBL2 enhances the binding between EWS-FLI1 and Pol II CTD" is not evidenced.
- Several unnatural points are recognized in the current structure of the paper, probably because of a drastic change in the paper structure. In particular, the degron system is explained again in Figure 5 even though it was already used in Figure 1. Therefore, the entire paper structure has to be drastically reorganized and resubmission should be considered.

Minor comment

- Line 116: "we identified 45 direct transcriptional targets of the RUVBL2-Pol II axis for the first time (i.e., c-Myc and Junb, etc.)" This has been done using mouse ES cells only. Therefore, it needs to be stated that this fact was found in mESC.
- Line 117: " Further, we found that these genes are involved in diverse molecular functions, indicating that the RUVBL2-Pol II axis is essential for various cellular functions in different biological contexts." This comment seems overstated as complete evidences are not presented in the manuscript.
- Figure 1A: This model cannot account for the condensate-mediated transcriptional regulation claimed in this paper; it should be changed to a figure similar to that shown in Fig, 1(a) by Lim et al., Curr Opin Genet Dev, 2021.
- Line 142: The subunits of RNA pol I and III, such as POLR1C, POLR1B, and POLR3B, are included as interactors with RUVBL2 (Figure 1B). Is it conceivable that RUVBL1/2 are also involved in other RNA pol functions? If so, it should be discussed in the Discussion.
- Line 145: Fig. 1C shows only overlapping peaks of RUVBL1/2, but it seems that their signal peaks do

not necessarily overlap in the RUVBL1/2 ChIP-seq data in Figure 6C. Please show the quantitative data of overlap between RUVBL1 and RUVBL2 ChIP-seq (as supplementary data).

- Line 152: Here, the authors suddenly mention about INO80 and P400. Since there are potential readers from a different and even unrelated fields, a prior explanation is necessary regarding the interaction of RUVBL1/2 with these factors.
- Line 162, "Gerber et al," should be used to cite references and the use of full names should be avoided.
- In Fig. 1, the name "RPB1 CTD" appears, but "PolII CTD" is used in other parts of the manuscript. The paper should use RPB1 CTD throughout.
- Line 171: POLR2E, which corresponds to RPB5, is detected by ChIP-MS of RUVBL2 (Figure 1B). Therefore, assuming that Pol II is assembled in the cytoplasm and transported to the nucleus, discuss the possibility (if any) of RUVBL2 binding to Pol II and their co-transportation to the nucleus.
- Although "data not shown" has been used several times, appropriate data should be included in support of claims or comments in the Supplementary information in accordance with the submission rules of Nature communications. <https://www.nature.com/ncomms/submit/how-to-submit>
- Line 206: The sentence containing "which is reminiscent of..." should be stated in the Discussion.
- Line 216: GFP has been used as a negative control, but the dose is not listed in Methods or Legend.
- Line 224: The claim "RUVBL1/2 promoted the cocondensation of EWS-FLI1 and Pol II CTD" is misleading. The Methods section states that this experiment uses human RPB1 with a hepta repeat count of 26, not 52, in the CTD. It has already been reported that the number of repeats can change RPB1 properties. Quintero-Cadena et al., Mol Cell, 2020; Sawicka et al., EMBO J, 2021. Even if the data has been used as it is, the number of hepta repeats should be clearly mentioned with the statement that the results are for RPB1 with 26 hepta repeats.
- Figure 2k: The Roman numerals related to Figure 2j are confusing. Please use different expression patterns.
- Line 243: "The GFP" should be written as "the GFP".
- In line 244, what "RUVBL2 genomic localization" refers to is not clear, and a suitable explanation is needed.
- Line 280: Data show that RUVBL1/2 co-localization peak and Pol II peak co-localize; however, the possibility of localization of RUVBL2 alone at the promoter has not been verified or clarified with data. RUVBL2 has been shown to localize at super-enhancers; therefore, data should be presented to show whether RUVBL2 has a real tendency to localize at the promoters.
- Line 291: "Additionally, auxin treatment for..."- this sentence needs to be more specific for a clear understanding.
- Line 371: "Western blot analysis...". How many hours after Auxin treatment did the degradation happen? Moreover, the authors used both words, IAA and Auxin, in the manuscript. Please pick any one of them and use it consistently throughout the manuscript.
- Figure 5e: The labels are significantly misaligned.
- Line 406: I did not understand the rationale behind motif analysis based on RNA-seq data when PRO-seq data was already shown; it should be re-analyzed using PRO-seq data. RNA-seq showed little change with Auxin treatment for 0.5 or 1 h except for genes, such as c-Myc, that responded very quickly; therefore, we may not see a direct effect on transcriptional regulation by RUVBL2, and this data is unnecessary.
- Figure 5h: The quantitative values for localization are not displayed, and therefore, the difference is not apparent.
- Figure 5j, I would like to see the percentages of genes included in each cluster, as the number of genes included in each cluster is different to begin with. However, on an average, only about 30% of the genes overlapped. Therefore, I did not find the basis for saying that they "largely overlapped."
- Line 435: "smg1" should be written as "SMG1".
- Line 440: This part should be moved closer to the citation of Figure 1g, j in the text.
- Line 450: "RUVBL2 also interacted..." This should be stated before the citation of Figure 5K, L in the text.
- Lines 462-463: The corresponding Extended Data Figure 3d-e data are missing.
- Line 485: The use of the word "Here" is incorrect because the PRO-seq data is already shown.

- Figure 6B and Extended Data Figure 4C: 0 h (Dox) data should be shown.
- Line 555: In the testis, RUVBL2 was highly expressed, and not overexpressed.
- Figure 6h: Figures should be prepared according to their order of appearance in the text.
- The first three sentences from line 606 should be stated in the Introduction. Otherwise, the reader will not understand the importance of this paper.
- In figures using Degron experiments (e.g., Figure 3b), the control experiment is denoted as "Dox". The sample to which Auxin was added also contained Dox, labeling it as "0 h" will be easy to understand.
- I do not understand why the analysis in Figure 1J focused on c-Myc, Slc16a3, and Junb as RUVBL2 target genes. They are direct target genes identified from PRO-seq and other analyses, but this has not been mentioned in the text and Fig. 1J. Therefore, an explanation is needed.
- Line 774: "mEGFP was used as the control, and RUVBL1 and RUVBL2 were examined to evaluate the protein-protein interaction frequencies." I do not understand what "frequencies" refers to.
- I do not understand what is represented by the data in Fig. 2E.
- Fig. 2L: A detailed explanation of what is being quantified is needed in the legend. Moreover, there is a reference to "red bar" in the legend, but no red bar is noticed in the figure.
- Fig. 3A: the statement "DNA was stained with DAPI." is not necessary because the image of DAPI staining is not shown.
- Fig. 3B: "Scale bar, 3 μ m." should be put in Fig. 3A.
- Fig. 3F, (left) and (right) may be mistaken for (upper) and (lower). Moreover, I do not understand know what the vertical axis of the boxplot indicates.
- Fig. 4E: Does the term "endogenous antibodies" mean antibodies that recognize endogenous proteins?
- Fig. 5H: "in the methods section" should be presented as "in the Methods section". A similar notation has been found elsewhere and needs to be corrected.
- Fig. 7B: The CTNNB1 motif is not shown in Fig. 7A and should be added here.
- Methods section: Please describe the origin of the mESC strain.
- Methods section: Please include information on the antibodies used.
- Methods section: In the "LacR tether imaging and data analysis" section, "IFA" has been mentioned without defining the term.
- Methods section: In describing GFP-EWS-FLI1 and CTD droplet assay spinning-disk confocal microscopy (Leica TCS SP8, 100 \times Oil)" has been mentioned. I guess that SP8 is not a spinning disk confocal microscope.
- The abbreviation BPB has not been defined in full.
- In the section "MNase-Seq library preparation", the description is in protocol form. In the paper, it should be described in the past tense. Also, "gel extraction kit" has not been described in detail. In addition, the sources and model numbers of the reagents used have not been described in several instances, therefore, making it impossible for the reader to conduct reproducible experiments.
- PRO-seq analysis, "To get the more accurate transcription changes, the spike-in of drosophila cells was used to normalized the PRO-Seq signals." A detailed explanation of this comment is necessary.
- deepTools (Ramirez et al., 2014): The citation format is incorrect.

Reviewer #3 (Remarks to the Author):

In this manuscript, the authors showed that RUVBL2 is involved in the regulation of Pol II cluster formation and transcription regulation. At first, the authors performed Pol II ChIP-MS using ES cells and identified many of proteins involved in transcription regulation, RNA splicing, cell cycle regulation and other cellular functions. Among the identified proteins, the authors focused on RUVBL1 and RUVBL2. The authors showed that RUVBL2 interacts with unphosphorylated Pol II CTD in vitro and promotes Pol II CTD clustering and transcription activation in cells. Rapid depletion of RUVBL2 using auxin-induced degradation system leads to the decreased number of Pol II cluster formation and inhibits global transcription in cells. Of note, tethering of RUVBL2 enhances Pol II clustering at the

active promoters in cells. Furthermore, the authors performed both RNA-seq and PRO-seq analysis and identified direct target genes of the RUVBL2, which include c-Myc, Junb and other genes. Many of these genes are hallmarks of cancers and encode proteins with diverse cellular functions. These results suggest novel activity for RUVBL2 in regulating transcription through Pol II cluster formation in the nucleus. Based on their findings, the authors propose that RUVBL2 functions as a “molecular chaperon” for Pol II clustering, however there is not sufficient evidence supporting this model.

Major comments,

(1) In this manuscript, critical point which should be addressed in the paper is whether RUVBL2 functions as a “molecular chaperon” for Pol II clustering, however the authors did not show the evidence supporting this important point. At least, the authors should test whether ATPase activity of RUVBL2 is required for Pol II clustering (Pol II foci formation), assembly of a variety of transcription factors with Pol II at RUVBL2 target genes (ChIP-seq), and transcription activation of these target genes (RNA-seq, PRO-seq). I recommend taking advantage of the mutant form of RUVBL2 which loses the ATPase activity and replacing the endogenous RUVBL2 with the exogenously expressed mutant form of RUVBL2. Otherwise, some specific inhibitors for RUVBL2 are also useful for these experiments.

(2) The authors propose the model that RUVBL2 directly promotes Pol II CTD clustering by enhancing LCD (Low complexity domain)-LCD interaction. To support the idea, the authors performed the in vitro droplet formation assay and showed that RUVBL1/2 promotes liquid droplets formation of Pol II CTD in RUVBL1/2 dose dependent manner (Figure 2). Did the author test whether the inhibitors of phase separation such as increased NaCl concentration or 1,6-hexanediol inhibits the droplets formation of Pol II CTD in vitro? In addition, does the ATPase mutant form of RUVBL1/2 lose the activity to promote droplet formation of Pol II CTD in vitro? In addition, the authors found that depletion of RUVBL2 decreased the liquid droplets of Pol II and CTCF in cells (Figure 3). Did the authors test whether 1,6-hexanediol inhibits these droplets formation in cells?

(3) The authors performed the in vitro transcription assay and showed that RUVBL1/2 activates the transcription initiation (Figure 2G), however the enhancement by RUVBL1/2 is very little in the in vitro transcription assay. Here, the critical question is how RUVBL1/2 activates transcription in vitro? The author should explain this mechanistic point in the manuscript. In addition, is the ATPase activity of RUVBL1/2 required for transcription activation in vitro?

(4) The authors showed that RUVBL1/2 is required for cell growth and involved in carcinogenesis (Figure 5 and Figure 7). Does the ATPase activity of RUVBL1/2 contribute to cell growth and carcinogenesis?

Minor comments,

(1) In this manuscript, the main story is that RUVBL2 functions as a molecular chaperon for Pol II clustering and plays a role in transcription activation of the target genes, however there are the results shown in the Figure 5, which do not directly or positively support the author’s main story. I recommend moving the Figure 5 to supplementary figure for better understanding the main story.

(2) The authors performed PRO-seq analysis and showed that depletion of RUVBL2 decreased the global transcription (Figure 3), however the depletion of RUVBL2 little affected the transcription of direct target genes of RUVBL2. The authors should explain the reason.

(3) In Figure 7, the author proposes that RUVBL2 regulates the transcription of direct target genes with a variety of transcription factors including c-Myc and E2F1. Does the RUVBL2 contribute to recruitment of those transcription factors to the promoters of direct target genes?

Reviewer #1

This article by Wang et al shows that RUVBL2 works as a co-activator for many transcription units by co-occupying promoter DNA with Pol II and transcription factors. Indeed, depletion of RUVBL2 decreases the number of Pol II clusters and nascent RNA synthesis, while tethering of RUVBL2 has the reverse effect. The authors further identify target genes that are linked to what they call the RUVBL2-Pol II axis, many being involved in cancer. The authors conclude for a novel activity for RUVBL2 in regulating Pol II cluster formation in the nucleus. Overall, this is a dense, but interesting article. The conclusions are generally well supported by the data. The various experiments are well controlled and provide new mechanistic conclusions. However, I have criticisms that should be addressed.

Response: We appreciate the reviewer's statement that our work is interesting and provides new mechanistic conclusions. The conclusions are well supported by the data, and the experiments are well controlled. We addressed the reviewer's criticism, as described in the response below.

Comments:

Fig 1. RUVBL2 associates with POL II in chromatin. What about POL II not in chromatin? Would it be possible that RUVBL2 also associates with POL II during enzyme assembly in the cytoplasm, and/or with nuclear POL II disengaged from chromatin? Has this been tested directly? RUVBL2 has been shown to be part of the so-called PAQosome/R2TP, a HSP90/HSP70 co-chaperone complex involved in the cytoplasmic assembly of Pol II (prior to nuclear import) and many additional complexes (other nuclear RNA pol, splicing factors, the ribosome, kinase networks, etc). Is the transcriptional co-activator function of RUVBL2 reported here a consequence of its co-chaperone function by assisting pre-initiation complex formation on the chromatin? Are other PAQosome/R2TP subunits (RPAP3, prefoldins, etc) found in the RUVBL2-Pol II clusters?

Response: RUVBL2 is part of the PAQosome/R2TP complex, which is known to assemble the Pol II complex in the cytoplasm^{1,2}. We agree with the reviewer that it is possible that RUVBL2 also interacts with disengaged Pol II in the nucleoplasm. We examined the interactions of RUVBL2 and other R2TP subunits like RPAP3 with Pol II in isolated cytoplasm, nucleoplasm, and chromatin fractions and found that RUVBL2 interacts with Pol II in all fractions, while RPAP3 mostly interacts with Pol II in the cytoplasm but not in chromatin fractions (Figure R1). We also agree with the reviewer that RUVBL2 may assist in preinitiation complex formation. Thus, we performed initiation factor TBP stimulated emission depletion microscopy (STED) imaging and found that RUVBL2 also contributes to preinitiation cluster formation in the nucleus (Figure R2). We added these figures to the revised manuscript.

Figure R1. Western blot to examine RUVBL2 and RPAP3 in RPB1 immunoprecipitation (IP) from different subcellular fractions. Cyto is the cytoplasm, Nup is the nucleoplasm, and Chr is the chromatin fraction. RPB1 was detected as a control.

Figure R2. The preinitiation complex factor TBP was labeled with AF594 and examined using stimulated emission depletion microscopy (STED) before and after auxin treatment. Representative images are shown on the left; the scale bar is 3 μ m. Cells from no fewer than 10 different fields were used to calculate the clusters per cell, and TBP clusters were identified as clusters larger than 9 pixels in the STED images (20 nm resolution). The box plots indicate the median (middle line), 25th and 75th percentiles (box), 5th and 95th percentiles (whiskers), average (purple circle) and outliers (single red points). Statistical significance was evaluated based on the two-tailed Student's t test. *** $p < 0.001$.

References:

1. Boulon, S. et al. HSP90 and its R2TP/Prefoldin-like cochaperone are involved in the cytoplasmic assembly of RNA polymerase II. *Mol Cell* 39, 912-924, doi:10.1016/j.molcel.2010.08.023 (2010).
2. Mao, Y. Q. & Houry, W. A. The Role of Pontin and Reptin in Cellular Physiology and Cancer Etiology. *Front Mol Biosci* 4, 58, doi:10.3389/fmolb.2017.00058 (2017).

Quantification data described at lines 233-239 shows very weak increases (48%, 60%, 64%). These pretty modest increases may be misleading. Can the authors explain why we should trust these results and the conclusions they lead to. Do alternate ways to do these experiments exist, leading to more convincing results?

Response: We agree with the reviewer that we did obtain a weak increase for the *in vitro* transcription, *in vitro* droplet, co-phase separation, and DNA curtain experiments. We showed that RUVBL1/2 enhances *in vitro* transcriptional activity by 48%, the condensed fraction of RPB1 CTD clusters by 60%, the condensed fraction of RPB1 CTD and EWS-FLI1 co-phase separation by 60%, and RPB1 CTD puncta counts at EWS-FLI1 puncta recruited to the GGAA regions at DNA curtains by 64%. Even though they were different experiments, they showed similar increases, and the results were also consistent with the magnitudes of changes in Pol II clustering (34%), nascent RNA synthesis (23%), and tethering effects (61%) in cells. Together, our quantification suggests that the RUVBL2-mediated effects on Pol II clustering and transcriptional activation are generally consistent,

supporting our conclusion that RUVBL2 regulates Pol II clustering. On the other hand, RUVBL2 is a coactivator that does not dramatically increase transcription compared with transcriptional activators. For example, recent publications showed that the coactivator MED1 enhanced transcription activities less than 2-fold with the same system as we used in this study³, and the corepressor RPAP2 represses transcription by approximately 65%⁴, which is a magnitude similar to ours. We have added this information to the revised manuscript.

References:

3. Chen, X. et al. Structures of the human Mediator and Mediator-bound preinitiation complex. *Science* 372, doi:10.1126/science.abg0635 (2021).
4. Wang, X. et al. RPAP2 regulates a transcription initiation checkpoint by inhibiting assembly of pre-initiation complex. *Cell Rep* 39, 110732, doi:10.1016/j.celrep.2022.110732 (2022).

Reviewer #2

Wang et al. focused on transcriptional regulation of chaperones and identified the molecular chaperones RUVBL1/2 that could interact with RNA-polymerase II (Pol II) large subunit (RPB1), as evident from ChIP-MS analysis. Although the involvement of RUVBL1/2 was suggested in transcriptional regulation through interactions with specific transcription factors, the detailed mechanism was not clear owing to their involvement in RNA processing and other biological phenomena. In this study, the authors found a direct interaction between RPB1 and RUVBL1/2 through biochemical experiments. They also found that RUVBL1/2 may promote Pol II clustering, which may be a mechanism for enhancing transcriptional activity. Furthermore, from PRO-seq analyses with the auxin degron system, they found that RUVBL1/2 could directly regulate transcription initiation. RNA-seq analysis indicated the involvement of RUVBL1/2 in post-transcriptional regulation through RNA processing and other mechanisms.

Response: We agree with the reviewer that we found a direct interaction between RPB1 and RUVBL1/2 and that RUVBL1/2 promotes Pol II clustering, which underpins a mechanism for enhancing transcriptional activity.

I recognize that the novelty of this study is high enough, as the authors show that RUVBL1/2 is involved in Pol II clustering, and suggest, through different experimental approach, that these factors might directly regulate transcription. The absence of ATP in the droplet experiments indicates that the chaperone function of RUVBL1/2 is not involved in RPB1 clustering; however, this has not been fully investigated. RUVBL1/2 have been proposed to promote clustering via RPB1 CTD, but there is insufficient evidence. In addition, the structure of the paper is not well organized and needs substantial revision, and resubmission of the paper is recommended.

Response: We are delighted that the reviewer stated that the novelty of our study is high enough. The major conclusion of our study is that RUVBL2 promotes Pol II clustering and transcription at active gene promoters. Even though we think that we provided enough evidence for this conclusion, the reviewer helped us to design some tethering experiments, which we believe would further strengthen our work. Thus, we performed the experiments described below. We also discuss how the multivalent interactions and

RPB1 CTD might together contribute to Pol II-mediated clustering and have extensively revised the manuscript by following the reviewer's detailed suggestions.

Major comment:

- RUVBL1/2 are known molecular chaperones, involved in Pol II assembly in the cytoplasm and numerous other vital functions; in the droplet experiment, addition of RUVBL1/2 promoted RPB1 clustering in the absence of ATP (Figure 2c). This indicates that the chaperone activity of RUVBL1/2 was not involved in RPB1 cluster formation. In addition, appropriate control experiments like adding mCherry instead of RUVBL1/2-mCherry were not performed; therefore, it is not possible to determine whether RUVBL1/2 promotes clustering or it is an artifact owing to an increased protein concentration. The authors probably think that chaperone activity promotes RPB1 clustering (line 644). LacR experiments with RUVBL1/2 mutants (if can be generated), that can oligomerize but have no chaperone-like activity, may show whether RUVBL1/2 chaperone activity is required for RPB1 recruitment. LacR experiments, using a mutant that suppresses oligomerization (if such mutants are available), can indicate whether oligomerization is important for RUVBL1/2 chaperon function or RPB1 recruitment.

Response: We performed the control experiments by adding mCherry proteins as the reviewer suggested in the original Figure 2E, but we showed only the quantitative signals without representative images. We have added representative images of mCherry in *in vitro* droplets with RPB1 CTD to the revised Fig. 5e and Supplementary Fig. S4a.

Based on previous literature, we found that RUVBL2 E300Q causes defects in the ATPase activity of RUVBL1/2 complex⁵. Domains DI and DIII are involved in oligomerization and ATPase activity of RUVBL1/2 complex, and domain DII is known to interact with other client partners⁶. We repeated the tethering experiments with ATPase mutants and RUVBL2 truncations and found that DI-DIII domains or ATPase mutants did not have a capacity similar to that of full-length RUVBL2 (Figure R3). The DI, DII or DIII domain of RUVBL2 could also not significantly enhanced Pol II clustering as the full-length RUVBL2 (Figure R4). These results indicate that ATPase activity, domain DI-DIII-mediated oligomerization, and domain DII might together facilitate the roles of RUVBL2 in promoting Pol II clustering examined with tethering experiments in our study. The RUVBL2 ATPase mutant enhanced RPB1 CTD clustering *in vitro* (Figure R11), it was still less efficient than wild-type RUVBL1/2. The RUVBL2 ATPase mutant did not exhibit similar enhancement of Pol II clustering in tethering experiments, suggesting that the ATPase mutant may have dominant-negative effects on Pol II clustering in cells. The detailed molecular mechanisms need further investigation in the future. We have added the new figures and discussions in the revised manuscript.

Figure R3. RUVBL2 ATPase contributed to Pol II clustering in the LacR tethering assay. **a.** The ATPase core amino acid E300 was mutated to Q in the RUVBL2 ATPase mutant (PDB: 6H7X). **b.** Representative images of tethering experiments with the RUVBL2 ATPase mutant in U2OS-2-6-3 cells. LacR-BFP served as a negative control. The scale bar is 5 μ m. **c.** Relative enrichment of Pol II at the LacO loci after tethering of wild-type or ATPase-mutated RUVBL2. More than 10 different fields were used for the calculation and statistical analyses. For the box plot, most extreme data points are covered by the whiskers, except the outliers. Two-tailed Student's t test was carried out between BFP and other conditions, **p<0.01, ns is no significant difference;

Figure R4. RUVBL2 domain dissection to identify the roles of different regions contributing to Pol II clustering. a. Diagram of RUVBL2 domains. DI, DII, and DIII indicate domains I, II and III, respectively (PDB: 6IGM). b. Representative images of tethering experiments with different RUVBL2 domains in U2OS-2-6-3 cells. Pol II was imaged by immunofluorescence with RPB1 CTD antibody. BFP was used as a negative control for tethering. The scale bar is 5 μm . c. Relative enrichment of Pol II at the LacO loci after tethering of different RUVBL2 domains. More than 10 different fields were used for the calculation and statistical analyses. Two-tailed Student's t tests were carried out to compare BFP and other conditions. ** $p < 0.01$, ns indicates no significant difference.

References:

0. Assimon, V. A. et al. *CB-6644 Is a Selective Inhibitor of the RUVBL1/2 Complex with Anticancer Activity.* *ACS Chem Biol* 14, 236-244, doi:10.1021/acscchembio.8b00904 (2019).

0. Lopez-Perrote, A., Munoz-Hernandez, H., Gil, D. & Llorca, O. *Conformational transitions regulate the exposure of a DNA-binding domain in the RuvBL1-RuvBL2 complex.* *Nucleic Acids Res* 40, 11086-11099, doi:10.1093/nar/gks871 (2012).

- Using the RPB1 degron cell line and droplet analysis, the authors conclude that RUVBL1/2 promote RPB1 clustering via RPB1 CTD. However, auxin treatment for 1 h did not completely eliminate RPB1-CTD (Figure 1I). Furthermore, in IP assays using the RPB1-NTD antibody, RUVBL1/2 binding was not completely inhibited. Therefore, it is not still clear whether RUVBL1/2 binds only to the CTD of RPB1 or to other domains within RPB1. RPB1 has a tendency to form clusters via the CTD. In addition, oligomerized RUVBL1/2 were used in this experiment, and it is conceivable that there are multiple RPB1 binding sites within the complex, promoting aggregation via multivalent interactions (Musacchio, EMBO J, 2022). One main topic of this paper is that RUVBL2 promotes Pol II clustering, and this point should be clearly emphasized.

Response: We agree with the reviewer that whether RUVBL2 binds only to the CTD of RPB1 or to other domains of RPB1 is unclear given the findings of the RPB1-CTD degron experiment. However, we do not think this conflict with our conclusion that RUVBL2 regulates Pol II clustering via the RPB1 CTD. We have provided sufficient experimental evidence of a direct interaction between RUVBL2 and CTD that regulates RPB1 CTD clustering *in vitro* and *in vivo* in the submitted manuscript. We agree with the reviewer that it is possible that multiple RPB1 binding sites within the complex promote clustering via multivalent interactions. We have toned down our statement and discussed the roles of multivalent interactions together with the RPB1 CTD in mediating Pol II clustering in our revised manuscript.

- One key claim of the paper is that "Mechanistically, RUVBL2 enhances LCD-LCD interactions between transcription factors and Pol II CTD." However, only EWS-FLI1 has been examined for co-clustering with RPB1, and therefore, it cannot be generalized as a phenomenon. ChIP-seq data show the colocalization of RUVBL1/2 with many transcription factors (Figure 7B); however, its relationship to RPB1 clustering is not clear. Indeed, while INO80 appears to co-localize on ChIP-seq (Extended Data Figure 3B), it does not interact directly with Pol II (Figure 1G). Also, the relationship to LCD-LCD

interaction has not been examined at all, and it is extremely wild to come to this conclusion. Did RUVBL2 promote EWS-FLI1 to form puncta alone? If RPB1 can bind to both EWS-FLI1 and RUVBL2, and if RUVBL2 further facilitates RPB1 recruitment, then the addition of RUVBL2 may allow a definite detection of RPB1 condensate and therefore, "RUVBL2 enhances the binding between EWS-FLI1 and Pol II CTD" is not evidenced.

Response: We agree with the reviewer that we should limit our claims to EWS-FLI1 because only EWS-FLI1 has been examined by the co-phase separation experiment with RPB1 CTD. In our LacR tethering experiments, we found that tethering RUVBL2 to the gene promoter under mock conditions did not cause evident recruitment of Pol II (Revised Figure 4E). On the other hand, we showed purified EWS-FLI1 and RPB1 CTD co-phase separation *in vitro* in the absence of RUVBL2, and their clustering effects increased after the addition of RUVBL2. We indeed found that purified RUVBL2 proteins promote EWS-FLI1 clustering *in vitro* (Figure R5), but this was not shown in the original submission. A previous publication in Dr. Zhi Qi's lab showed that the FLI1 DNA binding domain could not interact with the RPB1 CTD in the DNA curtain, but EWS-FLI1 interacted, so the colocalization of the RPB1 CTD droplet and EWS-FLI1 puncta on the DNA curtain suggests interactions with the low-complexity domains of EWS-FLI1 and RPB1 CTD⁷. In our DNA curtain experiment, we first allowed EWS-FLI1 to bind to the DNA curtain to ensure that EWS-FLI1 recruitment was the same for both conditions and then added purified RPB1 CTD with or without RUVBL2 to the working buffer. When the flow was on, the RPB1 CTD was able to form clusters in the DNA curtain without RUVBL2⁷. To show the difference with or without RUVBL2 addition, interaction efficiency was defined as the ratio of the number of magenta clusters (RPB1 CTD) divided by the number of green clusters (indicates EWS-FLI1-bound DNA). Adding RUVBL2 significantly increased the RPB1 CTD interaction efficiency (Figures 5k-i). These results collectively suggest that RUVBL2 enhances the interactions between EWS-FLI1 and the RPB1 CTD. However, we could not completely rule out the possibility that RUVBL2 may slightly recruit Pol II, which may also contribute to the co-phase separation signals and DNA curtain signals of RPB1 CTD, as the reviewer suggested. We have revised the text and discussed this in the revised manuscript.

Figure R5. *In vitro* droplet assay for RUVBL1/2-enhanced EWS-FLI1-GFP clustering. Left panel: Representative images of the droplet assay using EWS-FLI1-GFP and RUVBL1/2. Right panel: Statistical analysis of left panel with the total number of droplets examined over a one-time *in vitro* droplet experiment: N= 30 for mock and N = 155 for RUVBL1/2. Significant differences were calculated by two-tailed Student's t test, **p<0.01, and the scale bar is 5 µm. This figure is only displayed in the response.

Reference:

7. Zuo, L. et al. *Loci-specific phase separation of FET fusion oncoproteins promotes gene transcription. Nat Commun 12, 1491, doi:10.1038/s41467-021-21690-7 (2021).*

- Several unnatural points are recognized in the current structure of the paper, probably because of a drastic change in the paper structure. In particular, the degron system is explained again in Figure 5 even though it was already used in Figure 1. Therefore, the entire paper structure has to be drastically reorganized and resubmission should be considered.

Response: We highly appreciate the reviewer's detailed suggestions regarding revising our manuscript. We reorganized the entire paper by following the reviewer's suggestions.

Minor comment:

- Line 116: "we identified 45 direct transcriptional targets of the RUVBL2-Pol II axis for the first time (i.e., c-Myc and Junb, etc.)" This has been done using mouse ES cells only. Therefore, it needs to be stated that this fact was found in mESC.

Response: We revised the manuscript as the reviewer suggested.

- Line 117: " Further, we found that these genes are involved in diverse molecular functions, indicating that the RUVBL2-Pol II axis is essential for various cellular functions in different biological contexts." This comment seems overstated as complete evidences are not presented in the manuscript.

Response: We toned down this statement in the revised manuscript. We replaced the original sentence with "Furthermore, we found that these genes are involved in diverse molecular functions, and postulated that the RUVBL2-Pol II axis may contribute to various cellular functions in different biological circumstances." in the revised manuscript.

- Figure 1A: This model cannot account for the condensate-mediated transcriptional regulation claimed in this paper; it should be changed to a figure similar to that shown in Fig. 1(a) by Lim et al., *Curr Opin Genet Dev*, 2021.

Response: We modified Figure 1A by following the reviewer's suggestions.

- Line 142: The subunits of RNA pol I and III, such as POLR1C, POLR1B, and POLR3B, are included as interactors with RUVBL2 (Figure 1B). Is it conceivable that RUVBL1/2 are also involved in other RNA pol functions? If so, it should be discussed in the Discussion.

Response: We have added text indicating that RUVBL1/2 may also be involved in other RNA polymerase functions in our discussion.

- Line 145: Fig. 1C shows only overlapping peaks of RUVBL1/2, but it seems that their signal peaks do not necessarily overlap in the RUVBL1/2 ChIP-seq data in Figure 6C. Please show the quantitative data of overlap between RUVBL1 and RUVBL2 ChIP-seq (as supplementary data).

Response: We added overlapping analyses for the RUVBL1 and RUVBL2 ChIP-Seq in the supplementary figures of the revised manuscript. The results showed that they largely overlapped at active gene promoters. The peaks that did not overlap in the Venn diagram still showed obvious chromatin binding signals in one or the other ChIP-Seq (Figure R6).

Figure R6. Comparison of RUVBL1 and RUVBL2 ChIP-Seq peaks. a. The Venn diagram illustrates the number of overlapping RUVBL1 and RUVBL2 ChIP-seq peaks in this study. b. Heatmaps of RUVBL1 and RUVBL2 ChIP-Seq signals at RUVBL1-specific peak sites and RUVBL2-specific peak sites are shown.

- Line 152: Here, the authors suddenly mention about INO80 and P400. Since there are potential readers from a different and even unrelated fields, a prior explanation is necessary regarding the interaction of RUVBL1/2 with these factors.

Response: We added a prior explanation for the interactions of RUVBL1/2 with INO80 and P400 in the revised manuscript.

- Line 162, "Gerber et al," should be used to cite references and the use of full names should be avoided.

Response: We corrected this citation as the reviewer suggested.

- In Fig. 1, the name "RPB1 CTD" appears, but "PolII CTD" is used in other parts of the manuscript. The paper should use RPB1 CTD throughout.

Response: We now use RPB1 CTD throughout the manuscript as the reviewer suggested.

- Line 171: POLR2E, which corresponds to RPB5, is detected by ChIP-MS of RUVBL2 (Figure 1B). Therefore, assuming that Pol II is assembled in the cytoplasm and transported to the nucleus, discuss the possibility (if any) of RUVBL2 binding to Pol II and their co-transportation to the nucleus.

Response: We now discuss the possibility of RUVBL2 binding to Pol II and cotransport into the nucleus as suggested by the reviewer.

- Although "data not shown" has been used several times, appropriate data should be included in support of claims or comments in the Supplementary information in accordance with the submission rules of Nature communications. <https://www.nature.com/ncomms/submit/how-to-submit>

Response: We deleted the text "data not shown" and added supporting claims in the supplementary information as suggested by the reviewer.

- Line 206: The sentence containing "which is reminiscent of..." should be stated in the Discussion.

Response: We revised the manuscript as the reviewer suggested.

- Line 216: GFP has been used as a negative control, but the dose is not listed in Methods or Legend.

Response: We added the dose of GFP to the Methods and Legends as the reviewer suggested.

- Line 224: The claim "RUVBL1/2 promoted the cocondensation of EWS-FLI1 and Pol II CTD" is misleading. The Methods section states that this experiment uses human RPB1 with a hepta repeat count of 26, not 52, in the CTD. It has already been reported that the number of repeats can change RPB1 properties. Quintero-Cadena et al., Mol Cell, 2020; Sawicka et al., EMBO J, 2021. Even if the data has been used as it is, the number of hepta repeats should be clearly mentioned with the statement that the results are for RPB1 with 26 hepta repeats.

Response: Thank you for pointing out the important information about the RPB1 CTD length. Shorter hepta repeats were also used in previous publications⁸⁻¹¹. We agree with the reviewer and now clearly mention that the results are for RPB1 with 26 hepta repeats in the revised manuscript as the reviewer suggested.

References:

8.Boehning, M. et al. RNA polymerase II clustering through carboxy-terminal domain phase separation. *Nat Struct Mol Biol* 25, 833-840, doi:10.1038/s41594-018-0112-y (2018).

9.Lu, H. et al. Phase-separation mechanism for C-terminal hyperphosphorylation of RNA polymerase II. *Nature* 558, 318-323, doi:10.1038/s41586-018-0174-3 (2018).

10. Hsin, J. P. & Manley, J. L. The RNA polymerase II CTD coordinates transcription and RNA processing. *Genes Dev* 26, 2119-2137, doi:10.1101/gad.200303.112 (2012).

11. Kwon, I. et al. Phosphorylation-regulated binding of RNA polymerase II to fibrous polymers of low-complexity domains. *Cell* 155, 1049-1060, doi:10.1016/j.cell.2013.10.033 (2013).

- Figure 2k: The Roman numerals related to Figure 2] are confusing. Please use different expression patterns.

Response: We used Arabic numerals related to Figure 2] in the revised manuscript.

- Line 243: "The GFP" should be written as "the GFP".

Response: We use "the GFP" in the revised manuscript.

- In line 244, what "RUVBL2 genomic localization" refers to is not clear, and a suitable explanation is needed.

Response: The RUVBL2 genomic localization indicated the RUVBL2 ChIP-Seq signals. We have made this clear in the revised manuscript.

- Line 280: Data show that RUVBL1/2 co-localization peak and Pol II peak co-localize; however, the possibility of localization of RUVBL2 alone at the promoter has not been verified or clarified with data. RUVBL2 has been shown to localize at super-enhancers; therefore, data should be presented to show whether RUVBL2 has a real tendency to localize at the promoters.

Response: RUVBL2 localizes to promoters (Figure R7). We added the figure to the revised manuscript as the reviewer suggested.

Figure R7. Distribution of RUVBL1 and RUVBL2 peaks in various regulatory elements (upper panel) and various promoter categories (bottom panel) in mESCs. Detailed definitions of the regulatory elements and promoter categories can be found in the Methods section.

- Line 291: "Additionally, auxin treatment for..."- this sentence needs to be more specific for a clear understanding.

Response: We replaced the sentence with "Additionally, auxin treatment for 0.5~1 h led to decreased PRO-Seq signals both at the promoter-proximal regions and gene bodies for both the short and long genes, suggesting defects in transcription initiation" in the revised manuscript.

- Line 371: "Western blot analysis...". How many hours after Auxin treatment did the degradation happen? Moreover, the authors used both words, IAA and Auxin, in the manuscript. Please pick any one of them and use it consistently throughout the manuscript.

Response: We showed that RUVBL2 proteins were depleted by 59% and 65% after auxin treatment for 0.5 h and 1 h, respectively. We have made this clear and used "auxin" throughout the revised manuscript.

- Figure 5e: The labels are significantly misaligned.

Response: We have aligned the labels in the revised manuscript.

- Line 406: I did not understand the rationale behind motif analysis based on RNA-seq data when PRO-seq data was already shown; it should be re-analyzed using PRO-seq

data. RNA-seq showed little change with Auxin treatment for 0.5 or 1 h except for genes, such as c-Myc, that responded very quickly; therefore, we may not see a direct effect on transcriptional regulation by RUVBL2, and this data is unnecessary.

Response: Previous studies have revealed that RUVBL2 works with specific transcription factors by examining the expression of specific genes. We identified the promoter-associated motifs for the RUVBL2-affected genes with time-series RNA-Seq data. The results showed that the motif did not exhibit enrichment for specific factors.

- Figure 5h: The quantitative values for localization are not displayed, and therefore, the difference is not apparent.

Response: We have added quantitative values for the ChIP-Seq signals in the revised manuscript as the reviewer suggested (Figure R8).

Figure R8. The RUVBL1/2 ChIP-Seq densities (in Figure 6h) in the regions ± 200 bp of the TSSs were calculated and plotted as box plots. The Wilcoxon test was used to calculate the significance ($*p < 0.05$, $**p < 0.01$, $***p < 0.001$, ns indicates no significant difference). All significance levels were calculated by comparison with cluster 1. The numbers below the asterisk are the fold changes compared with cluster 1.

- Figure 5j, I would like to see the percentages of genes included in each cluster, as the number of genes included in each cluster is different to begin with. However, on an average, only about 30% of the genes overlapped. Therefore, I did not find the basis for saying that they "largely overlapped."

Response: We added the percentages of genes included in each cluster in the revised manuscript as the reviewer suggested.

- Line 435: "smg1" should be written as "SMG1".

Response: We corrected "SMG1" in the revised manuscript as the reviewer suggested.

- Line 440: This part should be moved closer to the citation of Figure 1g, j in the text.

Response: We moved the sentences in Line 440 to the sections before the citation of Figures 1G-J in the revised manuscript as suggested by the reviewer.

- Line 450: "RUVBL2 also interacted..." This should be stated before the citation of Figure 5K, L in the text.

Response: We moved the sentence in Line 450 to the sections before the citation of Figures 5K-L in the revised manuscript as suggested by the reviewer.

• Lines 462-463: The corresponding Extended Data Figure 3d-e data are missing.

Response: We mis-cited the figure. We have corrected it to Figure 3D in the revised manuscript.

• Line 485: The use of the word “Here” is incorrect because the PRO-seq data is already shown.

Response: We deleted “Here” from the revised manuscript.

• Figure 6B and Extended Data Figure 4C: 0 h (Dox) data should be shown.

Response: The 0 h (Dox) condition is not shown because the fold changes of the indicated condition relative to the “0 h (Dox)” condition are represented in the manuscript. We have made this clear in the figure legend of the revised manuscript.

• Line 555: In the testis, RUVBL2 was highly expressed, and not overexpressed.

Response: We changed “overexpressed” to “highly expressed” in the revised manuscript.

• Figure 6h: Figures should be prepared according to their order of appearance in the text.

Response: We moved the original Figure 6h to the end of Figure 8, and the original Figure 7l was moved to Suppl. Figure S8g in the revised manuscript.

• The first three sentences from line 606 should be stated in the Introduction. Otherwise, the reader will not understand the importance of this paper.

Response: We moved the first three sentences from line 606 to the introduction part as suggested by the reviewer.

• In figures using Degron experiments (e.g., Figure 3b), the control experiment is denoted as “Dox”. The sample to which Auxin was added also contained Dox, labeling it as “0 h” will be easy to understand.

Response: We corrected the “Dox” label to “0 h” as suggested by the reviewer across the revised manuscript.

• I do not understand why the analysis in Figure 1J focused on c-Myc, Slc16a3, and Junb as RUVBL2 target genes. They are direct target genes identified from PRO-seq and other analyses, but this has not been mentioned in the text and Fig. 1J. Therefore, an explanation is needed.

Response: The c-Myc, Slc16a3, and Junb genes have good enrichment of RUVBL2 ChIP-Seq signals at their gene promoters; thus, we selected them for qPCR validation. We mention this in the revised manuscript. These experiments were previously requested by the reviewers for revision in another journal. We regret the drastic change in the paper structure.

- Line 774: “mEGFP was used as the control, and RUVBL1 and RUVBL2 were examined to evaluate the protein–protein interaction frequencies.” I do not understand what “frequencies” refers to.

Response: We apologize for the confusion. We have deleted “frequencies” from the revised manuscript.

- I do not understand what is represented by the data in Fig. 2E.

Response: Fig. 2e shows a different quantitation method for the *in vitro* droplet results. The condensed fraction indicated the amount of material in droplets relative to the total material in the mixture. We have made this clear in the revised manuscript.

- Fig. 2L: A detailed explanation of what is being quantified is needed in the legend. Moreover, there is a reference to "red bar" in the legend, but no red bar is noticed in the figure.

Response: We have added details for the quantification in the legend of the revised manuscript. The red bar is deleted in the legend.

- Fig. 3A: the statement "DNA was stained with DAPI." is not necessary because the image of DAPI staining is not shown.

Response: We deleted the statement regarding DAPI from the revised manuscript.

- Fig. 3B: "Scale bar, 3 μ m." should be put in Fig. 3A.

Response: We moved “Scale bar, 3 μ m” to Fig. 3a in the revised manuscript.

- Fig. 3F, (left) and (right) may be mistaken for (upper) and (lower). Moreover, I do not understand know what the vertical axis of the boxplot indicates.

Response: The vertical axis indicates the densities of the PRO-Seq signals. We corrected the figure legend as the reviewer suggested.

- Fig. 4E: Does the term "endogenous antibodies" mean antibodies that recognize endogenous proteins?

Response: We corrected the statement to “antibodies that recognize endogenous proteins” in the revised manuscript.

- Fig. 5H: “in the methods section” should be presented as “in the Methods section”. A similar notation has been found elsewhere and needs to be corrected.

Response: We corrected this error throughout the revised manuscript as the reviewer suggested.

- Fig. 7B: The CTNNB1 motif is not shown in Fig. 7A and should be added here.

Response: We searched the motif from Homer and Jaspar motif databases with RUVBL1/2 bound peaks. The CTNNB1 motif is not in these databases, which is the reason that we did not show it in Figure 7a. RUVBL1 has been shown to directly interact with β -catenin in a previous study¹², which is consistent with the observation that the co-localization of RUVBL1/2 with CTNNB1 on chromatin in mESCs as shown in Figure 8b.

Reference:

12. Bauer, A., Huber, O. & Kemler, R. *Pontin52, an interaction partner of beta-catenin, binds to the TATA box binding protein. Proc Natl Acad Sci U S A 95, 14787-14792, doi:10.1073/pnas.95.25.14787 (1998).*

- Methods section: Please describe the origin of the mESC strain.
- Methods section: Please include information on the antibodies used.
- Methods section: In the "LacR tether imaging and data analysis" section, "IFA" has been mentioned without defining the term.
- Methods section: In describing GFP-EWS-FLI1 and CTD droplet assay spinning-disk confocal microscopy (Leica TCS SP8, 100x Oil) has been mentioned. I guess that SP8 is not a spinning disk confocal microscope.
- The abbreviation BPB has not been defined in full.
- In the section "MNase-Seq library preparation", the description is in protocol form. In the paper, it should be described in the past tense. Also, "gel extraction kit" has not been described in detail. In addition, the sources and model numbers of the reagents used have not been described in several instances, therefore, making it impossible for the reader to conduct reproducible experiments.
- PRO-seq analysis, "To get the more accurate transcription changes, the spike-in of drosophila cells was used to normalized the PRO-Seq signals." A detailed explanation of this comment is necessary.
- deepTools (Ramirez et al., 2014): The citation format is incorrect.

Response: We highly appreciate the reviewer's detailed suggestions again. We have extensively revised the Methods section by following the reviewer's suggestions.

Reviewer #3

In this manuscript, the authors showed that RUVBL2 is involved in the regulation of Pol II cluster formation and transcription regulation. At first, the authors performed Pol II ChIP-MS using ES cells and identified many of proteins involved in transcription regulation, RNA splicing, cell cycle regulation and other cellular functions. Among the identified proteins, the authors focused on RUVBL1 and RUVBL2. The authors showed that RUVBL2 interacts with unphosphorylated Pol II CTD in vitro and promotes Pol II CTD clustering and transcription activation in cells. Rapid depletion of RUVBL2 using auxin-induced degradation system leads to the decreased number of Pol II cluster formation and inhibits global transcription in cells. Of note, tethering of RUVBL2 enhances Pol II clustering at the active promoters in cells. Furthermore, the authors performed both RNA-seq and PRO-seq analysis and identified direct target genes of the RUVBL2, which include c-Myc, Junb and other genes. Many of these genes are hallmarks of cancers and encode proteins with diverse cellular functions. These results suggest novel activity for RUVBL2 in regulating transcription through Pol II cluster formation in the nucleus. Based on their findings, the authors propose that RUVBL2 functions as a "molecular chaperon" for Pol II clustering, however there is not sufficient evidence supporting this model.

Response: We agree with the reviewer that our results suggest a novel activity for RUVBL2 in regulating Pol II clustering and transcriptional activation with diverse transcription factors, which is the major conclusion and key point of our work. We regret that we did not fully understand the terminology "molecular chaperon" and misused it in

the submitted manuscript. Based on the literature that we found, RUVBL2 is an ATPase that has been proposed to function as a protein assembly chaperone for multisubunit complexes and functions as a potential protein disaggregase in mammalian cells^{13,14}. As stated by the three reviewers, our findings of RUVBL2 in regulating Pol II clustering and transcriptional activation with diverse transcription factors are new and novel. The detailed activities of RUVBL2 in regulating Pol II clustering are not the major focus of the current study. We discuss the protein chaperon-like potential of RUVBL2 in the revised manuscript.

Pol II foci examined by time-correlated photoactivated localization microscopy (tcPALM) are tiny and dynamic and form before initiation¹⁵. Pol II foci observed by immunofluorescence in fixed cells colocalized with nascent RNAs, suggesting that they may be elongating Pol II¹⁶. Transcription inhibition led Pol II to localize to the speckles, which may not be directly related to transcription^{16,17}. Thus, it is crucial to examine the small, dynamic Pol II clusters with tcPALM after acute depletion of RUVBL2. mEos3.2 was tagged into the N-terminus of RPB1, which did not affect the protein level of RPB1 in mESCs. Our Pol II tcPALM live-cell imaging exhibited a good signal-to-noise ratio and displayed both transient and stable clusters, as reported previously¹⁸. Strikingly, the acute depletion of RUVBL2 significantly decreased the number of both transient and stable clusters within 1 h after RUVBL2 depletion (Figure R9). We also performed initiation factor TBP super resolution imaging and found that RUVBL2 also contributed to preinitiation cluster formation in the nucleus (Fig. 3d, Supplementary Fig. 3d). This result demonstrates that RUVBL2 is necessary for the formation of preinitiation Pol II clusters in living cells.

To provide direct evidence of RUVBL2-mediated Pol II clustering at chromatin for the RUVBL2 target genes, we performed Pol II immunofluorescence and RNA FISH after immediate depletion of RUVBL2. *Bmp4* is a TGF-beta signaling factor and RUVBL2-directly targeted, actively transcribed gene, and its nascent RNA synthesis was dramatically decreased after RUVBL2 depletion, as measured by PRO-Seq. We then performed Pol II IF and RNA FISH with probes against *Bmp4* nascent transcripts within 1 h after auxin treatment. The results showed that RUVBL2 depletion significantly decreased Pol II clustering and nascent RNAs at the *Bmp4* locus (Figure R10). Together, these results demonstrate that RUVBL2 directly regulates Pol II clusters at a specific genomic locus.

We believe these evidence further strengthens our current conclusion.

Figure 9. RUVBL2 regulates the initiation Pol II clusters.

a. Super resolution image of endogenous Pol II labeled with mEos3.2 in living RUVBL2 degran mESCs (left) and representative images of transient and stable Pol II clusters and corresponding time-correlated photoactivation localization microscopy (tcPALM) traces (right). Transient and stable Pol II clusters correspond to areas boxed in blue and yellow, respectively. The dashed box indicates the nucleus; scale bars, 1 μ m in the whole cell in the super resolution image (left) and 200 nm in the transient and stable Pol II cluster images (right).

b. Histogram displaying the number of transient Pol II cluster probability distributions observed before and after RUVBL2 degradation.

c. Box-plot illustrating the transient clusters per cell ($n=25$ for auxin treatment 0 h and $n=37$ for auxin treatment 1 h) were calculated, and two-tailed Student's t tests were used for statistical analysis (box plot), $**p<0.01$.

d. The same as b, but displayed for the stable Pol II clusters.

e. The same as c, but displayed for the stable Pol II clusters. $n=25$ for auxin treatment 0 h and $n=37$ for auxin treatment 1 h were calculated and statistically analyzed. Two-tailed Student's t test was used to calculate the statistical significance. $**p<0.01$.

Figure 10. The RUVBL2 direct target gene Bmp4 was examined by single-molecule RNA FISH combined with RPB1 immunofluorescence. Representative images are shown, and the signals of the Pol II (RPB1 IF) cluster overlapping with nascent RNA foci were measured and statistically analyzed (two-tailed Student's t test) from no fewer than 10 different fields. The scale bar is 3 μ m, ** $p < 0.01$.

References:

13. Narayanan, A. et al. A first order phase transition mechanism underlies protein aggregation in mammalian cells. *Elife* 8, doi:10.7554/eLife.39695 (2019).
14. Zaarur, N. et al. RuvbL1 and RuvbL2 enhance aggresome formation and disaggregate amyloid fibrils. *EMBO J* 34, 2363-2382, doi:10.15252/embj.201591245 (2015).
15. Cisse, II et al. Real-time dynamics of RNA polymerase II clustering in live human cells. *Science* 341, 664-667, doi:10.1126/science.1239053 (2013).
16. Zeng, C., Kim, E., Warren, S. L. & Berget, S. M. Dynamic relocation of transcription and splicing factors dependent upon transcriptional activity. *EMBO J* 16, 1401-1412, doi:10.1093/emboj/16.6.1401 (1997).
17. Bregman, D. B., Du, L., van der Zee, S. & Warren, S. L. Transcription-dependent redistribution of the large subunit of RNA polymerase II to discrete nuclear domains. *J Cell Biol* 129, 287-298, doi:10.1083/jcb.129.2.287 (1995).
18. Cho, W. K. et al. Mediator and RNA polymerase II clusters associate in transcription-dependent condensates. *Science* 361, 412-415, doi:10.1126/science.aar4199 (2018).

Major comments:

(1) In this manuscript, critical point which should be addressed in the paper is whether RUVBL2 functions as a “molecular chaperon” for Pol II clustering, however the authors did not show the evidence supporting this important point. At least, the authors should test whether ATPase activity of RUVBL2 is required for Pol II clustering (Pol II foci formation), assembly of a variety of transcription factors with Pol II at RUVBL2 target genes (ChIP-seq), and transcription activation of these target genes (RNA-seq, PRO-seq). I recommend taking advantage of the mutant form of RUVBL2 which loses the ATPase activity and replacing the endogenous RUVBL2 with the exogenously expressed mutant

form of RUVBL2. Otherwise, some specific inhibitors for RUVBL2 are also useful for these experiments.

Response: We agree with the reviewer that it would be interesting to determine whether RUVBL2 functions as a “molecular chaperone” for Pol II clustering, but the major point of our paper is that RUVBL2 regulates Pol II clustering, as indicated by our title, abstract and numerous lines of evidence. There are two potential mechanisms of RUVBL2 for Pol II clustering regulation: ATPase activity and oligomerization activity, as inspired by the suggestions of reviewer 2. We believe that *in vitro* droplet assays and LacR tethering experiments with ATPase or oligomerization mutants may additionally strengthen our major conclusion. Regrading whether the ATPase activity related to the variety of transcription factors with Pol II at RUVBL2 target genes (ChIP-Seq) and transcriptional activation of these target genes (RNA-Seq, PRO-Seq) would not provide direct insights into the major point of Pol II clustering. Thus, we believe these sequencing experiments are warranted for further characterization of ATPase activity in the future.

For revision, we performed an RPB1 CTD droplet assay with a purified RUVBL1/2 ATPase mutant and an oligomerization mutant. The results showed that the RUVBL1/2 ATPase mutant could promote RPB1 CTD clustering (Figure R11), but it is slightly less efficient than wild-type RUVBL1/2. This phenomenon did not show differences in the presence or absence of ATP at the working concentration reported previously¹⁹ (Figure R11-12). Moreover, we also performed *in vitro* droplet assays with RUVBL1/2 DI-DIII (known to function in oligomerization) or the RUVBL1/2 DII domain, which is known to interact with client partners by crystal structures of RUVBL1/2 reported previously²⁰. Interestingly, the DII and RPB1 CTD clusters almost completely colocalized, and DI-DIII localized to the periphery of the RPB1 CTD clusters, suggesting that DII appears to directly interact with RPB1 CTD, while DI-DIII may contribute to the regulation and interaction at the periphery of the clusters. The results further showed that both oligomerization and RUVBL2 domain DII facilitate the enhancement of RPB1 CTD clustering *in vitro*, but their capacities are still weaker than those of full-length RUVBL1/2 (Figure R11). Together with the LacR tethering experiment with RUVBL2 truncations, these experiments provided direct insights into RUVBL2 in RPB1 CTD clustering *in vivo* and *in vitro*.

It is important to note that the *in vitro* droplet assay included only purified proteins, and the tethering experiments still had wild-type endogenous RUVBL1/2 proteins in cells and tethering of 256 repeats. The limitations of both assays may have created some biases for the different systems. Actually, the RUVBL2 ATPase and truncation mutants partially enhanced RPB1 clustering *in vitro* but not in the tethering experiment, indicating that these mutants may have exerted dominant-negative effects on Pol II clustering in cells. The detailed molecular mechanisms need further investigations. These results collectively imply that ATPase activity, domain DII-mediated interaction with Pol II, and domain DI-DIII-mediated oligomerization might together facilitate the roles of RUVBL2 in promoting Pol II clustering. We added the new figures, limitations, and discussions to the revised manuscript.

Figure 11. The domains of RUVBL2 (DI, or DI-DIII) slightly enhance RPB1 CTD clustering *in vitro*. Representative images of different RUVBL2 domains in the RPB1-CTD droplet assay (left). RUVBL1/2 indicate the copurified RUVBL1 and RUVBL2 proteins. mRUVBL1/2 indicates the RUVBL1 and RUVBL2 ATPase mutant. RUVBL1/2 DII indicates purified RUVBL1 DII and RUVBL2 DII proteins. RUVBL1/2 DI-DIII indicates copurified RUVBL1 DI-DIII and RUVBL2 DI-DIII proteins. For the detailed purification procedure, see the Methods section. The scale bar is 20 μ m. Condensed fractions of RPB1 CTD signals under the conditions displayed in the right panel. The condensed fraction was calculated for more than 5 different fields and statistically analyzed using two-tailed Student's t test for comparison with the mCherry control. * $p < 0.05$, ** $p < 0.01$, **** $p < 0.0001$, ns indicates no significant difference.

Figure 12. Representative images of the ATPase mutated RUVBL1/2 proteins in the RPB1-CTD droplet assay (left). The scale bar is 20 μm . Condensed fractions of RPB1 CTD signals under the conditions displayed in the right panel. The results of statistical analysis of no fewer than 5 different fields are shown. The condensed fraction indicates the amount of RPB1 CTD in droplets relative to the total RPB1 CTD in the mixture. Two-tailed Student's t tests were performed for comparison with the mCherry control. **** $p < 0.0001$.

References:

1. Zhou, C. Y. et al. Regulation of Rvb1/Rvb2 by a Domain within the INO80 Chromatin Remodeling Complex Implicates the Yeast Rvbs as Protein Assembly Chaperones. *Cell Rep* 19, 2033-2044, doi:10.1016/j.celrep.2017.05.029 (2017).
2. Gorynia, S. et al. Structural and functional insights into a dodecameric molecular machine - the RuvBL1/RuvBL2 complex. *J Struct Biol* 176, 279-291, doi:10.1016/j.jsb.2011.09.001 (2011).

(2) The authors propose the model that RUVBL2 directly promotes Pol II CTD clustering by enhancing LCD (Low complexity domain)-LCD interaction. To support the idea, the authors performed the *in vitro* droplet formation assay and showed that RUVBL1/2 promotes liquid droplets formation of Pol II CTD in RUVBL1/2 dose dependent manner (Figure 2). Did the author test whether the inhibitors of phase separation such as increased NaCl concentration or 1,6-hexanediol inhibits the droplets formation of Pol II CTD *in vitro*? In addition, does the ATPase mutant form of RUVBL1/2 lose the activity to promote droplet formation of Pol II CTD *in vitro*? In addition, the authors found that depletion of RUVBL2 decreased the liquid droplets of Pol II and CTCF in cells (Figure 3). Did the authors test whether 1,6-hexanediol inhibits these droplets formation in cells?

Response: We agree with the reviewer that it would be informative to test whether the ATPase mutant lost the ability to promote RPB1 CTD clustering *in vitro* and performed this experiment in the revised manuscript, as described above. For the experiments, an increased NaCl concentration or 1,6-hexanediol was used for the RPB1 CTD *in vitro* droplet assay as reported in the literature⁸. We set up the RPB1 CTD droplet assay by following the experimental protocols published in the same literature. Whether 1,6-

hexanediol treatment affects Pol II and CTCF clustering has also been previously investigated^{8,21-24}. Additionally, these NaCl and 1,6-hexanediol experiments have been performed by other groups and could not provide direct insights to aid our understanding of RUVBL2-mediated Pol II clustering. As RPB1 CTD clustering is affected by ion interactions (NaCl) and hydrophobic interactions (1,6-hexanediol), we have added this information and discuss how ion interactions and hydrophobic interactions might contribute to RUVBL2-mediated Pol II clustering in the revised manuscript.

References:

8. Boehning, M. et al. RNA polymerase II clustering through carboxy-terminal domain phase separation. *Nat Struct Mol Biol* 25, 833-840, doi:10.1038/s41594-018-0112-y (2018).
3. Shao, W. et al. Phase separation of RNA-binding protein promotes polymerase binding and transcription. *Nat Chem Biol* 18, 70-80, doi:10.1038/s41589-021-00904-5 (2022).
4. Duster, R., Kalthener, I. H., Schmitz, M. & Geyer, M. 1,6-Hexanediol, commonly used to dissolve liquid-liquid phase separated condensates, directly impairs kinase and phosphatase activities. *J Biol Chem* 296, 100260, doi:10.1016/j.jbc.2021.100260 (2021).
0. Sabari, B. R. et al. Coactivator condensation at super-enhancers links phase separation and gene control. *Science* 361, doi:10.1126/science.aar3958 (2018).
5. Lee, R. et al. CTCF-mediated chromatin looping provides a topological framework for the formation of phase-separated transcriptional condensates. *Nucleic Acids Res* 50, 207-226, doi:10.1093/nar/gkab1242 (2022).

(3) The authors performed the *in vitro* transcription assay and showed that RUVBL1/2 activates the transcription initiation (Figure 2G), however the enhancement by RUVBL1/2 is very little in the *in vitro* transcription assay. Here, the critical question is how RUVBL1/2 activates transcription *in vitro*? The author should explain this mechanistic point in the manuscript. In addition, is the ATPase activity of RUVBL1/2 required for transcription activation *in vitro*?

Response: Our *in vitro* droplet assay revealed that RUVBL2 enhances RPB1 CTD clustering. Our co-phase separation and DNA curtain experiments revealed that RUVBL2 enhances the interactions between the transcription factor EWS-FLI1 and the RPB1 CTD. We believe that the enhancement of RUVBL2 in RPB1 CTD clustering and the interaction between RPB1 CTD and other transcription regulators underlie the transcriptional activation of RUVBL2. Answering the question of whether the ATPase activity of RUVBL2 is required for transcriptional activation *in vitro* cannot provide direct insights into RUVBL2-mediated Pol II clustering; thus, further characterization of ATPase activity is warranted in the future. We toned down our statement in the revised manuscript.

(4) The authors showed that RUVBL1/2 is required for cell growth and involved in carcinogenesis (Figure 5 and Figure 7). Does the ATPase activity of RUVBL1/2 contribute to cell growth and carcinogenesis?

Response: RUVBL1/2 is highly expressed in cancer and has previously been shown to be important for carcinogenesis^{2,25,26}. The ATPase activity of RUVBL1/2 is a target for inhibitors. Studies with multiple RUVBL1/2 ATPase inhibitors have shown that the

ATPase activity of RUVBL1/2 is required for cell growth and carcinogenesis^{5,27-29}. We have made this clear in the revised manuscript.

References:

2. Mao, Y. Q. & Houry, W. A. *The Role of Pontin and Reptin in Cellular Physiology and Cancer Etiology. Front Mol Biosci* 4, 58, doi:10.3389/fmolb.2017.00058 (2017).
5. Assimon, V. A. et al. *CB-6644 Is a Selective Inhibitor of the RUVBL1/2 Complex with Anticancer 20. Activity. ACS Chem Biol* 14, 236-244, doi:10.1021/acscchembio.8b00904 (2019).
25. Santarius, T., Shipley, J., Brewer, D., Stratton, M. R. & Cooper, C. S. *A census of amplified and overexpressed human cancer genes. Nat Rev Cancer* 10, 59-64, doi:10.1038/nrc2771 (2010).
26. Huber, O. et al. *Pontin and reptin, two related ATPases with multiple roles in cancer. Cancer Res* 68, 6873-6876, doi:10.1158/0008-5472.CAN-08-0547 (2008).
27. Grigoletto, A., Neaud, V., Allain-Courtois, N., Lestienne, P. & Rosenbaum, J. *The ATPase activity of reptin is required for its effects on tumor cell growth and viability in hepatocellular carcinoma. Mol Cancer Res* 11, 133-139, doi:10.1158/1541-7786.MCR-12-0455 (2013).
28. Nano, N. et al. *Sorafenib as an Inhibitor of RUVBL2. Biomolecules* 10, doi:10.3390/biom10040605 (2020).
29. Yenerall, P. et al. *RUVBL1/RUVBL2 ATPase Activity Drives PAQosome Maturation, DNA Replication and Radioresistance in Lung Cancer. Cell Chem Biol* 27, 105-121 e114, doi:10.1016/j.chembiol.2019.12.005 (2020).

Minor comments:

(1) In this manuscript, the main story is that RUVBL2 functions as a molecular chaperon for Pol II clustering and plays a role in transcription activation of the target genes, however there are the results shown in the Figure 5, which do not directly or positively support the author's main story. I recommend moving the Figure 5 to supplementary figure for better understanding the main story.

Response: We apologize for not clearly and accurately delivering the major point of our work and that the reviewer is confused that the main story of our manuscript is that RUVBL2 functions as a molecular chaperon for Pol II clustering. As is clearly stated in the title and abstract, the main point of our work is that RUVBL2 regulates Pol II clustering and transcriptional activation. The results in original Figure 5 are important and should be kept in the main figure because these time-course RNA-Seq analyses revealed the molecular cascades of the RUVBL2-regulated gene network, which provide clear functional insights into the RUVBL2-mediated gene expression. Previous investigations of RUVBL2 have relied on examining specific gene expression or long-term perturbations, and they have been shown to respond to specific transcription factors as well. The protein degradation coupled with RNA-Seq experiments in our study showed that genes with different functions respond with different kinetics after RUVBL2 depletion, identified genes regulated by different transcription factors, and revealed the posttranscriptional regulation of RUVBL2. This information would be informative for the researchers who are interested in RUVBL2.

(2) The authors performed PRO-seq analysis and showed that depletion of RUVBL2 decreased the global transcription (Figure 3), however the depletion of RUVBL2 little affected the transcription of direct target genes of RUVBL2. The authors should explain the reason.

Response: The PRO-Seq analyses in Figure 3 indeed showed that RUVBL2 depletion led to a global decrease in nascent RNA synthesis, while the averaged signals decreased by 23% based on our calculation. The cutoff for the fold changes was a decrease of more than 50% for the identification of direct target genes. On the other hand, RUVBL2 is a coactivator for gene expression, which would not dramatically enhance gene expression, like the Mediator complex reported previously³⁰⁻³². We added this explanation to the revised manuscript.

References:

30. Henninger, J. E. et al. RNA-Mediated Feedback Control of Transcriptional Condensates. *Cell* 184, 207-225 e224, doi:10.1016/j.cell.2020.11.030 (2021).
31. Jang, Y. et al. MED1 is a lipogenesis coactivator required for postnatal adipose expansion. *Genes Dev* 35, 713-728, doi:10.1101/gad.347583.120 (2021).
32. Ito, K. et al. Critical roles of transcriptional coactivator MED1 in the formation and function of mouse adipose tissues. *Genes Dev* 35, 729-748, doi:10.1101/gad.346791.120 (2021).

(3) In Figure 7, the author proposes that RUVBL2 regulates the transcription of direct target genes with a variety of transcription factors including c-Myc and E2F1. Does the RUVBL2 contribute to recruitment of those transcription factors to the promoters of direct target genes?

Response: Multiple previous studies have shown that RUVBL2 interacts with specific transcription factors², and our motif analyses and genomic colocalization and ChIP-MS experiments also indicated that RUVBL2 functions with a variety of transcription factors. This is also the reason why RUVBL2 functions as a coactivator. Multiple publications have shown that RUVBL1/2 do not contribute to the recruitment of transcription factors to gene promoters, such as c-Myc and E2F1^{2,33,34}. Therefore, these previous studies may satisfy the reviewer's curiosity.

References:

2. Mao, Y. Q. & Houry, W. A. The Role of Pontin and Reptin in Cellular Physiology and Cancer Etiology. *Front Mol Biosci* 4, 58, doi:10.3389/fmolb.2017.00058 (2017).
33. Gnatovskiy, L., Mita, P. & Levy, D. E. The human RVB complex is required for efficient transcription of type I interferon-stimulated genes. *Mol Cell Biol* 33, 3817-3825, doi:10.1128/MCB.01562-12 (2013).
34. Boo, K. et al. Pontin functions as an essential coactivator for Oct4-dependent lincRNA expression in mouse embryonic stem cells. *Nat Commun* 6, 6810, doi:10.1038/ncomms7810 (2015).

REVIEWERS' COMMENTS

Reviewer #1 (Remarks to the Author):

This revised version of the manuscript entitled "The transcriptional coactivator RUVBL2 regulates Pol II clustering with diverse transcription factors" by Xiong Ji and colleagues has been significantly improved as compared to the original version. My previous comments/suggestions have been taken into account to my satisfaction. I think this study importantly advances our knowledge on transcriptional regulation and the role of RUVBL proteins in the process.

Reviewer #2 (Remarks to the Author):

The manuscript is of publication quality, pending some minor changes as described below.

Line 153: Fig. 1d appears after Fig. 1a in the text. In this case, Fig. 1d should probably be changed to Fig. 1b.

Fig. 6h: As the reviewer pointed out, the percentages within each cluster are supposed to be shown, but the legend does not indicate this. Therefore, readers don't know what the percentages indicate. An appropriate explanation needs to be added to the legend.

Line 656 : "ESCs" should be "mESCs".

Line 820: "red fluorescence" should be changed to "immunofluorescence".

Fig. 2e. It is not clear what the convex bottom graph indicates; if it indicates anti-sense strand data, the legend needs to be explained so that it can be understood. Also, if the data is anti-sense data, it is strange that density is negative, so there is a need to correct the axis. The same applies to Fig. 2g.

Fig. 2f. Are the sense and anti-sense data combined in this figure? This needs to be clearly stated in the legend so that it is clear.

Lines 894, 898 and Supplementary Fig. 5a legend : Does the term "endogenous antibodies" mean antibodies that recognize endogenous proteins?

Fig. 4d : Looking at the width of the bar indicating the standard deviation, I am not sure if there is really no significant difference. It is necessary to check.

Fig. 5j: If (1)~(6) in the panel are not specifically mentioned in the text or in the legend, please delete them; they look related to the symbols in Fig. 5k and cause confusion.

Supplementary Fig. 3a legened. It says "genotyping assay," but the reader can't tell anything from this figure. In the first place, it does not provide any necessary information such as whether this is an electrophoresis of PCR products, which position primers were used, what size band is expected to result from it, and how many base pairs of bands would be expected to appear if not knocked in. This should be corrected appropriately.

Supplementary Fig. 4e. If the comparison is with BFP, then the "ns" above the BFP data is not necessary.

Supplementary Fig. 4j legend. I think "Supplementary Fig. 4k." is a mistake for "Supplementary Fig. 4i."

Supplementary Fig. 4l legend. I think "Supplementary Fig. 4i." is a mistake for "Supplementary Fig. 4k."

Methods, line 407, "FiJi" should be "Fiji".

Reviewer #3 (Remarks to the Author):

The authors responded most of my concerns and now the manuscript is greatly improved. I recommend to accept the revised manuscript.

Reviewer #1 (Remarks to the Author):

This revised version of the manuscript entitled "The transcriptional coactivator RUVBL2 regulates Pol II clustering with diverse transcription factors" by Xiong Ji and colleagues has been significantly improved as compared to the original version. My previous comments/suggestions have been taken into account to my satisfaction. I think this study importantly advances our knowledge on transcriptional regulation and the role of RUVBL proteins in the process.

Response: Thanks for your appreciation of our work.

Reviewer #2 (Remarks to the Author):

The manuscript is of publication quality, pending some minor changes as described below.

Response: Thanks for your support for the publication of our work.

Line 153: Fig. 1d appears after Fig. 1a in the text. In this case, Fig. 1d should probably be changed to Fig. 1b.

Response: The Fig. 1d is actually "Supplementary Fig. 1d". We have corrected it in the revised manuscript.

Fig. 6h: As the reviewer pointed out, the percentages within each cluster are supposed to be shown, but the legend does not indicate this. Therefore, readers don't know what the percentages indicate. An appropriate explanation needs to be added to the legend.

Response: We have added the statement "The percentages in the bracket were the ratio of the number of genes in each cluster as shown in Fig. 6e" to the revised figure legend.

Line 656: "ESCs" should be "mESCs".

Response: We have corrected it.

Line 820: "red fluorescence" should be changed to "immunofluorescence".

Response: We have changed it.

Fig. 2e. It is not clear what the convex bottom graph indicates; if it indicates anti-sense strand data, the legend needs to be explained so that it can be understood. Also, if the data is anti-sense data, it is strange that density is negative, so there is a need to correct the axis. The same applies to Fig. 2g.

Response: We have changed the negative to the positive number below the 0 axis as shown in the revised Fig. 2e and Fig. 2g, and added the clarification "The profile above 0 indicates the signal on the sense strand, the profile below 0 indicates the signal on the antisense strand" in the revised legend.

Fig. 2f. Are the sense and anti-sense data combined in this figure? This needs to be clearly stated in the legend so that it is clear.

Response: The merged sense and antisense PRO-Seq density were shown here. We have clearly stated it in the revised figure legend.

Lines 894, 898 and Supplementary Fig. 5a legend: Does the term "endogenous antibodies" mean antibodies that recognize endogenous proteins?

Response: The endogenous antibodies indicate the antibodies that recognize endogenous proteins. We used it to distinguish the GFP tag antibody and the antibody against the protein self (endogenous antibodies). We changed the "endogenous antibodies" to "antibodies that recognize endogenous proteins" in the revised manuscript.

Fig. 4d : Looking at the width of the bar indicating the standard deviation, I am not sure if there is really no significant difference. It is necessary to check.

Response: We have double-checked the comparative test and labeled the p value on the panel, and showed all the points in the revised figure. There are no significant differences among the groups, although a few out-of-layer points in the BFP group may lead to the average value seeming a little bigger than others.

Fig. 5j: If (1)~(6) in the panel are not specifically mentioned in the text or in the legend, please delete them; they look related to the symbols in Fig. 5k and cause confusion.

Response: We deleted item (1)-(6) in Fig. 5j, as suggested by the reviewer.

Supplementary Fig. 3a legened. It says "genotyping assay," but the reader can't tell anything from this figure. In the first place, it does not provide any necessary information such as whether this is an electrophoresis of PCR products, which position primers were used, what size band is expected to result from it, and how many base pairs of bands would be expected to appear if not knocked in. This should be corrected appropriately.

Response: We have drawn the graphic diagram to illustrate the genotyping assay with the labeling of the positions of PCR primers. The genotyping primers "F1/R1" were located lateral side of the homologous arm, respectively. If correctly knocked in, the F1/R1 PCR products of mEos3.2-RPB1 would be expected to a band close to the 3k DNA ladder in the agarose gel electrophoresis, while wildtype RPB1 products are neared to 1k DNA ladder, we have added this information to the revised figure legend.

Supplementary Fig. 4e. If the comparison is with BFP, then the "ns" above the BFP data is not necessary.

Response: We have removed the "ns" above the BFP.

Supplementary Fig. 4j legend. I think "Supplementary Fig. 4k." is a mistake for "Supplementary Fig. 4i."

Response: We have corrected it.

Supplementary Fig. 4l legend. I think "Supplementary Fig. 4i." is a mistake for "Supplementary Fig. 4k."

Response: We have corrected it.

Methods, line 407, "FiJI" should be "Fiji".

Response: We have corrected it.

Reviewer #3 (Remarks to the Author):

The authors responded most of my concerns and now the manuscript is greatly improved. I recommend to accept the revised manuscript.

Response: Thanks for your support for the publication of our work.